



# Technical Note: An illustrative introduction to the domain dependence of spatial Principal Component patterns

Christian Lehr[1,2] and Tobias L. Hohenbrink[3,4]

[1] Leibniz Centre for Agricultural Landscape Research (ZALF), Müncheberg, Germany
[2] University of Potsdam, Institute for Environmental Sciences and Geography, Potsdam, Germany
[3] TU Braunschweig, Institute of Geoecology, Division of Soil Science and Soil Physics, Braunschweig, Germany
[4] German Weather Service (DWD), Agrometeorological Research Centre, Braunschweig, Germany

*Correspondence to*: C. Lehr (christian.lehr.1@uni-potsdam.de)



## Abstract

Principal Component Analysis (PCA) of synchronous time series of one variable, e.g. water level or discharge, measured at multiple locations, has been applied in a wide spectrum of hydrological analyses. Principal Components (PCs) were used in

regionalisation and to identify dominant modes, signals, processes or other hydrological properties of the analysed system. The possibility that the PCs of such analysis can exhibit domain dependence (DD) found only little recognition in the hydrological PCA literature so far. DD describes the situation in which the spatial PC patterns are mainly determined by the size and shape of the analysed spatial domain. Domain size means the spatial extent of the analysed data set, domain shape the spatial arrangement of the data sets´ locations. Thus, instead of the hydrological functioning of the analysed system, the

spatial PC patterns rather reflect the functioning of the PCA within the context of the data set´s spatial domain. The effect is caused by homogeneous spatial autocorrelation in the analysed series, a common feature in hydrological data sets. DD patterns are distinct, with strong gradients and contrasts, and can come together with substantial accumulation of variance in the leading PCs. In addition, DD can cause effectively degenerate multiplets, i.e. PCs which are not well separable. All these features are highly suggestive and easily lead to wrong hydrological interpretations. Consequently, DD should be considered

for any application in which the PCs are used to draw conclusions about spatially distinct properties of the analysed system. DD patterns calculated for the analysed spatial domain can be used as reference to test whether spatial PC patterns differ significantly from pure DD patterns. We present two methods, one stochastic, one analytic, to calculate DD reference patterns for defined spatial correlation properties and arbitrary spatial domains. With a series of synthetic examples, we explore the DD effect with respect to a) domain shape, b) domain size and spatial correlation length and c) effectively

degenerate multiplets. Particular focus is given to the effect of DD on the explained variance of the PCs and the contrasts of their spatial patterns. Finally, considering DD is discussed. Accompanying this technical note, R-scripts to (i) demonstrate and explore the DD effect, and (ii) perform the presented DD reference methods are provided.



## 1    Introduction

In hydrology, Principal Component Analysis (PCA), also known as Empirical Orthogonal Function (EOF) analysis or Karhunen–Loève Transform, is a popular tool to analyse spatio-temporal data sets. The analysed data can be structured in various ways (Richman, 1986; Demšar et al., 2013). Here, the focus is on PCA of data sets comprising synchronous time series of one observed variable, e.g. water level, with the time series (a) being distributed in space at multiple locations and (b) being used as variables for the PCA. This is known as S-mode PCA (Richman, 1986) or atmospheric science PCA (Demšar et al., 2013). In this setting, the covariance among the time series from the different locations is analysed (Richman, 1986; Isaak et al., 2018). For each PC there is a temporal and a spatial pattern. The PCs are series of the same length as the analysed time series and can be plotted against the common time index (temporal PC patterns). The eigenvector of each PC is associated with the complete set of locations and can be plotted against the locations´ coordinates (spatial PC patterns). With this, the leading PCs provide a compact description of the spatio-temporal variability of the data set.

A non-exhaustive list of hydrological applications comprises S-mode PCA to describe the spatio-temporal variability of streamflow (Smirnov, 1973; Bartlein, 1982; Lins, 1985ab, 1997; Kalayci and Kahya, 2006), groundwater level (Winter et al., 2000; Longuevergne et al., 2007; Lehr and Lischeid, 2020), lake water level (Lischeid et al., 2010), soil moisture (Korres et al., 2010; Nied et al., 2013; Hohenbrink et al., 2016), precipitation (Kumar and Duffy, 2009; Thomas et al., 2012), drought (Karl and Koscielny, 1982; Santos et al., 2010; Ionita et al., 2015), or river water temperature (Isaak, et al., 2018). In stark contrast to its widespread use, the possibility that the PCs of such analysis can exhibit domain dependence (DD) is rather unknown in hydrological PCA literature.

DD describes the situation in which the spatial PC patterns from S-mode PCA are mainly determined by the size and shape of the analysed spatial domain, meaning the spatial extent of the data set and the spatial arrangement of its locations (Buell, 1975, 1979; Richman, 1986). If the spatial autocorrelation of the data set´s variable is homogeneous across the domain, its size and shape induce distinct sequences of spatial PC patterns due to the variance maximization of the PCs and the orthogonality constraint of the PCs` eigenvectors (Jolliffe, 2002; Wilks, 2006). Buell (1975) identified classical sequences for data sets with basic geometric domain shapes and isotropic spatial autocorrelation (e.g. Figure 1). The spatial pattern of PC 1 is a weighted spatial average emphasizing the centroid of the network ("mean behaviour"). The PC 2 pattern is a gradient depicting the variability along the axis of the longest extent of the domain. The PC 3 pattern covers the next largest spread of spatial variability orthogonal to the spatial patterns of PC 1 and PC 2, etc. Given the functioning of the PCA, the sequence simply reflects (a) that the covariance between the locations has its maximum in the centroid of the network because it is the point which is on average closest to all other locations, and (b) that the only structure in the variability of the data set is the homogeneous decay of covariance with distance (Dommenget, 2007). On a sphere the resulting spatial PC patterns of such a data set would be the spherical harmonics (North and Cahalan, 1981).

Ignorance about DD can easily lead to wrong interpretations of PCA results. DD patterns are distinct, with strong gradients and contrasts, and therefore highly suggestive to indicate physically meaningful drivers or properties of the analysed system.



In the climatological literature DD was intensely discussed (Buell, 1975, 1979; Horel, 1981; Richman, 1986, 1987, 1993; Jolliffe, 1987; Legates, 1991, 1993). Apparently, the topic did not reach the hydrological community, even though the effect

of size and shape of the network geometry on the results was observed in early hydrological S-mode PCA applications (Smirnov, 1973; Bartlein, 1982; Lins, 1985b). For that reason, we want to raise attention to the DD effect among PCA users in the hydrological community again to reduce the risk of drawing wrong hydrological conclusions from spatio-temporal PCA.

DD is of importance for any application in which a PCA of observed data is used to draw conclusions about spatially distinct

properties of the analysed system. This concerns descriptive applications in which the spatial PC patterns are used to identify dominant hydrological modes (Smirnov, 1973; Bartlein, 1982; Lins, 1985ab, 1997; Kalayci and Kahya, 2006; Thomas et al., 2012; Ionita et al., 2015) or regions with similar hydrological behaviour (regionalisation) (Karl and Koscielny, 1982; Santos et al., 2010; Nied et al., 2013), as well as the interpretation that they represent the spatial variability of concrete hydrological signals (Longuevergne et al., 2007; Lewandowski et al., 2009), hydrological processes (Hohenbrink et al., 2016; Isaak et al.,

2018; Scholz et al., 2024) or physical properties (Korres et al., 2010; Lischeid et al., 2010). For all those applications it is essential that there is a physical counterpart for the spatial PC patterns in the analysed system. Thus, DD touches the very basic question whether the applied combination of data set and data analysis method allows inference on the analysed system.

DD is critical in particular for any interpretation of the PCs based on correlation analysis with other variables (Korres et al.,

2010; Lischeid et al., 2010; Hohenbrink et al., 2016; Isaak et al., 2018; Scholz et al., 2024). In case of "strong DD" the correlation between their spatial patterns depends mainly on the selected spatial domain. Consider for example a soil texture gradient in west-east direction and the classical Buell patterns in Figure 1. Depending on the selected domain the spatial patterns from different PCs would correlate strongly, moderately or not at all with the gradient. Consequently, those correlations would be neither useable for the interpretation of the PCs nor for the identification of predictors for their spatio-

temporal patterns. Thus, spatial PC patterns should be checked for DD prior to any interpretation implying causal relationships.

When checking for DD, it has to be considered that DD patterns are original for every combination of spatial domain and spatial correlation properties of the analysed data set. Thus, the "classical Buell patterns" are DD patterns for the distinct combinations of size and shape of the domain, spatial covariance function and spatial correlation length used in Buell´s

(1975) numerical experiments (e.g. Figure 1). Spatial PC patterns of real-world data sets can be expected to deviate from those archetypes due to possible differences in all these aspects. In addition, there might be a blurring effect of measurement errors. For spatially regular distributed data sets with strong homogeneous autocorrelation and domain boundaries similar to one of Buell´s basic domains the DD patterns of the leading PCs are commonly visually easy to recognize as Buell-like. This gets less clear for those of the low ranked PCs. They are more finely detailed and less robust against deviations from Buell´s

settings. DD patterns from data sets with more complex domain shapes and / or spatially irregular distributed locations, which is the common case in hydrology, can differ substantially from Buell´s archetypes. All in all, visual recognition by





comparison with Buell patterns is rather limited. Comparison with DD patterns calculated for the analysed spatial domain overcome these limitations (Cahalan et al., 1996; Dommenget, 2007). They can be used as reference to test whether spatial PC patterns differ significantly from what has to be expected from DD alone.

The objective of this technical note is to introduce (i) the domain dependence effect and (ii) the application of domain dependence reference patterns to the hydrological community. We illustrate our introduction with synthetic examples only. This ensures that the statistical properties of the examples, in particular their spatial correlation properties and spatial domains, are strictly defined. It further clarifies that all observed effects are solely caused by the specified statistical properties. Another advantage is that series of examples with systematic differences can be constructed to study the effects

of specific properties, e.g. spatial correlation length or spatial extent, on the PCA results.

Note that we aim for an illustrative introduction for PCA practitioners. For a mathematically rigid introduction to the DD phenomenon see Buell (1975, 1979) and North and Cahalan (1981). Scripts to explore the DD effect and calculate DD reference patterns for defined spatial correlation properties and arbitrary spatial domains are provided (Lehr, 2024). After presenting the two implemented DD reference methods and the scripts, a series of synthetic examples is used to explore the

DD effect with respect to a) domain shape, b) domain size and spatial correlation length and c) effectively degenerate multiplets, i.e. PCs which are not well separable. Particular focus is given to the effect of DD on the explained variance of the PCs and the contrasts of their spatial patterns, both common indicators for the interpretation of PCA results. Finally, considering DD is discussed.

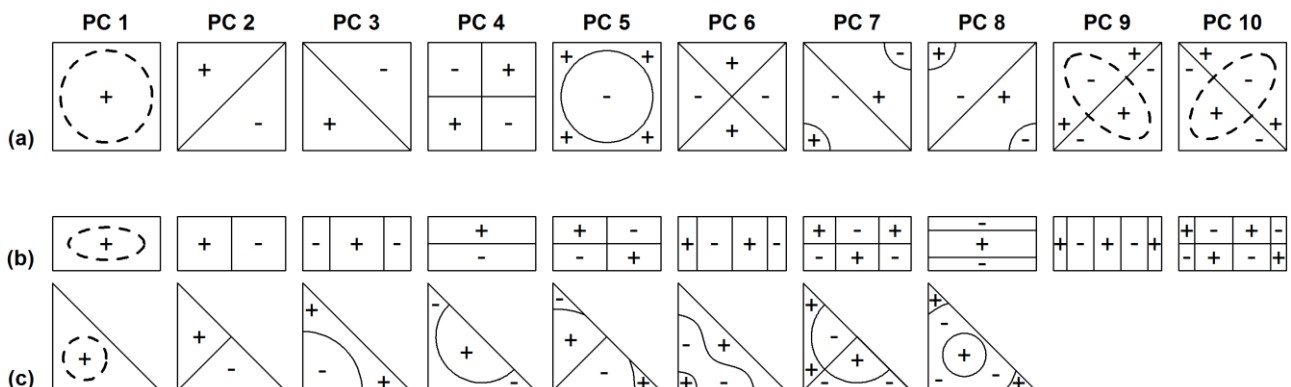


**Figure 1 Exemplary reproduction of some "classical Buell patterns" for differently shaped domains of relatively similar size: (a) 6 × 6 square, (b) 5 × 10 rectangle and (c) 8 × 8 triangle (Figure 2, 5 and 4 adapted from Buell, 1975). The signs indicate positive and negative values of the spatial PC patterns. The patterns are for data exhibiting exponentially decaying spatially isotropic autocorrelation with spatial correlation length of 2 grid cells (function F1, scale parameter L = 2 in Buell (1975)). The spatial PC**
**patterns of the rectangular shape are for the gaussian covariance function (function F2 in Buell (1975)) but Buell noted that the patterns of the exponential function are essentially the same. The dashed circle of PC 1 indicates that its pattern is of one sign in the entire domain with absolute values being highest within the circle and fading out towards the domain boundaries.**





## 2    Methods

### 2.1. Principal Component Analysis

PCA maps a m × n data matrix to n new linearly uncorrelated variables, the Principal Components (PCs), such that the PCs successively maximise represented fractions of the data set's variance (Wilks, 2006). The data set´s variance is defined as the sum of variances of the variables. It equals the sum of the diagonal elements (trace) of its covariance matrix. PCA can be performed as eigenvalue decomposition of the variables' covariance matrix or as singular value decomposition of the variables' matrix with the variables being centred to their mean (Jolliffe, 2002). Unfortunately, the terminology is not used consistently throughout the literature. Here, we follow the terminology used by Jolliffe (2002) and Jolliffe and Cadima (2016) for the eigenvalue approach.

Each PC is associated with an eigenvalue, scores and an eigenvector. The scores are the values of a PC. The variance of the scores of a PC equals its eigenvalue. The ratio of a PC eigenvalue to the sum of all PC eigenvalues gives the fraction of the data set's variance assigned to that PC. The PCs, i.e. the PC scores, are calculated as linear combination of all analysed variables (non-locality). The coefficients in this linear combination are termed loadings. The loadings of a PC are defined as the elements of the eigenvector associated with that PC. The eigenvectors of all PCs define the orthogonal basis of the new ordination system the analysed data is mapped to (orthogonality constraint). Subject to the eigenvectors being orthogonal and the PCs being uncorrelated, the linear combinations of the PCs provide the optimal linear functions to successively maximise variance accounted for (variance maximization). The maximum variance that can be described by a linear combination of the analysed variables is assigned to the first PC, the maximum of the remaining variance to the second PC, and so forth. Thus, the leading PCs provide a compact description of the data set´s variance. It is quite common that a few PCs suffice to summarize a major part of a data set´s variance.

### 2.1.1.    S-mode PCA

In S-mode PCA, the analysed variables are synchronous time series distributed in space at multiple locations (Richman, 1986). Thus, the PC scores are series of the same length as the analysed time series (temporal PC patterns) and the loadings yield values for each location (spatial PC patterns), describing the weighting of the analysed time series to calculate the PC scores. All PC series are linearly uncorrelated with each other, each PC series is associated with a spatial pattern and all spatial PC patterns are orthogonal to each other. Note that here, we perform S-mode PCA only.





### 2.1.2. Correlation matrix based PCA, correlation loadings and contrasts of spatial PC patterns

Normalizing the variables to zero mean and standard deviation one (z-scaling) prior applying PCA ensures equal weighting of the analysed variables. This is of particular importance if the range of values between the analysed variables differs

substantially. A PCA with z-scaled variables is identical to an eigenvalue decomposition of the correlation matrix of the analysed variables. In hydrology, correlation matrix based PCA is to our knowledge more common than covariance matrix based PCA.

In correlation matrix based PCA, normalizing the loadings of a PC with the square root of its eigenvalue is equivalent to the Pearson correlation between the scores of that PC and the analysed variables. Thus, the loadings are normalized to the

Pearson correlation range from -1 to 1. To prevent confusion, we use the term "correlation loadings" for these normalized loadings. Using correlation loadings can simplify reading and interpretation of the PCA results. For S-mode PCA, the normalization enables direct comparison of the contrasts of spatial patterns from different PCs or PCAs. Here, we define the contrast of a spatial PC pattern as the range between the minimum and maximum of the correlation loading values of that PC. Thus, the maximum contrast possible would be 2. Note that this is different from the "classical loadings", used in the

linear combination to calculate the PC scores, which are not normalized to a common range. In the following, the spatial PC patterns are described with correlation loadings only.

### 2.2. DD reference patterns

DD reference patterns are the DD patterns of a distinct combination of spatial domain and spatial correlation properties.

They can be used as null hypothesis to test whether spatial PC patterns differ significantly from what has to be expected from DD alone. Here, we tested the association between two patterns with simple Pearson correlation. The statistical significance of the correlations was assessed with t-tests and the significance level 0.05.

### 2.2.1. Stochastic method

In the stochastic method, PCA is applied on simulated data sets to derive DD reference patterns. The data sets consist of synchronous spatially distributed time series exhibiting spatial but no temporal autocorrelation. Each data set is produced by concatenating realizations of random fields with identical spatial correlation properties (Figure 2). The spatial autocorrelation is defined with a spatial covariance model. Each realization of the field represents one instant of time of a data set. Thus, at each location the respective time series consists of a sequence of random numbers. The number of field

realizations gives the length of the simulated time series. As the data sets consist of spatially correlated white noise time series, their temporal PC patterns are white noise as well. The spatial PC patterns of the data sets are solely determined by





the spatial domain and the spatial correlation properties defined in the simulation. The spatial PC patterns of data sets simulated with identically parameterized random fields differ due to the randomness in the simulations. Therefore, a three-step procedure is applied to get stable patterns.

Step 1: An ensemble of data sets with identical spatial domain and spatial correlation properties is simulated. Each of the data sets is analysed separately with a PCA, resulting in a PCA ensemble.

Step 2: The stability of the spatial PC patterns is assessed by pairwise correlating the spatial patterns of all possible combinations of PCs with identical ranks from the PCA ensemble. For each PC rank, the mean $R^2$ of the correlations is used to describe the overall congruence of the respective spatial PC patterns.

Step 3: For each PC rank (a) the mean spatial patterns from all PCAs of the ensemble and (b) their standard deviation patterns are calculated. They are calculated as the mean and standard deviation of the correlation loadings of PCs with identical rank from the PCA ensemble.

The mean spatial PC patterns are the DD reference patterns for data sets with the spatial domain and the spatial correlation properties defined in step 1. The standard deviation patterns serve as their spatially discrete uncertainty estimation. The variance represented with the DD reference patterns ("explained variance") is estimated with the mean and standard deviation of the explained variances of PCs with identical rank from the ensemble.

PCs with identical rank from different data sets of an ensemble might exhibit basically the same spatial pattern but with opposite signs due to the randomness of the field simulations, i.e. the pattern of one data set might be basically a negative version of another one. For the calculation of mean and standard deviation of the spatial PC patterns of an ensemble (step 3), the spatial patterns of PCs with identical rank are therefore harmonized such that they all are correlating positively. Thus, the correlation loadings of PCs that are correlating negatively with those of identically ranked PCs from the first data set are multiplied by -1 and therefore reversed.

Note that the suggested method requires the use of correlation loadings to describe the spatial PC patterns. Thus, it is restricted to correlation matrix-based S-mode PCA, meaning the analysed series have to be z-scaled (Sections 2.1.1 and 2.1.2). Furthermore, the mean spatial PC patterns are derived from a data set ensemble, not from a distinct single data set. Thus, they cannot be scaled to classical loadings and they cannot be used to calculate PC scores.

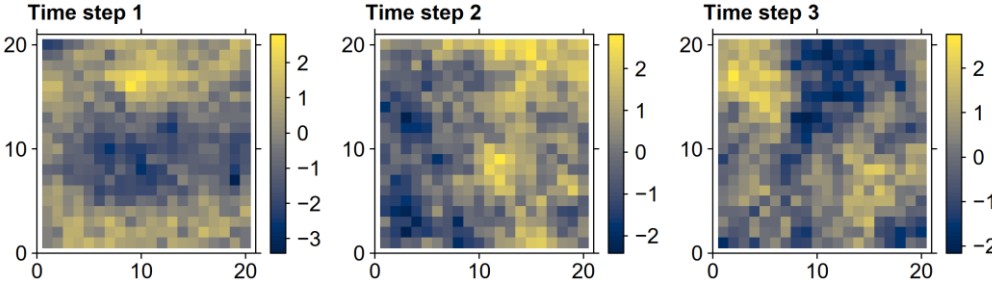

**Figure 2 Three realizations of a 20 × 20 random field simulated with an isotropic exponential covariance model and spatial correlation length of 10 cells representing three instants of time of a simulated data set.**





### 2.2.2. Analytic method

Another possibility to produce DD reference patterns is to perform a PCA with the "analytic", or "exact", covariance matrix (North et al., 1982; Cahalan et al., 1996; Dommenget, 2007) of a spatially homogeneous covariance function. The analytic

covariance matrix consists of the covariances between all of the data set´s locations calculated directly with their interpoint distances from the function.

Confidence limits to identify clearly separated eigenvalues and eigenvectors can be estimated e.g. with North´s rule of thumb (North et al., 1982; Hannachi et al., 2007) based on the data set´s effective sample size $n^*$, also known as number of independent observations in the sample or the number of degrees of freedom (Hannachi et al., 2007). The 95 % confidence

interval of the eigenvalue $\lambda_k$ is given by $\delta\lambda_k \sim \lambda_k\sqrt{2/n^*}$. In our case here, $n^*$ equals the length of the analysed time series because the series do not exhibit temporal autocorrelation. The confidence interval for the associated eigenvector $u_k$ can then be estimated with $\delta u_k \sim (\delta\lambda_k/\Delta\lambda)u_j$ where $u_j$ is the eigenvector of $\lambda_j$, the closest eigenvalue to $\lambda_k$, and $\Delta\lambda$ the spacing $(\lambda_j - \lambda_k)$ between both eigenvalues. For consistency with the stochastic method (Section 2.2.1), the eigenvectors (spatial patterns) were scaled to correlation loadings (Section 2.1.2).

A brief review of different variants using the analytic covariance matrix to produce PCA reference patterns is given in Appendix A.

## 3 Scripts

The selection of scripts accompanying this technical note (Lehr, 2024) contains: (1) A Demo in which the DD of PCs is

demonstrated by visual examination of the spatial PC patterns from single simulated data sets, (2) an implementation of the stochastic DD reference method (Section 2.2.1), and (3) an implementation of the analytic method (Section 2.2.2) based on Dommenget (2007) and the associated Matlab scripts. The user can define domains with distinct sizes and shapes, and the spatial correlation properties. The scripts and their documentation can directly be used for educational purposes. We recommend going first step by step through the Demo to get into the functioning and logic of the scripts. For the Demo and

the stochastic reference script, it is best to start with the pdf documentation which includes a formatted version of the script, extra annotations and sample results. All scripts are written in R (R Core Team, 2019). The simulations of the data sets are performed with the "RandomFields" package (Schlather et al. 2015, 2020). PCA is performed with the function "prcomp" and the significance test of the correlation analysis with the function "cor.test" from the default "stats" package (R Core Team, 2019).






## 4 Exploring the DD effect

### 4.1. First examples

As a start, we estimated DD reference patterns for Buell´s (1975) three basic geometric domain shapes (Figure 1) using the stochastic method (Section 2.2.1). Ensembles of 100 data sets were simulated for each of the three shapes. The domain

boundaries are shown in Figure 3. All cells within the boundaries were used. (Note that the shape of a domain means the spatial arrangement of the data set´s locations. It should not be confused with the shape of its boundary.) The sides of the square, the long side of the rectangle and the legs of the perpendicular triangle were 20 cells long, the short side of the rectangle was 10 cells long. Thus, the rectangular and the triangular domain were of half the size of the square. Each data set was simulated with a spatially isotropic exponential covariance model and a spatial correlation length of 10 cells.

For the reliability of the stochastic DD reference patterns, their stability is essential. Figure 4 summarizes the results of the stability analyses (step 2 of the stochastic method) from a series of ensembles with identical spatial domain and spatial correlation properties but different time series lengths. Thus, the plot shows for each PC rank the dependency of its spatial patterns' stability from the time series length if all other parameters used in the simulation are identical. Based on that information it can be decided whether additional ensembles with longer time series shall be simulated to improve the

estimation. Here, we considered a time series length of 10 000 sufficient for all three domains. Figure 5 shows the mean spatial PC patterns of these ensembles. Those are the stochastic DD reference patterns. Most of them correspond to the Buell patterns shown in Figure 1. Some exhibit switches in the ranking, e.g. PC 3+4 of the rectangular domain or PC 7+8 of the square domain. The uncertainty estimation of the stochastic DD reference patterns, given by the standard deviation of the spatial PC patterns from the data set ensembles, is shown in Figure 6. Exemplarily, the mean and standard deviation patterns

of the square domain are shown in more detail (Figure 7). The scales provide information on the magnitude of both patterns. To make use of the standard deviation patterns (Figure 7b) as uncertainty estimation of the DD reference patterns, it is necessary to consider their magnitudes in relation to the contrast from the mean spatial patterns (Figure 7a). In addition, the fractions of variance assigned to the DD reference are given.

The stability of the DD patterns reflects their distinctness in the sequence of spatial PC patterns according to the PCA

constraints. It depends on the specific combination of domain size and shape and spatial correlation properties of the data set. For example, for the properties here, PCs 8 to 10 of the triangle are more stable than the ones of the rectangle (Figure 4c+b). Generally, there is the tendency that the spatial patterns of low ranked PCs, which contain also more fine details, require longer times series to gain stability. It seems counter intuitive at first that PC 2 of the rectangle stabilizes faster than its PC 1 (Figure 4b). It indicates that for the properties of the simulated data the rectangular domain shape gives a clearer orientation

for the spatial pattern of PC 2 than for the one of PC 1. Thus, especially for short time series the orientation of the gradient along the long side of the rectangle (PC 2) is more distinct than the position of the monopole in the centroid of the rectangle





(PC 1) (Figure 5b). Similarly for the triangle, the orientation of the gradient patterns of PC 2 and 3 induced by its long side are more distinct than the position of its PC 1 monopole (Figure 4c and Figure 5c).

PCs with ambiguous orientation of spatial patterns are more likely to occur for symmetric domain shapes than for
asymmetric ones (North et al., 1982). The basic geometric domain shapes used here exhibit rotational symmetry of order 4 (square), order 2 (rectangle) and order 1, i.e. no rotational symmetry, (triangle). Accordingly, within the range of the analysed time series lengths the number of PCs that exhibited unstable spatial patterns differed between the domain shapes (square: 8, rectangle: 5, triangle: 2 in Figure 4). Unstable spatial PC patterns are indicative for effectively degenerated multiplets and will be discussed in Section 4.4.

The stochastic reference script enables to produce catalogues of stability plots and DD patterns like in Figure 4 and Figure 7 for data sets with different spatial domains and spatial correlation properties. Both plots in combination can be used to explore how the properties of a data set affect the DD patterns. Sample catalogues are provided with the scripts (Lehr, 2024).

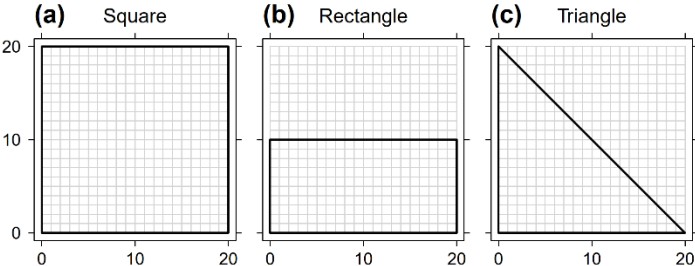

**Figure 3 (a) Square, (b) rectangular and (c) triangular domain boundaries on the 20 × 20 grid. The grid cells represent locations from a data set.**





**Figure 4 Stability of the spatial patterns from the leading ten PCs in relation to the time series length of the simulated data within the (a) square, (b) rectangular and (c) triangular domain boundaries of Figure 3. All cells within the boundaries were used. For each domain the results from 12 data set ensembles are shown. Each ensemble consists of 100 data sets simulated with identical time series length, an isotropic exponential covariance model and a spatial correlation length of 10 cells. Each simulated data set was analysed separately with PCA. Symbols depict the mean $R^2$ of the correlation between the spatial patterns of all PCs with identical rank derived from the respective ensemble. The legends in (c) apply also to (a) and (b) of the respective row.**





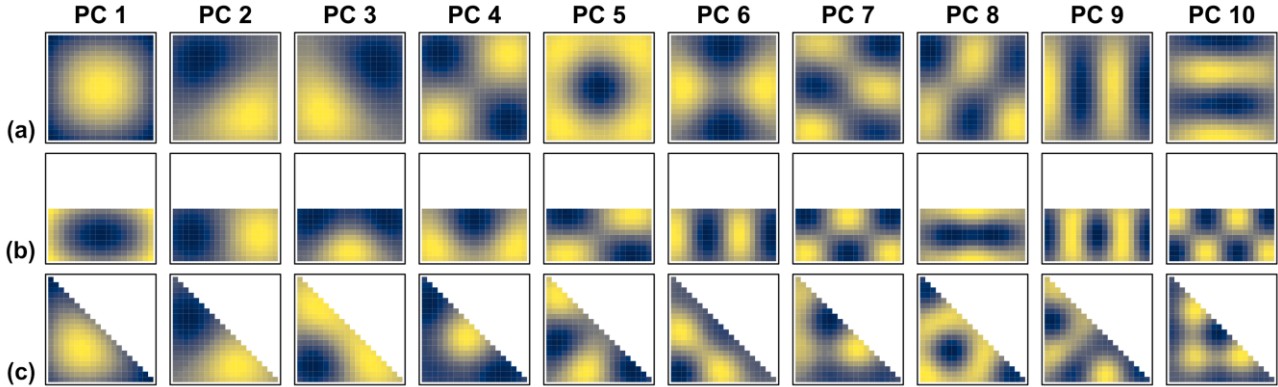

**Figure 5 Overview of the leading ten mean spatial PC patterns (DD reference patterns), estimated with the stochastic method from the data set ensembles with time series length 10 000 shown in Figure 4. Instead of the +/- schemes used by Buell (1975) (Figure 1) we use colour gradients to picture the spatial patterns.**


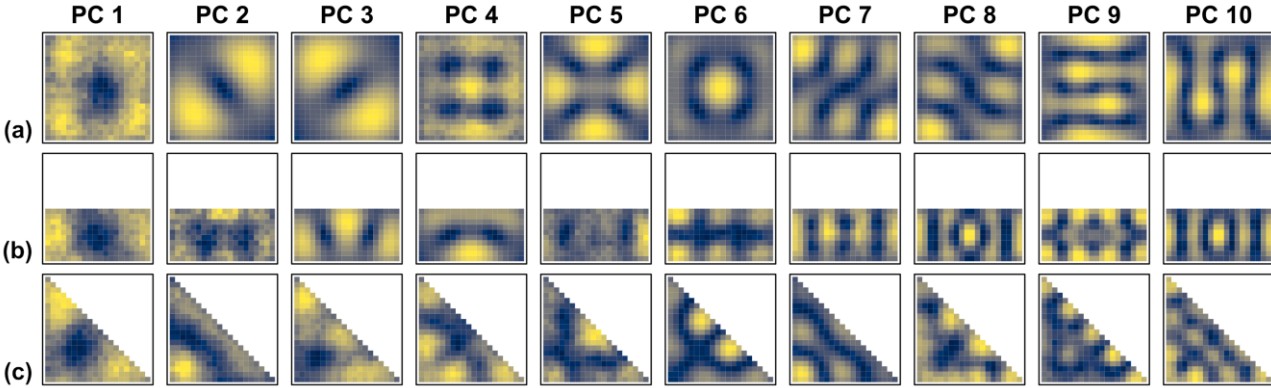

**Figure 6 As in Figure 5 but for the standard deviation patterns (uncertainty estimation of the stochastic DD reference patterns). From blue to yellow the colour gradients depict increasing uncertainty.**





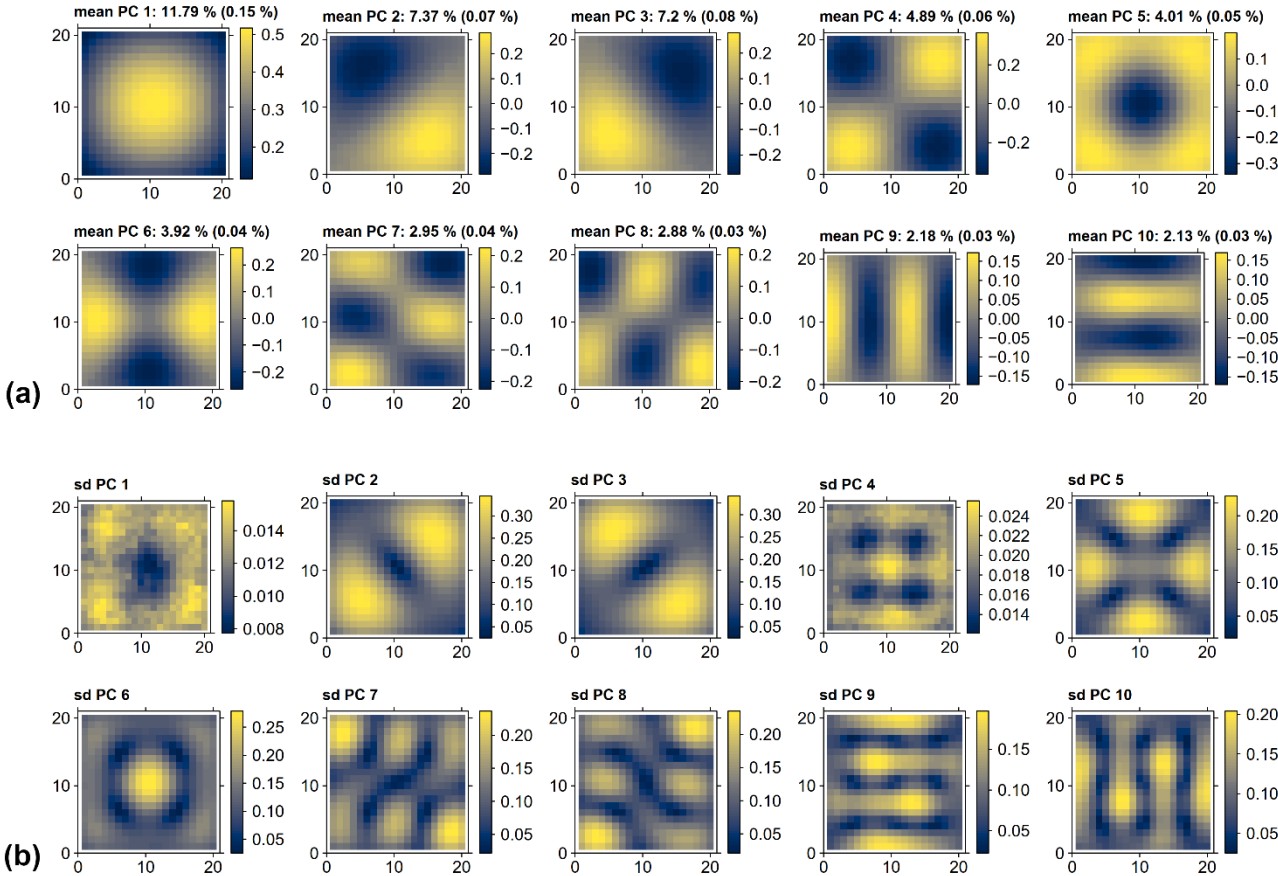

**Figure 7 Detail plot of mean (a) and standard deviation (b) of the spatial patterns from the leading ten PCs of the data set ensemble with the square domain shown in Figure 5a and Figure 6a. The panel titles of (a) contain the mean (and standard deviation) of the fractions of variance assigned to the respective PCs from the 100 PCAs of the ensemble.**

### 4.2. Domain shape

For data sets with identical spatial correlation properties and similar domain size, the DD patterns are original for every domain shape. This is obvious for domains of such simple and clearly different shape like the three geometric shapes used so far. The sequence of their DD patterns is visually easy to recognize and kind of intuitive. For more complex shapes, the DD patterns are less foreseeable and visual recognition is rather limited.

For demonstration, we compared the DD patterns from data sets with identical spatial correlation properties in which all cells within the three geometric boundaries of Figure 3 were selected (Figure 5) with two variants in which only 40 % of the cells were randomly selected. In the first variant the subsampling was spatially homogeneous (Figure 8), in the second spatially heterogenous (Figure 9). The domain of the second variant contained a subregion with higher sampling probability than the





rest of the domain, i.e. within each domain there is one area in which the locations cluster. Clusters of locations have more

weight in the calculation of the PCs analogue to the calculation of a weighted spatial mean (Karl et al., 1982). For the DD

pattern of PC 1 the effect is obvious. Its monopole is placed in the centroid of the network. In comparison with the regular

variant (Figure 1, Figure 5 and Figure 8) it is therefore shifted according to the density of the locations (Figure 9). The

patterns of all other PCs are not so easy to foresee.

Visually, the domains of the subsampling variants are still clearly of square, rectangular and triangular shape. Their leading

DD patterns are recognizable as distinct spatial patterns. Most of those from the homogeneous subsampling variant (Figure

8) appear like noisy counterparts of the all cells patterns (Figure 5). In the heterogeneous case (Figure 9), the patterns of the

square domain appear again relatively similar, while for the triangular and rectangular domain only a few PCs exhibit

visually similar patterns, e.g. PC 2.

But what about the similarity of patterns formed by congruent selections of cells from the different variants? In other words,

does calculating spatial PC patterns from two different domains result in different relations between the values at locations

with coincident coordinates? To check that, we correlated the patterns of the subsampling variants with the patterns formed

by the corresponding subsets from their all cells counterpart (that is, the all cell patterns clipped with the coordinates of the

subsampling variant). For example, the patterns from the homogeneously subsampled square (Figure 8a) were correlated

with the patterns from the all cells square (Figure 5a) clipped with the coordinates of the subsampled square.

For the spatial patterns of the homogeneous subsampling variant and the all cells variant, the correlation analysis confirmed

the visual impression of overall similarity (Table 1). But it also showed that there are differences and that the differences

generally increase towards the lower ranked PCs. The comparison with the heterogeneous variant yielded substantially

stronger deviations (Table 2). Thus, generally, visual recognition of Buell like patterns in S-mode PCA results is a concrete

indication for DD. However, it is so in particular for the leading PC patterns from domains with rather homogeneous spatial

arrangement of locations within boundaries similar to Buell´s archetypes. Even for domains of similar size and identical

spatial correlation properties, inhomogeneous distribution of locations alone can result in DD patterns substantially deviating

from what one might expect with the classical Buell patterns in mind.

Side note: The spatial PC patterns of the subsampling variants required shorter time series lengths to stabilize (Figure 10 and

Figure S1) than the all cells variant (Figure 4). This indicates that the subsampling resulted in a more unbalanced

arrangement of locations and therefore a more distinct orientation for the order of the orthogonal spatial PC patterns.





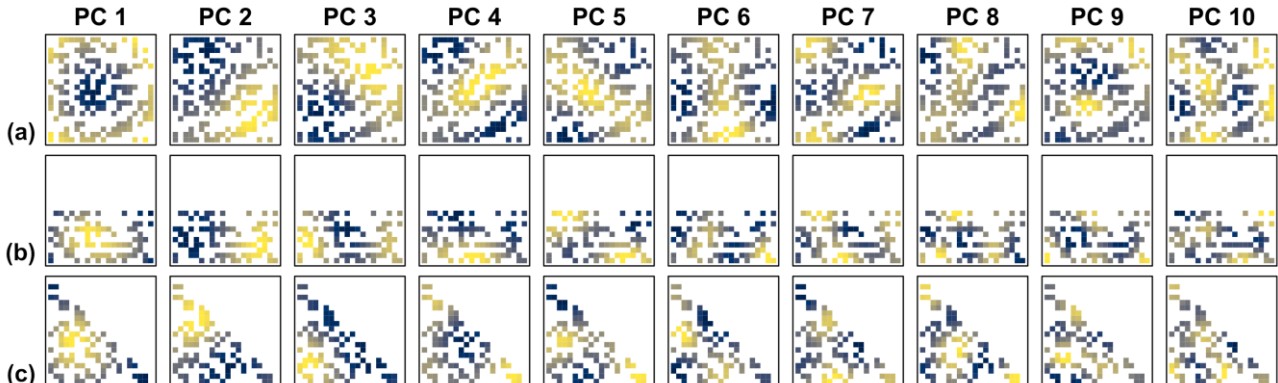

**Figure 8 DD reference patterns as in Figure 5 but for a random selection of only 40 % from the cells within the three geometric domain boundaries of Figure 3. The sampling probability was homogeneous across the domain (spatially homogeneous case).**


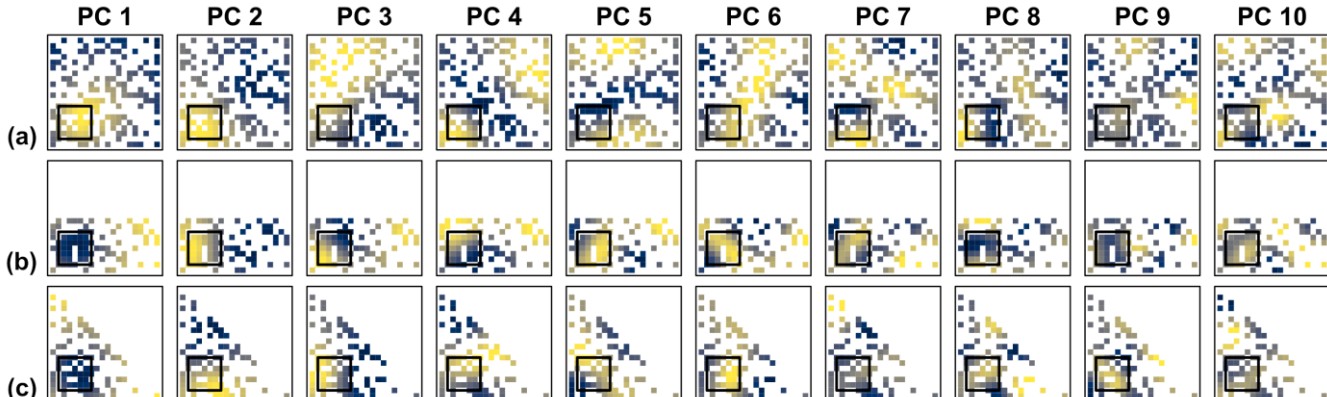

**Figure 9 DD reference patterns as in Figure 5 but for a random selection of only 40 % from the cells within the three geometric domain boundaries of Figure 3. The sampling probability within the small square in the lower left was three times higher than in the rest of the domain (spatially heterogeneous case).**






**Figure 10 Stability of the spatial PC patterns as in Figure 4 but for the patterns of Figure 8.**

|  | PC 1 | PC 2 | PC 3 | PC 4 | PC 5 | PC 6 | PC 7 | PC 8 | PC 9 | PC 10 |
|---|---|---|---|---|---|---|---|---|---|---|
| Square | 0.94 | 0.97 | 0.93 | 0.52 \5 | 0.45 \4 | 0.79 | 0.73 | 0.55 | 0.59 \10 | 0.38 \9 |
| Rectangle | 0.87 | 0.97 | 0.80 \4 | 0.78 \3 | 0.83 | 0.52 \7 | 0.32 | 0.43 \9 | 0.40 \8 | 0.23 |
| Triangle | 0.89 | 0.95 | 0.92 | 0.87 | 0.93 | 0.80 | 0.84 | 0.84 | 0.42 \10 | 0.28 \9 |

**Table 1 $R^2$s of the strongest correlations between the DD patterns of the square, rectangular and triangular shaped domains from**
**the homogeneous subsampling variant (Figure 8) and the patterns formed by the corresponding subsets from their all cells**
**counterpart (that is, the all cell patterns (Figure 5 (a) to (c)) clipped with the coordinates of the subsampling variant (Figure 8 (a)**
**to (c))). Mostly the best correlating patterns were of identical PC rank. If the best correlation was with a clipped all cell pattern of**
**different rank, that rank is given after the "\". For example, for the rectangular shape the PC 3 pattern of the subsampling variant**
**correlated best with PC 4 of the clipped all cell patterns. All correlations were significant ($p < 0.05$).**





|  | PC 1 | PC 2 | PC 3 | PC 4 | PC 5 | PC 6 | PC 7 | PC 8 | PC 9 | PC 10 |
|---|---|---|---|---|---|---|---|---|---|---|
| Square | 0.56 \3 | 0.85 \3 | 0.97 \2 | 0.43 | 0.61 \6 | 0.33 | 0.44 | 0.43 \9 | 0.25 | 0.30 |
| Rectangle | 0.85 \2 | 0.76 | 0.52 \4 | 0.85 \3 | 0.48 \6 | 0.37 \7 | 0.49 \10 | 0.39 | 0.56 | 0.19 \7 |
| Triangle | 0.73 | 0.64 | 0.43 | 0.49 | 0.86 \6 | 0.30 \8 | 0.42 | 0.39 \10 | 0.15 \10 | 0.49 \9 |

**Table 2 As in Table 1 but for the heterogeneous subsampling variant (Figure 9).**

### 4.3. Domain size and spatial correlation length

The ratio between domain size and the spatial correlation length affects the fractions of variance allocated to the PCs (Figure 11) as well as the contrasts of the spatial PC patterns (Figure 12). If there is no spatial correlation (spatial "white noise"), the spatial patterns of all PCs are white noise. All PCs represent the same fraction of variance, one divided by the total number of PCs. The magnitudes of the contrasts of their spatial patterns are small and on the same level. For spatial correlation length increasing from zero towards infinity, the data sets´ series from all locations get more and more similar, converging

towards identity of all series (perfect correlation). If the latter is reached, there is no variance in the data that could be distributed and, consequently, there are no patterns or contrasts in the PC patterns. In between the two extremes, successive allocation of variance to the PCs and spatial PC patterns with distinct contrasts appear.

For the variance allocation, it is simple. Increasing correlation lengths result in increasing accumulation of variance in the leading PCs, converging towards accumulation of the total variance in PC 1.

For the contrasts, it is more complex. The maximum contrasts appear for correlation lengths in the order of magnitude of the domain size. The exact maximum is specific for the different PCs and depends on the particular domain shape. For example, for the triangular domain here (Figure 12c), the contrasts of the PC 1 patterns peak at a correlation length of 13 cells, the ones of the PC 2 patterns at a correlation length of 21 cells (not shown). The increase of the contrasts between zero correlation length and the correlation lengths of the maximum contrasts reflects the increasing fraction of covarying

locations that support the poles of the DD patterns. The decrease of the contrasts between the correlation lengths of the maximum contrasts and infinite correlation length reflects the increasing similarity of all locations which leads to smoother spatial PC patterns with contrasts converging towards zero.

Within a DD sequence, the magnitude of the contrasts differs between the PCs. Generally, they peak at PC 2 (Figure 12) and decay with decreasing PC order (Figure S2). In this sequence it is first the coarse structures with stronger contrasts that are

described and then the more fined detailed structures which tend to be smoother (Figure 5 and Figure 12). The "spatial average" pattern of the PC 1 monopole generally exhibits contrasts on low to intermediate level compared with the "strongest contrast" pattern of the PC 2 dipole.





Substantial accumulation of variance in the leading PCs is commonly interpreted as indication for dominant processes or modes of the analysed system. In particular the combination with distinct PC patterns exhibiting strong contrasts is highly

suggestive. The results demonstrate that both aspects are rather limited indicators and not sufficient for such interpretation. Quite the contrary, if spatially homogeneous autocorrelation is dominant in the data, both have to be expected.

Note also that the effect of the autocorrelation is spread over all PCs. Thus, for process identification etc., it is the question whether the features of interest cause signatures (spatio-temporal heterogeneities) distinct enough to stick out of the homogeneous background (Cahalan et al., 1996). Next question is whether they get clearly assigned to single PCs or whether

they are as well smeared over several, if not all, PCs.

Whether at all a set of PCs is an appropriate model to describe the features of interest from the analysed system is a different question that should be considered carefully for each individual case. For example, for physical processes or modes of geosystems, the S-mode PC properties orthogonality of spatial patterns and linear uncorrelatedness of temporal patterns are heavy constraints (Buell, 1979; Jolliffe, 2002; von Storch and Zwiers, 2003; Hannachi et al., 2007; Monahan et al., 2009).


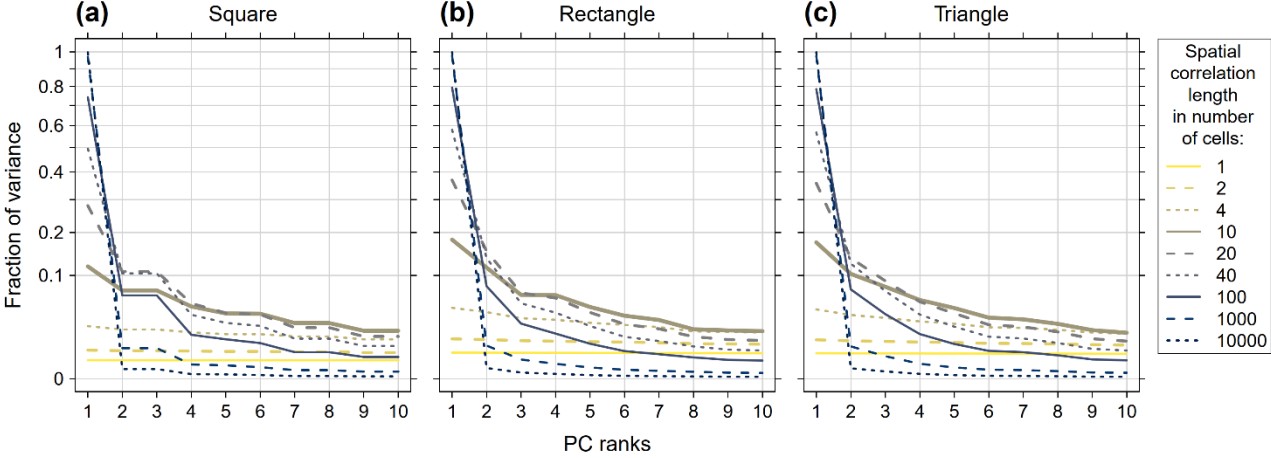

**Figure 11 Variance representation of the ten leading PCs modelled with the analytic DD reference method using an isotropic exponential covariance model, nine different spatial correlation lengths and the domain boundaries (a) square, (b) rectangle and (c) triangle from Figure 3. All cells within the boundaries were used. The Y-scale is square root transformed for better readability.**


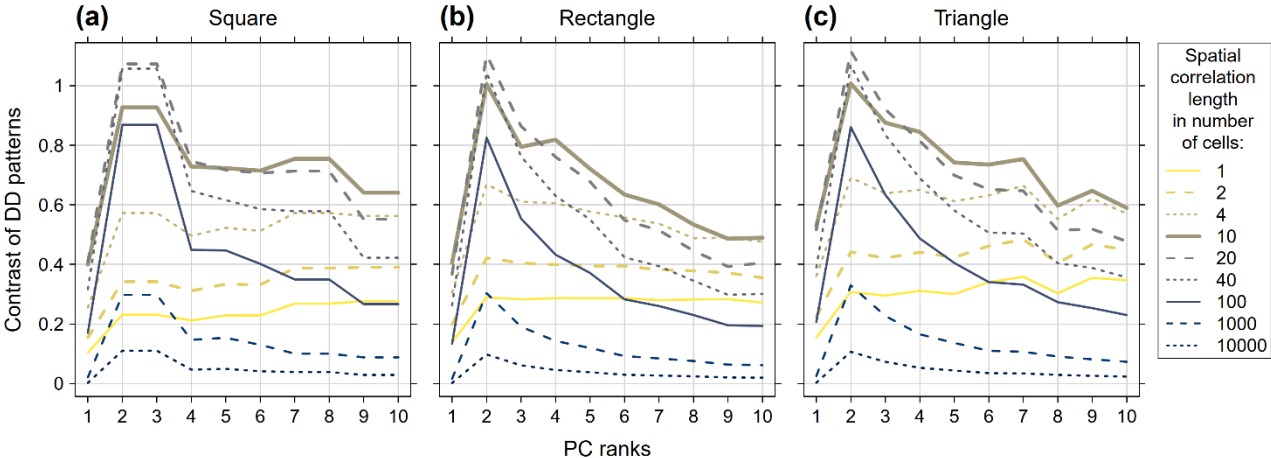

**Figure 12 As in Figure 11 but for the contrasts of the DD patterns.**

#### 4.4. Effectively degenerate multiplets

Effectively degenerate multiplets are PCs with consecutive ranks, often PC pairs, which are not well separated by the PCA (North et al., 1982). They are indicated by noticeably similar eigenvalues (fractions of explained variance) considering their position in the ranking of the PCs, e.g. PC 2+3 in Figure 11a and PC 3+4 in Figure 11b, both for spatial correlation length of 10 cells. Within the subspace spanned by the multiplets' eigenvectors their rotation is arbitrary. All eigenvectors of the multiplet are needed to adequately describe the multiplets' subspace. Consequently, the multiplet should not be split for

summarizing the data set, interpretation, further analysis (North et al., 1982) or rotation (Jolliffe, 1987, 1989). The concept of effectively degenerate multiplets (short: effective multiplets, as in Wilks (2006)) is closely related to degeneracy of eigenvalues. For clarification we provide a brief introduction in Appendix B.

   In S-Mode PCA, spatial and temporal patterns are associated to the PCs. Often the hope is that the leading PCs represent the dominant spatio-temporal features of the data set. In case of effective multiplets, one spatio-temporal feature is described by

the two or more PCs forming the multiplet. This feature can be described with any linear combination of the spatio-temporal patterns of the involved PCs (Appendix B). For example, a degenerated PC pair could indicate "a signal that is propagating in space" (von Storch and Zwiers, 2003; Roundy et al., 2015) like the Madden-Julian Oscillation (Kessler, 2001). Note that such signal might be further modified by lower ranked PCs that are clearly separated from the degenerated pair (Kessler, 2001; Roundy et al., 2015). Thus, at first glance an effective multiplet could be considered indicative for a rather complex

spatio-temporal feature. But as we showcase here, it might as well simply result from DD.

   In the only spatial correlation case applied here, the temporal PC patterns are white noise (Section 2.2.1). Thus, the issue of one spatio-temporal feature being represented by two or more PCs is reduced to spatial features only. Effective multiplets are



built by PCs of which the orientation of their eigenvectors, i.e. their spatial patterns, in the DD sequence is ambiguous. Therefore, their patterns are very sensitive to even small variations of the analysed data. All the multiplet members in
combination describe a spatial feature of the data set. Thus, in case of a degenerated pair, a variation in the one pattern implies a complementary variation in the other. For the simple geometric shapes here, the pair´s spatial patterns from an ensemble of data sets simulated with identical spatial domain and spatial correlation properties will usually exhibit two predominant patterns with ambiguous ranking. Gradual variations of the predominant patterns and the switches in the ranking result simply from the randomness of the simulations.

For example, the two predominant spatial patterns of the degenerated pair formed by PC 3 and 4 from data sets simulated with rectangular domain (20 × 10 cells), isotropic spatial correlation with exponential decay and a correlation length of 10 cells (Figure 5b) randomly switch rank between distinct data sets (Figure 13). This results in the low stability of the PC 3 and 4 patterns from the respective ensemble (Figure 4b). For both PCs, the correlation of the ensemble´s patterns converge for long simulated time series around a mean $R^2$ of 0.5, indicating that the degeneracy of this pair cannot be resolved with longer
time series. The complementarity of both parts of the pair is visible in the ensemble´s mean and standard deviation patterns. The standard deviation pattern of PC 3 reflects an absolute variant of the mean spatial pattern of PC 4, and vice versa (Figure 6b and Figure 5b). The $R^2$s of the correlation between the two patterns were 0.64 and 0.78, respectively.

Symmetry of the domain shape triggers degeneracy (North et al., 1982). Thus, generally, it is recommendable to check spatial PCA results from data with symmetric domains for DD induced degeneracy. For example, the data sets simulated
with the square domain yielded the four effectively degenerated PC pairs PC 2+3, PC 5+6, PC 7+8 and PC 9+10 (Figure 4a). Again, the complementarity within the pairs yields standard deviation patterns of the one PC reflecting an absolute version of the mean spatial patterns of the counterpart PC (Figure 7). The $R^2$s between the respective two patterns were all larger than 0.81.

Asymmetrical distribution of locations diminishes the likelihood of DD induced degeneracy. In the subsampling variants
(Figure 10 and Figure S1) most of the degenerated PC pairs of the all cells variant (Figure 4) disappeared. The subsampling reduced the symmetry of the domain shape, resulting in a less ambiguous orientation for the eigenvectors. Consequently, the order of the DD sequence is clearer defined.

Effective degeneracy depends not only on the spatial domain but also on the effective sample size of the series, which equals here the time series lengths (Section 2.2.2). For example, in the triangular domain the effective degeneracy of the PC pair
6+7 which is prominent at a time series length of 2000 gradually disappears with increasing time series length (Figure 4c). Commonly, degenerated multiplets are detected qualitatively by checking for noticeably similar eigenvalues of PCs with adjacent ranks, forming steps in the sequence of the PC eigenvalues, or quantitatively with North´s rule of thumb (Figure S4). Analogue steps in the sequence of contrasts can serve as additional indication (Figure 12). With the stochastic DD reference method these steps are particularly pronounced, standing out as PCs of adjacent ranks with similar and rather low
contrasts given their position in the DD sequence, e.g. PC 2+3, PC 5+6, PC 7+8 and PC 9+10 for most correlation lengths in Figure S3a, and PC 3+4 for spatial correlation length of 10 cells in Figure S3b. It is an effect of averaging patterns that





switch ranks between the data sets from an ensemble. The magnitude of the drop depends on the specific patterns that are averaged.

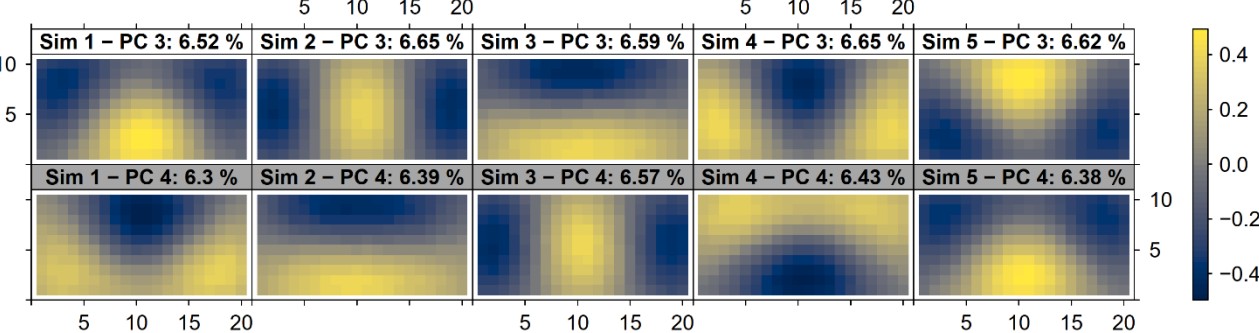

**Figure 13 Spatial patterns of the degenerated PCs 3 and 4 from five distinct data sets, each simulated with rectangular domain (20 × 10 cells), isotropic exponentially decaying spatial autocorrelation of correlation length 10 cells and time series length 10 000. Identical properties were used to simulate the ensembles from Figure 4b, Figure 5b and Figure 6b. The spatial patterns that belong to the same data set are plotted above each other with PC 3 on the top (white panel titles) and PC 4 on the bottom (grey panel titles). The index of the simulated data set and the fraction of assigned variance is given in the panel titles.**

## 5   Considering DD

The preceding section introduced different aspects of DD. It demonstrated the strong effect the domain´s size and shape can have on spatial PC patterns from S-mode PCA. In this section we continue with suggestions how to consider DD. Note, however, that S-mode is not the only option to analyse synchronous time series of one variable measured at multiple locations with PCA (Richman, 1986; Demšar et al., 2013; Isaak, et al., 2018). Recall that PCA summarizes the covariance of the data set´s variables. In S-mode PCA the time series from the different locations are used as variables and the values from the instants of time are used as observations. Thus, the covariance between time series from different locations is analysed. S-mode PCA can be used to identify locations with similar temporal patterns, for example groundwater wells with similar water level dynamics (Winter et al., 2000; Lehr and Lischeid, 2020). In T-mode PCA the instants of time are used as variables and the values from the different locations are used as observations, i.e. the S-mode data is transposed. Thus, the covariance between spatial patterns from different instants of time is analysed. T-mode PCA can be used to identify times with similar spatial patterns which can be useful in the analysis of system states. For example, Isaak et al. (2018) used it to identify winter and summer regimes of daily stream water temperature and the transition between both phases. Both modes are related and produce spatial and temporal PC patterns. Accordingly, DD should be considered in both. However, in hydrology T-mode PCA is way less common. So, for the introductory purpose here we restricted us to the S-mode case.





### 5.1. Detecting DD

#### 5.1.1.    Comparing spatial PC patterns from markedly different subdomains

The simplest way to check whether the spatial PC patterns of a data set are affected by DD is to visually compare the spatial PC patterns from sub-data sets with markedly different domains. It can serve as quick qualitative check to detect cases in which DD is a prominent feature. We recommend to perform partitioning of the original domain with basic geometric domain shapes like we used here. Thus, first take a subset with a square shaped domain, then taking from the squared shaped domain further subsets with rectangular and triangular domains of different orientation and compare the spatial PC patterns

of these subsets. This proceeding is demonstrated in the Demo script of the associated script selection (Lehr, 2024). If the subdomains are of similar size, the focus is primarily on the domain shape aspect.

Analogue, the PCA results can be checked for dependency from the selected domain size. However, we assume that commonly an analysis is focused on a specific scale and the domain size as well as the interpretation of results fit to that scale. Thus, usually the dependency from the domain´s shape should be more an issue than the dependency from its size.


#### 5.1.2.    DD reference patterns

DD reference patterns can be tailored for defined spatial domain and spatial correlation properties of a data set. Spatial PC patterns can be visually compared against the reference or checked for significant deviations from the reference with simple correlation analysis (Table 1). With the stability analysis of the stochastic method (Figure 4) or the confidence intervals of

the analytic method (Figure S4) it can be identified for each PC rank which time series length is required to reach stable and clearly defined DD patterns. Consequently, PCA results from (observed) data sets with identical spatial domain and spatial correlation properties but shorter time series have to be interpreted with the reservation that the DD might be stronger than the comparison with the reference suggests. The fractions of variance represented by the DD reference (Figure 11) can serve as benchmark, e.g. to consider only the patterns from PCs that represent more variance than their reference counterpart.

We introduced building DD reference patterns for data sets exhibiting isotropic spatial but no temporal autocorrelation. It enables to test the null hypothesis that the spatial PC patterns from observed data merely result from simple isotropic spatial autocorrelation between random white noise time series. The main feature of the null hypothesis is the ratio of spatial correlation length to the domain size, in particular to the distances between the data set´s locations. To our knowledge, such test was suggested first by Cahalan et al. (1996). They fitted models to observed precipitation and temperature data and

compared the eigenvalues and spatial PC patterns of observed and modelled data. Significant differences between the two eigenvalue spectra were considered to be "signal" and indicative for spatial anisotropies and inhomogeneities, "inhomogeneous processes", combined space and time correlation, or (secular) trends.





However, DD is not restricted to the isotropic case (see "directional functions" in Buell (1975)). An anisotropic example for our three basic domains is given in the supplements. Compared with the "default" isotropic case, the DD patterns are

distorted according to the direction and the ratio between longest and shortest spatial correlation length of the anisotropy (Figure S5 vs Figure 5). The spatial PC patterns tend to stabilize for shorter time series length (Figure S6 vs. Figure 4) and the PCs which form degenerated pairs are better separated (see the bigger differences between the fractions of assigned variance and the smaller magnitudes of the standard deviation patterns in Figure S7 vs. Figure 7). Both aspects reflect that the anisotropy gives a less ambiguous orientation for the DD sequence.

Elaborating on the DD of PC patterns from data sets with homogeneous autocorrelation in space and time is beyond the introductory scope here. However, spatially inhomogeneous temporal trends are indicative for distinct processes, modes or alike. They are likely to spread over more than one PC (Hannachi et al., 2007; Hannachi, 2007) and to affect the variance distribution among the PCs (Vejmelka et al., 2015). Thus, if the goal is not DD assessment but to construct reference patterns for the identification of distinct features, they should be considered.


## 5.2. Diminishing DD

### 5.2.1. Subsampling of domains

Analysing a subsampled data set with enlarged minimal distance between the locations can be used to diminish the DD of the PCA results. It can help to carve out features other than DD. On the other hand, informative local details might be

filtered out together with the excluded locations. If there is still DD, the new DD patterns of the subsampled data set might be harder to recognize visually because of the smaller number of locations per area. The choice of the threshold is critical for the analysis. Depending on the selected minimal distance, different features in the results might stick out, get diminished or even disappear. In any case, the spatial resolution of the analysed data set has to be considered in the interpretation of the results. Also, only stable PC patterns should be used to draw conclusions on the analysed system. The stable PC patterns are

those which are rather insensitive to the specific selection of analysed locations. They can be identified by comparing the PCA results from different subsamples (Smirnov, 1973; Lins, 1985a; Lehr and Lischeid, 2020).

### 5.2.2. Rotation of PCs

Another option to diminish DD is to rotate the PCs of interest (Richman, 1986; Dommenget, 2007). Different rotation

methods are available, e.g. varimax. The side effect is that the rotation changes the properties of the PCs. Depending on the applied method either the orthogonality of the eigenvectors, the uncorrelatedness of the PCs or both get lost (Jolliffe, 2002).




In addition, the variance among the rotated PCs is redistributed more evenly, potentially affecting which PCs are rated dominant (Jolliffe, 2002). Besides the selection of the rotation method, it has to be decided which PCs to rotate. Both choices influence the results and have to be considered in their interpretation.


## 6    Conclusion

Spatial patterns from S-mode PCA are regularly used for hydrological interpretations. In such analysis, homogeneous spatial correlation between the data sets` time series results in spatial PC patterns that are determined by the size and shape of the analysed spatial domain (domain dependence: DD). DD patterns are distinct, with strong gradients and contrasts. We showed that DD can come together with substantial accumulation of explained variance in the leading PCs. Thus, in contrast to what one might expect, neither distinct spatial PC patterns nor large fractions of explained variance in the leading PCs do necessarily indicate dominant hydrological processes or hydrologically meaningful properties. In addition, DD can induce effectively degenerated multiplets (effective multiplets). Without knowledge about DD, the multiplets can be misinterpreted as indication for complex spatio-temporal features. Without knowledge about multiplets, the multiplet members can be mistaken as effects of independent hydrological processes.

In summary, if DD is predominant, the spatial PC patterns do not reflect the hydrological functioning of the analysed system but rather the functioning of the PCA within the context of the data set´s spatial domain. Ignoring DD and effective multiplets easily leads to wrong hydrological interpretations. Consequently, DD should be considered for any application in which the PCs are used to draw conclusions about spatially distinct properties of the analysed system. In other words, it should be checked whether the spatial PC patterns differ significantly from patterns that result from the trivial case of nearby locations being homogeneously more related than those further apart.

Classical Buell patterns (PC 1: "mean behaviour", PC 2: gradient along the longest extent of the domain, lower ranking PCs: regular multipoles) and leading PCs with remarkably similar eigenvalues (effective multiplets) are an alert for DD. However, deviating patterns or clearly separated PCs are no contra-indication. DD patterns are original for every combination of spatial domain and spatial correlation properties. Thus, visual detection of DD is rather limited. Visual comparison of the spatial PC patterns from subdomains with markedly different shapes and / or sizes is practical merely as quick qualitative check.

To test whether spatial PC patterns differ significantly from DD patterns, reference patterns can be used as null hypothesis. We presented two methods to produce DD reference patterns. For the introductory purpose, we focussed on the stochastic method. The comparison of data sets simulated with identical spatial domain and spatial correlation properties showed directly the ambiguity of the PC ranking within DD induced multiplets, including the variations of the predominant patterns. Also, the work with simulated data is less abstract than the work with the analytic covariance matrix. For practical applications the analytic method is preferable. Its short computation time is a big advantage, especially when producing DD reference patterns for data sets with many locations.





**Appendix A – PCA reference patterns based on the analytic covariance matrix**

Deriving reference patterns with the analytic covariance matrix to evaluate PCA results was applied earlier by Cahalan et al. (1996) and Dommenget (2007). They modelled the evolution of a continuous meteorological field as stochastic spatially isotropic diffusion process, i.e. a spatial first order auto regressive (AR(1)) or "spatial red noise" process, and used the spatial PC patterns derived from the analytic covariance matrix of the model as null hypothesis for the spatial structure of climate variability.

Dommenget (2007) presented two adaptations of the analytic covariance matrix. In the first, for each pair of locations, the product of the standard deviations from the time series of the two locations is multiplied with their covariance calculated with the covariance function. The resulting spatial PC patterns provide the clean structure of the globally fitted covariance function weighted with the data set´s spatial distribution of covariance magnitude. In the second, the analytic covariance matrix is adapted to simulate the effect of areas with increased stochastic forcing. Areas with larger variance than the surrounding are defined and used for the weighting of the covariance matrix. In numerical experiments the effect of monopole, dipole or multipole structures in the data on the spatial PC patterns can be tested. Note that both variants are adaptations of the covariance matrix. Thus, other than in this study, the data must not be z-scaled prior PCA.

In addition, Dommenget (2007) suggested using the spatial PC patterns from an analytic covariance matrix as null hypothesis to find spatial PC patterns "that are most distinguished from those of the null hypothesis". These so called Distinct Empirical Orthogonal Functions (DEOFs) are derived by rotating the eigenvectors of the observed data to maximum difference in explained variance between the EOFs of observed data and those of the analytic covariance matrix. A Matlab script to perform DEOF analysis is available as supplementary material to Dommenget (2007). The DEOFs were suggested as starting point to identify teleconnections patterns or physical processes. Even though not in focus, DD patterns of the null hypothesis were observed and described as hierarchy of multipoles, "starting with a monopole as EOF-1, followed by a dipole, and then by higher order multi poles". In analogy to the spectrum of time series the DD sequence was interpreted as reflection of different spatial scales. The DEOF approach can be also used to compare the spatial variability modes from different data sets (Bayr and Dommenget, 2013). For data sets exhibiting temporal trends detrending prior applying DEOF is recommended (Hannachi and Dommenget, 2009).

**Appendix B – Effectively degenerated multiplets**

An eigenvalue is called degenerate if it is associated with more than one linearly independent eigenvector. That is, the eigenvalue is repeated (non-distinct), its multiplicity is larger than one. In the PCA case, the algebraic multiplicity of an eigenvalue (the multiplicity of the eigenvalue as a root of the characteristic polynomial) equals always its geometric





multiplicity (the dimension of its eigenspace) (Hefferon, 2020; Meyer, 2000) because PCA performs an eigenvalue decomposition of a symmetric matrix (see "spectral theorem for symmetric matrices", e.g. in Lay (2016) or "real spectral theorem" e.g. in Larson and Falvo (2009)). A degenerate eigenvalue together with its eigenvectors is called degenerate multiplet. The eigenvectors span the subspace of the degenerate multiplet. Within this subspace their orientation is not uniquely defined and they can be arbitrarily rotated (von Storch and Zwiers, 2003). Any linear combination of the

eigenvectors from the multiplet is as well an eigenvector of the eigenvalue (North et al., 1982; Hefferon, 2020).

In real-world data sets, perfectly symmetric distribution of variance such that degeneracy in the strict sense appears is unlikely to happen. However, if the eigenvalues of the "true population" are of very similar size, the sampling variability and errors can lead to "effective degeneracy" (North et al., 1982) with eigenvalues that are "indistinguishable within their uncertainties" (Hannachi et al., 2007) and eigenvectors that are random mixtures of the true population´s eigenvectors (North

et al., 1982).

This shall be illustrated with a slightly extended variation of an illustration given by Wilks (2006). We start with degenerated multiplets in the strict sense, i.e. an eigenvalue with more than one eigenvector. Consider a 3D point cloud with perfectly spheroid shape (idealized rugby ball). It has one long axis and two short axes of identical size. The cloud´s first eigenvector is aligned with the long axis and its eigenvalue depicts the variance of the cloud in this direction. The second and the third

eigenvector can be any pair of orthogonal vectors that are orthogonal to the long axis. They share a common eigenvalue. Thus, the variance representation is split in equal parts in the plane orthogonal to the first eigenvector. If we compare the eigenvectors from random subsamples of this data set, the orientation of the first one would be very stable, while the orientation of the second and third would exhibit large sampling variability. This correctly reflects the ambiguous orientation of the second and third true population eigenvector.

In the "effective degeneracy" case the eigenvalues are merely of very similar size. Consider again a spheroid shaped cloud but this time with the two shorter axes being of slightly different size (a slightly deflated rugby ball squeezed perpendicular to its long axis). Now the orientation of the second and third true population eigenvectors is distinct and both have distinct eigenvalues. Their share to the variance representation differs. If we compare again the eigenvectors from random subsamples of the data set, the question is whether the sampling is accurate enough to detect the slight difference in size of

the two shorter axes and whether the detection of the difference is stable among the subsamples? If this is not the case, the second and third sample eigenvalues are "effectively degenerate". Together with their eigenvectors they build an "effective degenerate multiplet". Thus, again the orientation of the second and third eigenvectors exhibits large sampling variability but this time because of the limited sampling accuracy. Due to the ambiguity of their orientation the pair is a potentially arbitrary mixture of the unknown true population eigenvectors (Wilks, 2006). Within the "accuracy range" determined by the

subsampling, the fraction of the cloud´s variance depicted by the plane orthogonal to the first eigenvector is approximated with the ratio of the sum of the multiplets´ eigenvalues to the sum of all three eigenvalues.





**Author contribution**

CL developed the stochastic method and the scripts, designed the experiments and carried them out. CL prepared the
manuscript with contributions from TH.

**Competing interests**

The authors declare that they have no conflict of interest.

**Code availability**

The R scripts and their documentation are available at https://doi.org/10.5281/zenodo.11213430.

**Data availability**

Data with the spatial domains and spatial correlation properties used in this technical note can be produced with the
associated scripts.

**Acknowledgements**

We thank Philipp Rauneker and Katharina Brüser from the Leibniz Centre for Agricultural Landscape Research (ZALF) and
Marcus Fahle from the German Federal Institute for Geosciences and Natural Resources (BGR) for the fruitful discussions.
Furthermore, we thank Gunnar Lischeid from ZALF for the in-depth discussions and detailed comments.

**Financial support**

The authors thank the ZALF and the University of Potsdam for the possibility to use the institutional resources.





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
