# Peer review of "Technical Note: An illustrative introduction to the domain dependence of spatial Principal Component patterns"

_Hydrology and Earth System Sciences, 2024_

## Referee Comment (RC1)

[referee-annotated manuscript omitted]

---

## Author Comment (AC1)

**Reply to referee 1**

Comments of referee 1 are in black.

Replies of the authors (AR) are in blue.
* * *
**Comments of referee 1**

https://hess.copernicus.org/preprints/hess-2024-172#RC1

Review of Technical Note: An illustrative introduction to the domain dependence of spatial Principal Component patterns by Lehr and Hohenbrink.

**Major comments**

This manuscript attempts to extend the study of how analyzing data on various shaped spatial domains affects the principal component loading patterns. The extension is both in content, as new material is added to the existing literature and the authors hope to gain the audience of hydrologists who, by and large, have not been exposed to such a concept. The importance of the work lies in several areas (expanded on below) but the key one is that if the PC loading patterns match those that are expected to arise from the shape of the domain, rather than the covariance fields, the recommendation should be a full stop on continuing. Therefore, understanding domain dependence is a necessary, but not sufficient condition, for physical interpretation of PC loadings.

Let me add that I like this paper and believe it can be a useful addition to the literature, helping analysts to interpret their eigenanalyses. Therefore, I hope the authors view my extensive comments with that in mind. If I come across as opinionated it is because of my lengthy work in this area and if it seems direct, that is my nature. Regardless, I like this manuscript and hope it gets published after further revisions.

AR: Thank you a lot for your comments and the appreciation of our work. Your comments are really helpful and the literature you suggested as well. Thank you also for the kind and respectful personal comment directly above, putting the degree of detail and accuracy in your comments into context.

We like to take the opportunity to put our replies to your comments into context as well. You spend a lot of effort to examine our work in detail and you offer us a lot of additional information in a high degree of detail. We highly appreciate that. If we refrain from some of your suggestions it is sometimes simply because it goes beyond the scope of what we want to offer here in the journal HESS. Our work is meant as an illustrative introduction for PCA users in the field of hydrology who probably never heard of domain dependence and its effects on the explained variance distribution, contrasts of the spatial PC patterns, degenerated multiplets, etc. - not to mention different scalings of PCA eigenvectors or scores, the congruence coefficient, etc.

To our feeling, the manuscript is already quite full with all the aspects we included. In fact, during the writing process we were even discussing to omit the whole multiplet aspect. Thus, we prefer not to extend it further with new aspects. Furthermore, we want to find a balance with

referee 2 who advocated mainly for restructuring the current manuscript instead of extending it further.

Now for the general comments. The paper builds upon the pioneering work of C. Eugene Buell. Those papers are cited. Buell (1979) left the reader with this final thought on the subject of domain dependence in the last line of his conclusions, stating that unless domain dependence was accounted for, on interpreting EOFs, "*Otherwise, such interpretations may well be on a scientific level with the observations of children who see castles in the clouds*". That is a pretty direct and strong statement. Digging deeper into why that can occur, the manner in which individual EOFs were being analyzed in the 1970s,...,2020s is by inferring physics by visual inspection of the magnitudes and gradients of the EOFs when plotted on maps. There was no external or internal validation of the patterns, only conjecture. With over 50 years of this practice, little attention was paid to whether this was a wise idea and thousands of such EOF studies emerged, with claims of the importance of the magnitudes and shapes of the patterns, many of which looked suspiciously like those patterns Buell generate. However, we should be wiser today and the authors are telling the investigator that if the covariance fields vary across a given domain shape but the same basic Buell patterns emerge, perhaps it is castles in the clouds rather than physics. However, there may be something more than a chimera, a mixture of signal and domain dependence. We come to learn later in the manuscript that a third confounding factor, namely the degeneracy of PC loading patterns with closely spaced eigenvalues, playing a role. It is good to see these factors considered.

Next, let's discuss PCA as a technique. According to those who understand the method, there is general agreement that PCA is useful for data reduction. In other words, in the type of analysis in the manuscript, the time series at n gridpoints or locations can have their covariances explained in k PCs where k<

1. Given the above prologue, the authors on lines 408-409 discuss "heavy constraints" of PCA that inhibit physical interpretation. To that good list, I'll add that it has been shown the leading PC, by virtue of the constraint of maximal variance can pull multiple unrelated sources of variation onto that leading PC, confounding physical interpretation. This should be added. The Karl and Koscielny citation (in your reference list already) shows this in their Appendix. Further details are given in the annotated manuscript (attached).

AR: Thank you very much. We will add it.

2. There is a general lack of agreement on terminology for eigenmodels, that leads to massive confusion among users of these techniques. At first when reading this manuscript, I thought the authors were applying EAOFs, only to change my opinion later in the manuscript that they were applying the PCA model. The original paper where EOFs were named EOFs, is generally attributed to Lorenz (1956). However, in that report, Lorenz refers to the displays as EOFs of space, and EOFs of time, to define what have now mutated somewhat into what are called "EOFs", and "Principal Components", respectively. Assuming a spatial analysis, those EOFs of space are unit length (sum of the squares of each EOF's coefficients = 1), whereas the EOFs of time are orthogonal vectors, each with a mean of zero and variance equal to the associated eigenvalue. In contrast, the PCA model, generally attributed to both Pearson (1901) and more fully to Hoteling (1933). weights (postmultiplies) the unit length eigenvectors (EOFs) by the square root of the corresponding eigenvalue to give "PC loadings". That seemingly minor change in the spatial patterns (keeping with the definition of space and time given for EOFs)

results in the time series calculation and properties being different. Those time series in the PCA model are called "PC scores" and have mean 0 and variance 1. They are also orthogonal. Flip the space and time definitions of these displays if the analysis is temporal. Because the two models result in different space and time patterns, they cannot be compared directly and the precise equations used are necessary to attempt to reproduce the findings of others. **I urge the authors to state clearly what model they are using immediately after the introduction and show the equation**. The situation becomes more complicated as users of these techniques tend to grab EOF/PCA code off of various statistical packages or Python code libraries, that often mislabel the results, never checking the specifics, thereby perpetuating the confusion. For the current paper, one must know if the analyses are applied to EOFs (unit length eigenvectors) or PC loadings (unit length eigenvectors postmultiplied by the square root of the corresponding eigenvalues). Further, it would be helpful to know if any of the results for domain dependence change as a function of the specific model invoked. **There is considerable confusion about this topic when reading this paper. It is important the model being used herein is stated unambiguously at the outset of this paper and the equation added in the methods section to avoid such confusion. Further adopt the correct terminology for that model and don't list any alternative terminology that might confuse the reader.**

AR: We agree, there is a lack of agreement in the literature regarding the terminology. For example, the distinction in PCA and EOF model you are suggesting, is only one option that can be also critizised (Jolliffe, 2002). Most often, we found the information that PCA and EOF have different roots but are interchangeable and that the related terms are used interchangeably (e.g. Hannachi et al., 2007; Wilks, 2006). It also appears in a paper that you are highlighting (Huth and Beranova, 2021). Thus, to our understanding it is not the main point here to decide between the models you are suggesting, but to clearly define the applied terms and to use them consistently throughout the manuscript. Here, we certainly agree and thank you for pointing out inconsistencies.

The PCA performed with function "prcomp" in R gives unit length eigenvectors (what you call EOFs). These are termed "loadings" in the documentation of the function in R. To our knowledge, this is also the way it is commonly used among PCA users in hydrology. These are the coefficients used to calculate the PCs. The PC scores have mean zero and the variance equals the associated eigenvalue (what you call PCs). Thus, prcomp applies what you call the EOF model.

As postprocessing step, we multiply the unit length eigenvectors with the square root of the corresponding eigenvalues (what you call PC loadings). Thus, the sum of the squared correlation loadings of a PC equals its eigenvalue. They are equivalent to the Pearson correlation of the PCs and the analysed variables, since we apply correlation matrix PCA. To emphasize this, we call them "correlation loadings".

We are aware that the term "correlation loadings" is is not commonly used. However, given the lack of agreement regarding the terminology, we prefer that the reader might stumble upon "correlation loadings" and is forced to check our definition rather than using the term "loadings" where the reader might think of either unit length eigenvectors or the scaled version.

We use correlation loadings here for several reasons. They provide the Pearson correlation range from −1 to 1 which is for most users easy to grasp. The common range also enables to directly compare the contrasts of the spatial PC patters from different PCs or PCAs (L167). Finally, it is prerequisite for the calculation of the stochastic DD reference patterns (L208).

In our set up here, the correlation loadings define the spatial PC patterns, the eigenvectors the unscaled spatial PC patterns and the PCs the temporal PC patterns. Note however, that the focus in our work is on the spatial PC patterns. The analysed time series are all z-scaled white noise series, resulting in the temporal PC patterns being white noise as well. Furthermore, the mean correlation loadings from the stochastic method, cannot be used to calculate scores, like with the classical loadings (L211).

For clarification, we will add the equations for the PCA in section 2.1 and for the correlation loadings in section 2.1.2 and rephrase the second paragraph of section 2.1.2 to:

"In correlation matrix based PCA, normalizing the unit length loadings $a_j$ of a PC j by multiplying it with the square root of its eigenvalue $\lambda_j$ is equivalent to the Pearson correlation between the scores of that PC and the analysed variables. Thus, the loadings are normalized to the commonly well-known Pearson correlation range from -1 to 1 which simplifies reading and interpretation of the PCA results. The sum of the squared correlation loadings of a PC equals its eigenvalue. Note that these normalized loadings are different from the "classical loadings", used in the linear combination to calculate the PC scores, which are not normalized to a common range. To prevent confusion, we use the term "correlation loadings" for the normalized loadings. In the following, the spatial PC patterns are described with correlation loadings c only.

$$c_j = a_j * \sqrt{\lambda_j}$$

For S-mode PCA, the normalization enables direct comparison of the contrasts of spatial patterns from different PCs or PCAs. Here, we define the contrast of a spatial PC pattern as the range between the minimum and maximum of the correlation loading values of that PC. Thus, the maximum contrast possible would be 2."

Hannachi, A., Jolliffe, I. T. and Stephenson, D. B.: Empirical orthogonal functions and related techniques in atmospheric science: A review, International Journal of Climatology, 27, 1119-1152, https://doi.org/10.1002/joc.1499, 2007.

Jolliffe, I. T.: Principal Component Analysis, 2nd ed., New York, Springer, 2002.

Wilks, D. S.: Statistical Methods in the Atmospheric Sciences, 2nd ed., Elsevier, 2006.

3. The treatment of eigenvalue degeneracy is generally well addressed with one exception that potentially plagues nearly every applied eigenanalysis: eigenvalue degeneracy at the truncation point (k). If those PCs associated with closely spaced eigenvalues between k and k+1 have information that is intermixed, problems arise and data is intermixed with noise on the kth retained PC loading vector. Your paper presents 10 PCs, therefore, the spacing between the 10th and 11th eigenvalues should exceed the North et al. criterion. Does it? Let the reader know.

Further, this needs to be mentioned because it can cause the loss of a domain dependence pattern simply because the way eigenvalues are ordered in descending order makes them more likely to be closely spaced as the smallest eigenvalues head toward the tail (presumably noise) where the analyst would normally truncate the analysis to discard the k+1,...,nth eigenvalues, perhaps using some other criterion (e.g., based on percent variance extracted, eigenvalue magnitude).

Related to this, I wonder why eigenvalue degeneracy is not addressed earlier in the paper as it seems to affect domain dependence. If that is the case, then consider moving it earlier in the paper as those PC loadings arising from degenerate multiplets should not be expected to exhibit the domain dependent patterns but the multiplet may be dominated by the domain dependent patterns and those are intermixed into new patterns that don;t seem to be domain dependent patterns.

AR: Thank you. We will include the aspect of eigenvalue degeneracy at the truncation point. Therefore, we will expand the warning to split multiplets in L425 with an explicit statement about the truncation point:

"In particular, special care has to be taken that the truncation point of a PCA does not split a multiplet."

Here, we show the first ten PC patterns merely for illustration. We found it to be a good balance between showing the DD pattern sequences and some degree of detail, but not too much detail that it is still visually easy to grasp. Also, to our experience most S-mode PCA applications in hydrology use substantially less than ten PCs, our casual guess would be around four. To clarify this, we will add an explanation to the truncation point used in our study in L260:

"Note, that here and in the following we show the results for the first ten leading PCs. The decision was taken merely for the illustrative purpose. We found it to be a good balance between showing the DD pattern sequences and some degree of detail, but not too much detail that it is still visually easy to grasp. There was no other specific truncation criterion, e.g. based on eigenvalue magnitude or percent variance extracted, applied."

We did not check for degenerated multiplets formed by PC 10 + PC x, because we did not analyze the PC 10 patterns further and used them only as examples for illustration. We will state this explicitly in L473, including your hint on intermixing of signal and noise:

"Note also, that intermixing might be easier overlooked for the smaller eigenvalues that are more closely spaced. If the analysist selects PCs to separate noise from signal, this could possibly result in truncation within a multiplet and consequently intermixing of noise and signal in the last considered PCs. Here, we selected the first ten PC merely for the illustrative purpose (section 4.1). If the goal would be to further analyze PC 10, it would be necessary to check its patterns for intermixing - also with the subsequent PCs, in particular PC 11. Indications for intermixing in the PC 10 pattern can be seen in the stability plots of Figures 4a, 10c, S1a+c. In case of Figure 10c, PC 9 does not show sign of intermixing, thus, in this case the intermixing is probably with PC 11."

Thank you also for your hint that intermixing might mask (expected) DD patterns. We will add a short paragraph on that in L453:

"Note, however that degeneracy might cause domain dependent patterns that don´t seem to be DD patterns because they are intermixed into new patterns. For example, in Figure 13 the patterns of the multiplet pairs of simulations 1, 4 and 5 exhibit different patterns than those of simulations 2 and 3."

Regarding the order of our sections, we like to point out again that the motivation of our work is to provide an illustrative introduction for PCA users in hydrology. We come back to this because it relates a lot to what we are presenting in which order. We start with simple examples

to introduce the general phenomenon. Then, we get more specific and complex. Step by step, we focus on different aspects of DD and link it to PCA features we assume to be of interest for the PCA practitioners in hydrology and regularly used.

In section 4.1, we use the Buell patterns to introduce the general phenomenon, the concept of stability of the PC patterns and the use of the scripts. In section 4.2, we continue with the domain shape aspects, including irregular distribution of the locations. In section 4.3, the ratio of domain size versus spatial correlation length and its effects on the explained variances and the contrasts of the DD patterns comes into play. Thus, in each subsection of section 4, we introduce new aspects, building on the earlier ones.

We assume that the effectively degenerated multiplets will be the most abstract and difficult to grasp part for most of our readers, and probably also the furthest away from what they are used to. That is why the degenerated multiplets come in their own section as the last of the phenomena we want to introduce.

4. Comparison of PC loading patterns is accomplished with correlations. S-mode PC loading (and that of EOFs) interpretation depends on the magnitude of the PC loadings plotted on a map (and in general, the magnitude of the PC loadings/EOFs is important in any mode). Therefore, correlations subtract each PC loading/EOF vector mean (pattern mean), so two patterns with different means can have their large correlations, yet their magnitude patterns will be much different and the grid boxes (I think what you refer to as cells) with the maximum PC loadings will be in different geographical (or topological) locations in your domains. If that is the case, the the correlation is suboptimal for such comparisons. Find a better metric that includes magnitude in terms of comparison. I suggest the congruence coefficient, though others exist that preserve the vector magnitudes.

AR: One of the major benefits of Pearson correlation is that it is well known and the results in terms of r or $R^2$ can easily be contextualized by the reader. We assume that most, if not all, readers of the manuscript will know it from their own work. This includes the well-known sensitivity to outliers (see your comment on L175). To our experience Pearson correlation is regularly used in hydrology for the comparison of spatial patterns from PCs and potential explanatory variables or other patterns of interest.

Note also that all PC patterns in our study here are (1) from correlation matrix based PCA, thus, all analyzed variables are z-scaled, and (2) correlation loadings, thus, scaled to the common range [-1, 1]. Furthermore, all PC patterns that are compared with Pearson correlation are based on data sets simulated with either (a) identical spatial correlation properties and identical domain (step 2 of the stochastic approach) or (b) identical spatial correlation properties (the correlation exercise with the all cells variant and the spatially homogeneous and heterogeneous subsampling variants in section 4.2.).

We assume that for our examples here, neither differences in magnitude of the patterns nor the effect of the pattern mean subtraction by the correlation analysis versus deviations from zero by the congruence coefficient (see your comments on L192 and 336) are much of an issue. Thus, for our analysis and the introductory purpose here, we think simple Pearson correlation is sufficient.

Therefore, we prefer to keep it simple and stay with Pearson correlation for the presented analysis and results and include a discussion on the limitations of using Pearson correlation and the benefits of the congruence coefficient in section 5.1.2..

5. It seems odd that after the paper establishes the details and importance of domain dependence, it has no results on how rotating those PCs affects such dependence. There is only a scant mention of the possibility of this near the end of the paper, mostly in the context of rotating degenerate multiplets. However, rotation can be applied to PC loadings associated with non-degenerate eigenvalues and it will affect domain dependence patterns. Please consider adding a section on rotation and show those patterns to comment about how domain dependence is addressed by post processing the PC loadings with a rotation.

AR: In our study here, there are no physical structures, hydrological signals, modes or processes integrated in the simulated data sets. Thus, there are no signals to detect. The idea of our work was to demonstrate to PCA users who are not aware of DD and the related discussion that even without any physical structures, processes or modes, suggestive patterns can emerge. The numerical experiments are designed for that. Adding new numerical experiments to evaluate the performance of unrotated versus rotated PCs in identifying (hydrological) signals is beyond our introductory scope here.

This is a basic difference to the extensive study of Compagnucci and Richman (2008) who simulated data sets with a series of typical atmospheric flow patterns (plasmodes) to test the performance of unrotated versus rotated PCs, both from S- and T-mode PCA, in recovering the plasmodes. We agree that it would be very interesting and valuable to perform a similar study with typical hydrological signals instead of the atmospheric plasmodes. However, we think that this is material for another standalone study.

Here, we will extend the section 5.2.2 on rotation and include your literature recommendations and some of your thoughts and hints. See also our reply to your specific comments to L341, 557, 558, 573, 574.

6. The manuscript discusses accounting for domain dependence prior to attempting **physical interpretation**. Both the abstract and the introductions discuss how ignorance about domain dependence can easily lead to the wrong interpretations of PCA results (e.g., "Ignorance about DD can easily lead to the wrong interpretations of PCA results. DD patterns are distinct, with strong gradients and contrasts, and therefore highly suggestive to indicate physically meaningful drivers or properties of the analyses system". I agree with this statement and, assuming it is valid, the reader will want to know abut the right interpretations of PCA results. The manuscript further states (correctly) that the analyses proceed from data that are formed into a correlation (or covariance) matrix, either explicitly and implicitly and that matrix (or the standardized data in the case of SVD) are decomposed into eigenvectors that should be capable of summarizing the correlations/covariances of the data (after ensuring they do not represent domain dependence patterns). Therefore, some additional discussion of how to interpret those eigenvector (in the case of the present manuscript, PC loadings and PC scores), after passing a domain dependence assessment, must be added. It seems the majority of patterns shown in the paper suffer from domain dependence or from the effects of eigenvalue degeneracy combined with domain dependence. Would that be the null hypothesis for other investigators?

The main recommendation to assess such a hypothesis of domain dependent patterns (according to the manuscript) seems to be to visually assess the similarity but it leaves the reader asking, "then what do I do?". Presently, there is a suggestion to visually assess the analyzed patterns and compare to the domain dependent patterns for a similarly shaped domain. Two issues with visual assessment are (a) the reliability of the same pattern under the eyes of different analysts may well have one analyst believing there is a strong resemblance, and the pattern should not be further interpreted, yet a second analyst may think it has some resemblance but not that much

to reject it as domain dependent. Further, (b) the nature of a qualitative visual assessment means any one analyst can see some resemblance to domain dependent patterns in their visual assessment and then discount it based on personal bias. A more quantitative approach to avoid (a) and (b) would be a direct numerical comparison using a matching coefficient (e.g., congruence coefficient). In that case, a recommendation could be made, such as, if the congruence coefficient exceeds some value (e.g., > 0.8), the analysis is dominated by domain dependence and the unrotated PC loadings/EOFs should not be analyzed physically. The assessment of the physical interpretation gets even trickier at this point. If the PC loading pattern based on either visual assessment or congruence coefficient value is thought not to be sufficiently contaminated be domain dependence, it does not mean it is physically interpretable as a meaningful mode without further investigation. Recall what the PCA does. It summarizes the correlation/covariance structure into a set of k PC loadings and k PC scores. Do we know if any of those structures relate well to the correlation/covariance matrix from which they were drawn? Without such a step, physical interpretation would seem unwise (we're back to the castles in the clouds but now from the "heavy constraints"). Because the manuscript is motivated by finding physically important modes, a revised manuscript should address or provide some suggestions on how to confirm if a mode is physically realistic or related to the correlations/covariances (or not). There is some literature on this topic, ranging from never physically analyze any PC structures (in that case domain dependence is moot because domain don't affect the ability of PCA to extract most of the variance from a dense correlation/covariance matrix) to, in many cases, the PC structures can be analyzed after confirming similarity to the correlations/covariances . I suggested examining the Compagnucci and Richman (2008) and Huth and Beranova (2021) papers for starters. The latter asks the specific question about what is a "true mode" whereas the former addresses the question about if certain analysis modes can retrieve the modal patterns. Of course, there are other alternatives, such as using a technique not rooted in eigenvectors. However, if the paper offers a path to identifying domain dependence that undercuts physical interpretation, some remedy should be offered.

AR: Yes, the suggestion is to use DD reference patterns as null hypothesis (see L175, 520-527, 582-589 and Appendix A). In the revised manuscript, we will include a discussion on the limitations of the visual assessment and the use of Pearson correlation and the congruence coefficient for numerical comparison in section 5.1.2.. See also our reply to your major comment 4.

We agree that DD is just one of the aspects that should be checked prior physical interpretation. It is not enough to check whether the patterns are sufficiently free from DD. Or as you stated in the beginning of your major comment section, it is "a necessary, but not sufficient condition for physical interpretation" (see also your comment on L575). Thank you for the references and the different hints in your comments to the physical interpretation of the PC patterns in L23, 72, 81, 408, 428, 574, 575 and your major comment 1. We will include them, expanding the discussion on physical interpretation of the PC patterns in Lines 406-409 and move the whole discussion to a new paragraph after L73 in the introduction.

Regarding the identification of physically realistic modes, we will include references to the work of Compagnucci and Richman (2008) and Huth and Beranova (2021) in the extension of section 5.2.2. (see our reply to your major comment 5).

In hydrology, spatial PC patterns have been also used to describe the spatial variability of distinct hydrological signals, processes or physical properties (L78-80). Building on the idea with the plasmodes (see our reply to your major comment 5), we think it would be very

interesting to conduct more numerical experiments with hydrological simulation models to test whether any of the implemented hydrological features of the model can be uncovered with the patterns of the PCs. The test data could be, for example, spatially distributed groundwater level series simulated with a groundwater model. Again, this could include the comparison of the performance of unrotated versus rotated and / or S- versus T-mode PCA. In the revised manuscript we like to include these ideas as an outlook to future research at the end of the conclusion (see also our reply to your comment on L573).
* * *
**Specific comments**

Numerous specific comments are listed in the annotated manuscript (attached).
**Citation**: https://doi.org/10.5194/hess-2024-172-RC1

**L23:**
What is the proper "interpretation" for PCA? At one end of the spectrum, some might claim PCA is simply a data compression technique with little or no possibility of physical interpretation. At the other end, some may claim that the individual PCs can be interpreted as physically meaningful entities. Some discussion of this might be in order prior to the discussion of domain dependence, particularly as Buell (1979), in the final sentence of his conclusions, states, "Otherwise, such interpretations may well be on a scientific level with the observations of children who see castles in the clouds."

AR: We will elaborate on that in a new paragraph after L73 in the introduction. See also our reply to major comment 6.

**L36:**
This may be the situation; however, there is a general lack of agreement on these models, that leads to massive confusion among users of these techniques.

The original paper where EOFs were named "EOFs", is generally attributed to Lorenz (1956). However, in that report, Lorenz refers to the displays as EOFs of space, and EOFs of time, to define what have now mutated somewhat into what are called "EOFs", and "Principal Components", respectively. Assuming a spatial analysis, those EOFs of space are unit length (sum of the squares of each EOF's coefficients = 1), whereas the EOFs of time are orthogonal vectors, each with a mean of zero and variance equal to the associated eigenvalue.

In contrast, the PCA model, generally attributed in idea to Pearson (1901) and more fully to Hoteling (1933). The PC loadings in that PC model weights (postmultiplies) the unit length eigenvectors (EOFs) by the square root of the corresponding eigenvalue to give "PC loadings". That seemingly minor change in the spatial patterns (keeping with the definition of space and time given for EOFs) results in the time series calculation and properties being different to close the PC model. Those time series in the PCA model are called "PC scores" and have mean 0 and variance 1. They are also orthogonal.

Flip the space and time definitions of these displays if the analysis is temporal.

Because the two models result in different space and time patterns, they cannot be compared directly and the precise equations used are necessary to attempt to reproduce the findings of others.

The situation becomes more complicated as users of these techniques tend to pull EOF/PCA code off of various statistical packages or Python code libraries, that often mislabel the results, never checking the specifics, thereby perpetuating the confusion.

For the current paper, one must know if the analyses are applied to EOFs (unit length eigenvectors) or PC loadings (unit length eigenvectors postmultiplied by the square root of the corresponding eigenvalues). Further, it would be helpful to know if any of the results for domain dependence change as a function of the specific model invoked. **There is considerable confusion about this topic when reading this paper. It is important the model being used herein is stated unambiguously at the outset of this paper and the equation added in the methods section to avoid such confusion. Further adopt the correct terminology for that model and don't list any alternative terminology that might confuse the reader.**

Lorenz, E. N., 1956: Empirical orthogonal functions and statistical weather prediction. Statistical Forecasting Project Rep. 1, MIT Department of Meteorology, 49 pp.

Pearson K. On lines and planes of closest fit to systems of points in space. Philosophy Magazine. 1901;2(6):559-72.

Hotelling, H. (1933) Analysis of a complex of statistical variables into principal components. Journal of Educational Psychology, 24, 417-441. http://dx.doi.org/10.1037/h0071325

AR: We addressed this comment in our reply to major comment 3.

**L72:**
Although I agree with this sentiment, showing that it is reasonable to expect any physical conclusions to be drawn from PCA should be discussed first. Assuming the basic covariance structure carries important information about the physical processes, do the PC loading patterns (or EOFs) relate well to the underlying covariance structures? The investigator must confirm this prior to expectation of the patterns being related to physical processes. Add a few sentences on that.

AR: We will do so in a new paragraph after L73. See also our reply to major comment 1 and 6.

**L81:**
True, but domain dependence is one of several factors that hampers physical interpretation of the system. Others include:

(1) Data are related either explicitly or implicitly (by expressing them in anomaly or standardized anomaly form) via covariances or correlations. Such matrices express only the linear relations in the data. Further, subtracting a mean in either covariances or correlations assumes stationarity of the mean and variance, often violated by processes such as climate change and hydrology affected by climate change.

(2) The eigenanalysis technique is limited to only linear relationships between the covariances or correlations and the EOFs/PC loadings.

(3) The first eigenvector extracts maximal variance and often pulls in different sources of variability onto the leading vector, mixing the different sources. This is shown clearly in Karl and Koscielny (1982) in their Appendix Fig. 14A (top panel), where their data vectors X1, X2 and X3 are highly related as group 1 and where their vectors X4, X5 and X6 are also highly related as group 2, but groups 1 and 2 are nearly orthogonal. However, as they show, the first PC lies directly between groups 1 and 2, thereby describing neither accurately and introducing distortion by merging them so that their projections (PC loadings) would all be positive of nearly equal magnitude on PC loading 1, suggesting to the investigator that there is only one grouping.

(4) All eigenvectors, beyond the first, are orthogonal to all the previous eigenvectors, and the hydrological processes are rarely if ever orthogonal.

AR: We will include these aspects in a new paragraph after L73. See also our reply to major comment 1 and 6.

**L93:**
This would include the degree of linear association and the scale of the spatial correlation or covariance field with respect the the domain size.  One might hypothesize that processes that are either weakly linear or mostly nonlinear and the linear part is relatively small, that domain dependence might dominate.  Similarly, if the size of the spatial covariance/correlation data function is nearly the same size as the spatial domain used, the domain dependence might be different than for small scale processes, of perhaps 1/3rd of the domain size.

AR: We demonstrated this and analyzed the effect of domain size and spatial correlation length in section 4.3. The analysis of non-linearity is beyond our introductory scope here.

**L96:**
I'm not sure other papers have examined this. If that is true, add it as a unique aspect of this research.

AR: We do not examine this here. However, the blurring effect of measurement errors can be simulated with the stochastic method, using relatively short time series with rather unstable spatial PC patterns. We will include a statement on this in L284.

**L99:**
Define "low ranked PCs". I'm assuming these are PCs associated with smaller eigenvalues. If so, state that. If not, define it explicitly. If it is for the eigenvectors associated with small eigenvalues,  the work of North et al. (1982) [cited extensively in your manuscript] and Quadrelli et al. (1989) on degenerate multiplets claim intermixing of the variance structures when he eigenvalues between adjacent eigenvectors are close in magnitude. That may be what you are seeing here.

Quadrelli, Roberta, Christopher S. Bretherton, and John M. Wallace. "On Sampling Errors in Empirical Orthogonal Functions." Journal of Climate 18, no. 17 (September 1, 2005): 3704–10. http://dx.doi.org/10.1175/jcli3500.1.

AR: Yes, your assumption is right. We will define it there and include a few sentences to the intermixing and rotation together with the references you provided:

"This gets less clear for those of the PCs with smaller eigenvalues (low ranked PCs). They are more finely detailed and less robust against deviations from Buell´s settings. Furthermore, there might be intermixing of the variance structures when the eigenvalues from successive eigenvectors are of very similar size (North et al., 1982; Quadrelli et al., 2005). These PCs which are not well separated with the PCA are called effectively degenerated multiplets (North et al., 1982). For their separation, additional post-processing is required, e.g. rotation of eigenvectors (Richman, 1986; Jolliffe, 1989)."

**L99:**
becomes

AR: We will change according to your suggestion.

**L101:**
Yes but that could be the aforementioned intermixing of the variances. In fact, it is possible that even Buell patterns for large sample sizes may intermix if the eigenvalues between successive eigenvectors are very close in magnitude.

AR: We agree. This is addressed in our response to your first comment on L99 above.

**L116:**
Here you mean "PCs which are not separable without additional post-processing (e.g., rotation of the PC loadings to separate the sources of variability for all eignvectors or for those eigenvectors with closely spaced eigenvalues)." Richman (1986) and Jolliffe (1989). You cite both these papers presently but the logical conclusion is missing.

AR: Yes. We will include your specification on the post-processing / rotation in the newly added lines where we introduce the effectively degenerate multiplets (see our reply to your comment on L99) and reduce the sentence here to "... and c) effectively degenerate multiplets."

**L135:**
See my earlier comment. It is more serious than terminology. The original models of EOF versus PCA have specific terminology  and give different results. Those model display names have mutated over time and the terminology has been intermixed. Unless one state the mathematics of the model invoked, the reader has little idea what model is invoked. That also means that any conclusions for the EOF model need to be verified for the PCA model as the EOFs vs. PC loadings have different magnitudes and properties and the PCs vs. PC scores have different magnitudes and properties.

AR: Please see our reply to major comment 2.

**L138:**
It appears this is the EOF model. If so, despite Jolliffe's terminology, the majority of the literature envoking EOFs will call the displays EOFs and PCs (for space and time displays, respectively). Wilks textbook has a reasonable section on the varied terminology of EOF/PCA, although even that excellent book it is not exhaustive in this regard. You might try to simplify the sentence here where eigenvalue, scores, eigenvectors are mentioned. Alternatively (and perhaps the superior solution) would be to show the compact equation(s) in this manuscript for the model being invoked and that would clarify any confusion by the readers.

**Addendum...**After reading further, it seems you may have used actual PC loadings and PC scores. If so, ignore my comments about not using PC scores but, in that case, drop the discussion of EOF beyond the beginning of the introduction, as it serves only to confuse the readers. Further, clearly state what model you are using in section 2.1 with the equation for that model.

AR: Please see our reply to major comment 2.

**L140:**
See earlier comment. Please refrain from using the term "PC scores" if you are using an EOF model. PC scores are defined only for the PC model that weights the unit length eigenvectors by the square root of the corresponding eigenvalues. For the present manuscript, if you are using EOFs, then you can call them PCs and drop the "i.e., the PC scores". However, if you are using the PCA model, then use the terminology PC loadings and PC scores but drop other terms that will serve to confuse.

Addendum...See comment above. There is a need to clearly define the model used, define the appropriate terminology of the model displays and use on those terms throughout.

AR: Please see our reply to major comment 2.

**L143:**
This sentence seems to be awkward or a fragment at a minimum. Please clarify.

AR: "The eigenvectors of all PCs define the orthogonal basis of the new ordination system into which the analysed data is projected (orthogonality constraint)".

**L143:**
"mapped to" what? This sentence seems incomplete.

AR: Please see our reply to your comment on L143 above.

**L147:**
Yes, see earlier comment. Often the process of maximal variance extraction runs counter to interpretation of the sources of variability in physical systems (unless there is a single physical mode that encompasses the full extent of the domain where PC 1 can explain it -- rarely the situation). I mentioned this earlier and pointed to the Appendix of Karl and Koscielny (that you cite in this manuscript).

AR: Please see our reply to your comment on L81.

**L154:**
This is vague as "PC series" is undefined.

In the **EOF model:**

(1) the EOFs are not uncorrelated because their mean is not zero. However, they are orthogonal (if fact, orthonormal) by either column or by row as V'V and VV' = I.

(2) The PCs are uncorrelated as their means are zero. Additionally, their variance is the eigenvalue. They are uncorrelated by column and therefore orthogonal by column.

==========

**In the PCA model** (with eigenvectors scaled by the square root of the eigenvalue):

(1) the PC loadings are not uncorrelated because their mean is not zero. However, they are orthogonal by column only (their diagonal is the eigenvalue), V'V = D.

(2) the PC scores have mean = 0 and variance = 1. They are uncorrelated by column. Because they're uncorrelated, they're orthogonal by column with the diagonal equal to the degrees-of-freedom (normally, that would be n-1 if the correlation/covariance matrix is not singular).

Given these differences, hopefully you can appreciate the importance of stating unequivocally the specific model invoked. Specifically, where you say: All PC series are linearly uncorrelated with each other" is incorrect for both the EOF and for the PCA model. for the EOFS or for the PC loadings, as neither has zero mean column vectors.

**Therefore rephrase "PC series" in this sentence and be precise to specify uncorrelated by row or by column.**

AR: The term "PC series" means here the temporal PC patterns. It evolved somehow informal among colleagues as a short form for the PC scores which in the S-mode PCA case are time series of the same length as the analysed time series. Thank you for pointing out that it causes confusion without this background. We will replace it the two times it appears in L154:

"All temporal PC patterns are linearly uncorrelated with each other, each temporal PC pattern is associated with a spatial pattern and all spatial PC patterns are orthogonal to each other."

Please see also our reply to your major comment 2 about the PCA terminology.

**L155:**
Two comments:

1. The discussion prior to this section seemed ambiguous as to which eigenmodel was being invoked. Please fix that.

2. Does this imply that other modes of PCA do not suffer from domain dependence? For example, Q-mode is a field x station data matrix, giving a station x station covariance matrix.

AR:

To 1.: Please see our reply to your major comment 2.

To 2.: We did not investigate that and it is beyond the scope of our work here. However, we would hypothesize that in case of a homogeneous correlation structure, it can be an issue there as well.

**L163:**
Earlier, it seems that you were using EOF, here is suggests you are using PCA. State clearly from the outset which model is being used and stick with that terminology. If it is the PC model, them PC loadings and PC scores.

**L167:**
Two comments:

1. The desire to have PC loadings within the same range would fit the idea of EOFs better as those are all unit length eigenvectors. It may explain why Buell used EOFs, rather than PCs, to describe domain dependence. That said, there is nothing preventing any arbitrary scaling of the eigenvectors, as long as point (2) is noted.

2. The loadings normalized to anything other than the square root of the corresponding eigenvalue will no longer close the PC model using the standard formulation. I think that is what you are attempting to say in the last sentence, but it could be clarified.

AR: Please see our reply to your major comment 2.

**L175:**
Two comments:

1. Pearson correlation is leveraged by outliers. If two maps are being compared with relatively few common points on both maps extreme in the same direction, but the remainder of the gridpoints not in agreement, the correlation may be large.and exceed some t-test at alpha = 0.05. Because of that, field significance should be examined for difference fields of the maps, the pairwise comparison of spatial patterns of the combinations of PCs, or use a resistant statistic. For the field significance, here is an excellent test:

On "Field Significance" and the False Discovery Rate: By D.S. Wilks, Journal of Applied Meteorology and Climatology, Volume 45, Issue 9, 2006, pages 1181–1189.

2. t-tests assume Gaussian distributions. Correlation distributions are not Gaussian, particularly in the tails (where most of the matches would occur) because its range is limited to -1 to +1. You could Fischer z-transform the correlations first to partly mitigate this or, better yet, apply a permutation test to the maps, as the permutation test is distribution free.

AR: We think for our purpose here, simple Pearson correlation and the t-test are sufficient. This combination was also used in the study by Huth and Beranova (2021) you recommended.

For a general comment on why we use Pearson correlation, please see our reply to your major comment 4.

**L192:**
Because the mean of the PC patterns (i.e., the mean of each vector of PC loadings) is not zero, and the interpretation of the PCs is a function of the magnitude of the PC loadings, the correlation of PC loading vectors by subtracting out the mean, is an inferior metric for PC loading comparison. Lorenzo-Seva and ten Berge (2006) make a good case for the congruence coefficient, which does not remove the mean prior to the comparison. This metric has been used in the geosciences literature for such comparison.

Lorenzo-Seva, U., & ten Berge, J. M. F. (2006). Tucker's congruence coefficient as a meaningful index of factor similarity. Methodology: European Journal of Research Methods for the Behavioral and Social Sciences, 2(2), 57–64. https://doi.org/10.1027/1614-2241.2.2.57

AR: Thank you for the literature recommendation. For our reply, please see our reply to major comment 4.

**L193:**
Explain why the ranks are substituted for the values.

AR: We are not sure what you mean here. Assuming that you are asking why the analysis is performed separately for each PC rank, this is why we want to identify the stability of the spatial patterns and of their ranking prior calculating the mean spatial PC patterns.

**L194:**
Given the earlier comment about congruence coefficient and the use of correlations here, using the word "congruence" here is not good form.

AR: We will replace it with "similarity".

**L199:**
Similar comment about the distributions of the variability. Do you test for symmetry to determine the variance in each tail is approximately similar (and hence a single standard deviation holds)? One way to test that is to calculate the skewness of each of these patterns to decide if the skewness magnitude exceeds 0.5 and therefore would not be sufficiently symmetric to assume symmetry.

AR: No, we did not test that. We assume that in our case here, in which the compared PC patterns stem from data sets simulated with an identical parametrisation, it is negligible, especially for the mean spatial PC patterns which are calculated based on a large number of simulated data sets (100) with rather long time series length (10 000).

**L225:**
Great. That was my earlier suggestion. Extend the commentary to the instability resulting from degeneracy arising from closely spaced eigenvalues.

AR: Thank you. Assuming you are referring to your comments in L99-116, please see our reply there. To improve the structure of the method section, we will move L222-228 "Confidence limits ... both eigenvalues." in to a new section 2.3 "North´s rule of thumb".

**L242:**
Does "prcomp" give unit length eigenvalues or PC loadings? I suggest checking this manually as some of the R codes I have investigated say the output is one thing but really supply something else or simply wrong.

AR: "prcomp" gives unit length eigenvectors. See also our reply to your major comment 2.

**L250:**
Define "cell". It appears in two previous Figure captions too. I assume cell means grid box but a formal definition is required.

AR: Yes, you are right. We will add the following definition in L182:

"The grid cells (cells) of the random field represent the locations of a data set."

**L255:**
Two comments:

(1) The x-axis on Figure 4 presents some challenges because the behavior normally asymptotes after about 2000 observations for the first few PCs. However, by showing 10,000, the reader cannot pick out the number of observations required to find sufficient stability in those leading few PCs.

(2) By the time an analyst extracts PCs beyond the first few, one wonders how many degenerate multiplets emerge. The problem with not knowing that is if the adjacent eigenvalues are separated sufficiently to exceed the North et al. criterion, then fewer observations than Figure 3 suggest are required to provide stability.

Conversely, if the leading eigenvalues in PCs 1-3 are separated by less than the North et al. criterion, the results shown in Figure 3 may be too optimistic compared to such cases with degenerate multiplets in the first few PCs.

**Some comment on the eigenvalues and their separation in these examples is critical to interpret Figure 3 and the unravel the effects of degeneracy arising from multiplets from that of domain dependence. Ideally this would precede the discussion of domain dependence as that effect seems to affect domain dependent patterns.**

AR: Figure 4 is meant as an overview figure. Section 4.1. is meant as starting point from which we introduce step by step new aspects in the following sections. Thus, at this point in the manuscript we do not want to get more detailed. We also do not want to open the topic with the degenerate multiplets at this early stage. We explain the logic of the order of the sections and why the degenerate multiplets come last in our reply to your major comment 3. Please see there.

**L260:**
Useful information but for applied research rarely are there 10,000 observations, much less 10,000 independent observations. Is there advice for the analyst using 100 to 1000 observations (more in the typical range)?

AR: We would advise the analyst to fit a spatial covariance function to the data and calculate stochastic DD reference patterns with that function for the spatial domain of interest. If the time series of the observed data are shorter than the time series length that is required to reach stable DD patterns, the PCA results would "have to be interpreted with the reservation that the DD might be stronger than the comparison with the reference suggests." (L514–518). We provide this information in section 5.1.2.. Here in section 4.1, we do not want to open this topic already. We explain the logic of the order of the sections in our reply to your major comment 3. Please see there and our reply to your comment on L255.

**L265:**
If I'm interpreting the variance percentages correctly, it seems the patterns for PC2 and PC3 may have very closely spaced eigenvalues, not separated sufficiently according to the North et al. criterion. There are a few others too that seems degenerate.

AR: You are right. But we do not want to open this topic in this section already. Please, see our reply to your comment on L255 and your major comment 3.

**L273:**

That may be part of it but the degeneracy of the adjacent eigenvalues may be a major contributor and therefore it needs to be factored into the explanation. Perhaps that would make it more intuitive?

AR: Please, see our reply to your comment on L255 and your major comment 3.

**L283:**

Good but without doing that, the conclusions drawn in this section are subject to unnecessary uncertainty. Perhaps remove the conclusions from the paragraphs above and save them until after the degeneracy analysis. Alternately, run the degeneracy analysis first to clarify the instability in these results.

AR: Please, see our reply to your comment on L255 and your major comment 3.

**L285:**

production of

AR: We will change that.

**L302:**

Are these "PC patterns" the PC loadings? Please define the terminology, then use that same terminology throughout the paper, including figures and tables.

AR: These are the stochastic DD reference patterns calculated as mean correlation loadings. Please see our reply to your major comment 2.

**L318:**

delete "and kind of intuitive"

AR: We will change that.

**L319:**

predictable, a priori,

AR: We will change that.

**L319:**

more

AR: We will change that.

**L321:**

deleted space in front of %

AR: In accordance with the HESS guidelines, we will keep the space in front of the %.

**L323:**

Two comments:

(1) You use heterogenous here (check for a typo, as I think you really want to use heterogeneous) and inhomogeneous in other locations below. If those are the same thing, stick with the exact same word throughout.

AR: Thanks for the typo hint. Yes, we meant that and will change the phrasing consistently to heterogeneous.

(2) Motivate **why** homogeneous versus heterogeneous are both used.  The former would apply to those analyses using **gridded datasets** with a regularly spaced grid, whereas the latter might apply to those analysts using actual irregularly spaced **station data**.

AR: Thanks for the hint. We will add a sentence for motivation in L319:

"...visual recognition is limited. This holds in particular for spatially irregular distributed locations which is the common case in hydrology."

**L328:**
predictable.

AR: We will change that.

**L331:**
as

AR: We will change that.

**L332:**
whereas

AR: We will change that.

**L334:**
I follow the thought but these sentences read awkwardly. Please reword more succinctly.

AR: We will rephrase it to:

"The similarity of patterns formed by congruent selections of cells from the different variants is of particular interest. It addresses the question whether the spatial PC patterns calculated from two different domains result in different relations between the values at locations with coincident coordinates. This is visually only poorly assessible. Therefore, we correlated the patterns of the subsampling variants with the patterns formed by ..."

**L336:**
**Important.** As stated earlier, PC loading (and that of EOFs) interpretation depends on the magnitude of the PC loadings. Correlations subtract each vector mean (pattern mean), so two patterns with different means can have their large correlations, yet their magnitude patterns will be much different and the grid boxes (I think what you refer to as cells) with the maximum PC loadings will be in different geographical (or topological) locations in your domains.  If that is the case, the the correlation is suboptimal for such comparisons. Find a better metric. I suggested the congruence coefficient, though others exist that preserve the vector magnitudes.

AR: Please see our reply to your major comment 4.

**L341:**
Three comments:

(1) Perhaps, but if those "lower ranked" PCs are associated with closely spaced eigenvalues, we have no idea if the lower correlations arise from less DD or more degenerate patterns.

(2) If the results arise from PC loadings associated with closely spaced eigenvalues, Richman (1986) and Jolliffe (1989) have shown that those can be rotated to remove the degeneracy, and then the DD measured. This is mentioned near the end of the paper but it should be mentioned here too.

(3) In fact all the PC loadings could be rotated and then the DDs assessed as rotation has been shown to remove DD (at least beyond how the domain shape affects the correlation structure itself). Most people who interpret the PC loadings first rotated their PCs, so the lack of inclusion of rotated PCs is a shortcoming of the present manuscript.

AR: To (1): We agree.

To (2): At this point in the manuscript, we prefer to stay with unrotated PCs and discuss rotation later in its own subsection as part of the section how to consider DD.

To (3): We understand that you like to highlight the possibility of rotation. However, our focus here is to provide an introduction to DD and its side effects for PCA users in hydrology. For this, we believe that it makes sense to focus on unrotated PCs. Because, to our knowledge, many of the EOF / PCA techniques, and their characteristics and issues, are much less common in the hydrological literature than in the atmospheric sciences literature. We argue that your claim "most people who interpret the PC loadings first rotated their PCs" holds maybe in the atmospheric sciences literature, but not in the hydrological literature. For example, all the hydrological case studies we listed in our introduction were performed with unrotated PCs. We ourselves don´t have noteworthy practical experience with performing and interpreting rotational PCA. In fact, we debated whether to mention PC rotation at all during manuscript preparation, because we felt that the manuscript developed further away from the background and practical experience of the audience we like to address. However, we are aware that the atmospheric sciences literature is rich in this regard and that there is a lot to discover. Therefore, we came up with the compromise to restrict the examples in our introduction to unrotated PCs, but mention rotation as one possibility to continue from there on (section 5.2.2). We will gladly expand the section on rotation with your literature recommendations and some of your thoughts and hints.

We addressed this also in our reply to your major comments 3 and 5.

**L348:**
Will this be a footnote?

AR: A footnote would be linked to Figure 10. We are not sure whether this would work well. So, we like to leave it as it is.

**L364:**

Replace correlations with some metric that incorporates magnitude (e.g., congruence coefficient).

AR: Please see our reply to your major comment 4.

**L383:**

delete "it is simple"

AR: We will change that.

**L402:**

This makes sense because you are using unrotated PCs. Unless the autocorrelation is constructed to coincide with one of those PCs, there is little hope of isolating noise on any one PC. The situation may be less problematic if the PCs are rotated. That is one solution to the problem highlighted in this work.

AR: We will address that in section 5.2.2.

**L403:**

be salient against

AR: We will change that.

**L408:**

Two comments:

(1) This question of retrieval of the correct features of interest has been addressed in at least two published articles: Compagnucci and Richman (2008) and Huth and Beranova (2021).

Compagnucci, R. H., and M. B. Richman, 2008: Can principal component analysis provide atmospheric circulation or teleconnection patterns? Int. J. Climatol., 28, 703–726, https://doi.org/10.1002/joc.1574.

Huth, R. & Beranová, R. (2021). How to recognize a true mode of atmospheric circulation variability. Earth and Space Science, 8, e2020EA001275. https://doi.org/10.1029/2020EA001275

(2) You can add "maximal variance" to this list. It is clear that the first PC (associated with the largest eigenvalue) often merges several unique sources of variability when the spatial scale of the variability is smaller than the domain scale. This was mentioned in a previous comment in the work of Karl and Koscielny (in their Appendix), a citation in your list.

AR: Thank you for your comment and the literature. We will include both. Please see our reply to your major comment 1 and 6 and L81.

**L419:**

This section might be more logical to move this section to closer to the beginning of the paper because the intermixing of the unrotated PC loading signals needs to be addressed/accounted for before one can assess domain dependence.

Personally, I think both are issues but if the paper might apportion the percent of distortion associated with each of degeneracy and domain dependence (in percentage of distortion for example), the utility of this work would be enhanced.

AR: Regarding the order of the sections, please see our reply to your comments on L255-283 and to your major comment 3.

Regarding the apportion of the percent of distortion associated with degeneracy and domain dependence, we agree that this would be interesting for future work. Here, it is beyond the scope of our manuscript.

**L420:**
If it is not well separated by the "consecutive eigenvalues" in PCA.

AR: We are not sure what you mean here. Do you want some rephrasing?

**L428:**
It is surprising you say this given in the last paragraph you said, "For example, for physical processes or modes of geosystems, the S-mode PC properties orthogonality of spatial patterns and linear uncorrelatedness of temporal patterns are heavy constraints (Buell, 1979; Jolliffe, 2002; von Storch and Zwiers, 2003; Hannachi et al., 2007; Monahan et al., 2009)."

That said, it is possible but requires that the PC pattern modes must be assessed for their veracity (validity) by determining if those modes are similar to the patterns embedded in the correlation/covariance matrix from which the PCs were drawn.

AR: We will move the paragraph you mention to a new paragraph after L73 in the introduction where we will include your other comments regarding the physical interpretation of the PCs. Your second comment here will be addressed in the extension of section 5.2.2..

See also our reply to your major comment 6.

**L435:**
In situations where (1) domain dependence exceeds the physical signal and (2) the leading modes have closely spaced eigenvalues, the situation becomes intermixed. It would be ideal to unmix those sources in this work. At the least, issue a caveat.

AR: We are not working with physical signals here. The simulated data sets are designed to produce DD patterns and multiplets for demonstration. Adding new numerical experiments to perform hydrological signal identification is beyond our introductory scope. In the revised manuscript, we will issue a caveat in the extended discussion on rotation and physical signal identification. Please see also our reply to your major comment 5 and 6.

**L439:**
small variations of what?

AR: "small variations in the analysed data"

**L440:**
delete "d" in "degenerated pair"

AR: We will keep "degenerated pair" instead of the suggested "degenerate pair".

**L449:**

Please note here and below my previous criticisms of using correlations to compare PC loading vectors. Often correlating two PC loading vectors results in high correlation for magnitude configurations that don't match well. This occurs because correlations measure only the gradients and ignore the magnitudes, yet the interpretation of the PC loading vectors depends heavily on the locations of the maximum magnitude loadings.

AR: Please see our reply to your major comment 4.

**L456:**

How many patterns represent signal in applied research? There is a point in the eigenvalue spectrum where the associated eigenvectors represent either noise or signal with less variance than noise variance, and such eigenvectors would never be analyzed. This is why analysts nearly always truncate their n eigenvectors at k<<n.

AR: We agree. We show the 10 PC patterns for the introductory purpose of our work here, because we aim to demonstrate the DD phenomenon and its side effects. Please see also our reply to your major comment 3.

**L465:**

Weigh this finding against the average length of data analyzed in applied geophysical research. What is the advice to that analyst with say 100 or 500 independent observations?

AR: We would advise the analyst to consider subsampling a less symmetric domain (L459–462). We will add this suggestion in section 5.2.1. and rewrite L543–548:

"Analysing a subsampled data set with enlarged minimal distance between the locations can be used to diminish the DD of the PCA results. Reducing the symmetry of the domain can remove effective multiplets. Both can help to carve out features other than DD. On the other hand, informative local details might be filtered out together with the excluded locations. If there is still DD, the new DD patterns of the subsampled data set might be harder to recognize visually because of the smaller number of locations per area. The selected minimal distance, respectively the selection of locations, is critical for the analysis. Depending on the choice, different features in the results might stick out, get diminished or even disappear."

**L465:**

If the eigenvalues are exceedingly close on two adjacent eigenvectors, no sample size is sufficient for those eigenvectors to resolve the degeneracy - something that should be mentioned (e.g., see Richman, 1986, his Table II, where 10,000 observations were insufficient to resolve the true patterns once intermixed by degeneracy).

AR: We will include a sentence on that aspect at the end of L465 and add a paragraph break after the new sentence:

"However, for very symmetric domains no sample size might be sufficient to resolve the degeneracy (see Richman, 1986 and PC 2+3 and 5–10 of the square domain in Figure 4a)."

**L482:**
This seems to be part of the conclusions. If so, fold it into a larger conclusions and suggestions section.

AR: It is the suggestion section. Based on a suggestion of referee 2, it will be renamed to "Approaches to consider DD".

**L484:**
See previous comment on Q-mode, where the PC loadings are mapped spatially.

AR: We did not investigate that and will restrict us here to S-mode. See also our reply to your comment on L155.

**L490:**
This is a strange section. Yes, T-mode PCA is possible and even used by some (e.g., see previous citation to Huth and Beranova, 2021) but the idea of applying T-mode is made and never examined in the paper to determine if there is domain dependence in T-mode or how that would manifest. If you say this, then provide evidence that domain dependence may or may not be an issue in T-mode (or other modes). Assuming this is not added to a revision, at best much of this section can be reduced and placed into the conclusions under a paragraph on future research ideas.

AR: We see your point. In the revised version we plan to shorten it substantially.

**L496:**
rarely applied

AR: We will change that.

**L500:**
To

AR: We will keep our phrasing.

**L500:**
deleted "is to"

AR: We will keep our phrasing.

**L501:**
Such a comparison

AR: We will change that.

**L502:**
deleted "to perform"

AR: We will change that.

**L554:**
; Huth and Beranova, 2021

AR: We will add this.

**L556:**

This is more complicated than stated here. In the EOF model, the eigenvectors (EOFs) are orthogonal by column and by row. Under the PC model, the PC loadings are only orthogonal by column (not by row). Once rotated, both EOFs and PC loadings are no longer orthogonal by column. However, in the EOF model, the PCs are uncorrelated. Under the PC model, the PC scores are uncorrelated. Under orthogonal rotation, the PC scores are uncorrelated (and hence orthogonal by column). Under oblique rotation, the PC scores are correlated by column (and hence not orthogonal by column). This is all interesting mathematically, but neither the atmosphere or the hydrologic system follow anything remotely close to orthogonality. Therefore, this is a **validity** issue. If the PC patterns are not valid to represent the physical processes, all the mathematical niceties are meaningless if the PC loadings are to be analyzed individually. PCA is simply the incorrect model to represent the physics on each vector. If you are willing to forego physical analysis of each PC loading vector, then the PCA is an efficient linear representation of the total space but, in that case, domain dependence is not important. Once an analyst wants to add physical interpretation of each PC loading vector, all those maximal variance, and orthogonality features become useless in most cases, but now accounting for patterns with domain dependence becomes important. Again this topics of extracting the known sources of variability and of true modes are addressed in Compagnucci and Richman (2008) and Huth and Beranova (2021), among others.

AR: Thank you for your elaboration. For our scope here, we like to keep the simple more general statement regarding the side effects of rotation and add Wilks (2006) as second source. But we will extend the discussion on rotation in section 5.2.2. and physical interpretation, including the literature you suggested. See also our reply to major comment 5 and 6.

**L556:**
are relaxed

AR: We will keep our phrasing.

**L557:**

Perhaps that "redistributed variance" is the variance of the true modes of variability? Huth and Beranova would support such an interpretation. If that is the case, it is a more important aspect of the physical system than the eigenvalues (or the percent variance associated with each eigenvalue). One way to assess this is (assuming the correlations capture the physically meaning variations in the data) to determine if the PC loadings from a solution (unrotated, rotated) represent the underlying correlation functions. Once that is assessed, and if a PC loading pattern is associated with a correlation/covariance patternr, the statistics associated with these patterns are what describes the physical system.

AR: We will include these aspects in the extension of section 5.2.2.. See also our reply to major comment 5 and 6.

**L558:**

If you examine the results in this paper, for unrotated PCA, selection of k PCs to avoid truncation of degenerate multiplets is also critical, so the criticism of truncating PCs holds in general (unrotated PCA, rotated PCA) for all cases where the n PCs are not retained. Normally, k<<n PC are retained in unrotated solutions too, so selecting k is still an issue for unrotated PCA. Your results suggest that unless k is selected at a location in the ordered eigenvalues at a

location where the eigenvalue spacing exceeds the North criterion, too little eigenvalue spacing confounds the assessment of domain dependence.

If one rotates their PC loadings, previous research suggests that all the domain dependence seems to disappear. If that is the case, you could rotate and check your analyses for domain dependence and report on its reduction under rotation in this manuscript. You could also check the amount of the correlation functions applied that emerge with both the unrotated and rotated PC loadings and report those values and on the differences found.

AR: We agree that defining the truncation point is always an issue, be it for unrotated or rotated PCA. We will include the aspect of eigenvalue degeneracy at the truncation point. Therefore, we will expand the warning to split multiplets in L425 with an explicit statement about the truncation point. Please see our reply to your major comment 3.

Regarding rotation, we will expand section 5.2.2.. Please see our reply to major comment 5.

**L564:**
If you claim this, then can you point to a table of results that partitions the variance into correlation structure variance versus domain dependence variance)?

AR: We cannot point to such partition table. We refer to what we showed here, that is sequences of pure DD patterns in which the leading PCs were associated with substantial amounts of variance.

**L569:**
Should you add the following corollary? *"Without knowledge about the effects of degenerate multiplets, DD can be misinterpreted"*

AR: We assume, you are referring to the intermixing that can mask the expected DD patterns. Thus, we would rephrase your suggestion and add it to the list:

"Without knowledge about the effects of degenerate multiplets, DD can be overlooked because the degeneracy can mask the expected DD patterns."

**L573:**
**Regardless of the assessment of effective multiplets (including at the truncation point) and DD, not analyzing how well the PCs resemble the underlying covariance or correlation structure will often lead to the wrong hyrological interpretations.**

AR: To specific comments on L573–575.

We agree and understand that this is an important point for you. We think so as well. We will add a new paragraph at the very end of the conclusion. It seems to us, that the terminology "underlying covariance or correlation structure" points to atmospheric mode identification. In hydrology, mode identification is not as common as in the atmospheric sciences. Therefore, we will phrase our statement differently.

"However, it has to be noted that passing the check for DD and accounting for effective multiplets in the selection of the PCs are necessary but not sufficient conditions to assure physical meaningfulness. When single PCs, or combination of PCs, are assigned to distinct hydrological features, it should be carefully checked whether the S-mode PCA constraints

orthogonality of spatial patterns and linear uncorrelatedness of temporal patterns support such interpretation. Building on this study, a next research task could be a numerical experiment to evaluate which PCA variants (unrotated vs. rotated, S-Mode versus T-Mode) and which matching coefficients to compare the spatial PC patterns (Pearson correlation vs. congruence coefficient) work best for hydrological feature identification."

**L574:**
PCs are a fine method for data reduction or compact orthogonal description of data onto k PCs. Once the analyst jumps from such a well-accepted interpretation to analyzing or interpreting each individual PC, some assessment of how well each PC represents the data covariability must be performed. Even in cases with no degenerate multiplets and small DD, that does not guarantee (even hint at) an accurate portrayal of a physical process on an individual PC. Such a determination must be made after the analysis. This needs to be added to the conclusions to inform the reader that physically analyzing individual unrotated PCs is a suggested path for enlightenment about the physical system. Recall, the cautionary statment in the conclusion of Buell (1979): "Otherwise, such interpretations may well be on a scientific level with the observations of children who see castles in the clouds."

Sadly, Buell's comment holds for unrotated PCs in general, because of all those "heavy constraints", eigenvalue degeneracy and domain dependence. Again, I urge you to examine Compagnucci and Richman (2008) and Huth and Beranova (2021).

AR: Thank you for your elaboration and the references. We will include this. Please see our reply to your comments to L573 and your major comments 5 and 6.

**L575:**
Perhaps necessary but certain not sufficient to show physical meaningfulness.

AR: We will add this. Please see our reply to your comments to L573 and 574 and your major comments 5 and 6.

**L598:**
What does "clean structure" mean?

AR: We will rephrase it to "smooth pattern".

---

## Author Comment (AC2)

**Reply to referee 2**

Comments of referee 2 are in black.

Replies of the authors (AR) are in blue.
* * *
**Comments of referee 2**

https://hess.copernicus.org/preprints/hess-2024-172#RC2

This paper highlights a largely overlooked issue called domain dependence (DD), where the PCA results are influenced more by the size and shape of the spatial domain being analyzed than by the actual hydrological processes. This effect, caused by spatial autocorrelation in hydrological data, can lead to misleading patterns, accumulation of variance in leading PCs, and closely related (degenerate) PCs that are difficult to distinguish. The paper emphasizes the need to account for DD when interpreting PCA results and introduces two methods—stochastic and analytic—for generating DD reference patterns. These methods are demonstrated using synthetic examples, and R-scripts are provided to help users explore and address DD in their analyses. The results presented are solid. The paper covers all the aspects that are important for a user. However, there are redundancy and a lack of clarity in some of the sections. I suggest a major revision that's focused on organizing and presenting the materials. Please see my detailed comments below.

AR: Thank you for the clear and comprehensive summary of our work. Thank you furthermore for your helpful and motivating comments. We appreciate the work you have spent on the review.

**Major comments:**

It is good to have all the relevant terms explained in Section 2. However, as a hydrologist, I personally found the section 2 quite challenging to follow. Since the objective of this technical note is to raise attention to the DD effects among PCA users in the hydrology community, it is better to use terminologies and displayable items accessible/understandable to hydrologists especially in the method section.

I suggest adding 1) equations when necessary and 2) conceptual diagrams like hypothetical spatial and temporal PC graphs to explain PCA and S-mode PCA (they can be put in the appendix). The authors can also add workflow diagrams in both the method and discussion sections when they illustrate to practitioners how to consider DD, how to diminish DD, etc. Also, consider adding a real hydrological case at the end of the paper to illustrate the DD effects and how to deal with DD. That way, the value of the paper to hydrologists and other PCA users can be greatly improved.

AR: Thank you for your suggestions. In the revised manuscript, we will provide more equations in the main text, e.g. the equation for the calculation of the correlation loadings in section 2.1.2,

$$c_j = a_j * \sqrt{\lambda_j}$$

( 1 )

and in additional schemes. We will add the following schemes:

- a conceptual diagram for S-mode PCA (first draft in Figure 1) and
- workflow diagrams in the method section for (a) the stochastic method and (b) for the analytic method (first drafts in Figure 2 and Figure 3).

[Figure]

Figure 1: S-mode PCA, adapted after Fig. 9 in Richman (1986). n: number of locations, m: number of time steps. The eigenvalues define the explained variance, the loadings the unscaled spatial PC patterns and the scores the temporal PC patterns.

[Figure]

Figure 2: Stochastic DD reference method. n: number of locations, m: number of time steps, N: number of data sets, respectively PCAs, index j: PC rank, c: correlation loadings, a: loadings, λ: eigenvalue, S: stability, indices k, l: running indices for PCAs from the ensemble, $\tilde{c}$: harmonized correlation loadings, eVar: explained variance.

[Figure]

Figure 3: Analytic DD reference method.

We see your point with adding a real hydrological case at the end of the paper. However, for a number of reasons we like to refrain from it. The focus of our work is to illustrate the functioning of DD and its side effects. For this, we believe that it is best to use synthetic examples only. It (1) ensures clearly defined statistical properties, (2) clarifies that "all observed effects are solely caused by the specified statistical properties" and (3) enables "to study the effects of specific properties, e.g. spatial correlation length or spatial extent, on the PCA results." These points are already mentioned in the second last paragraph of the introduction (lines 106–110). For further clarification, we want to add a short phrase in line 106:

"To focus on the functioning of DD and its side effects, we illustrate our introduction with synthetic examples only. This ensures ..."

Furthermore, the manuscript is already quite extensive and we prefer not to extend it further. An analysis of a real hydrological data case would be material for another manuscript. As a matter of fact, the current manuscript evolved out of the work on a manuscript with spatially distributed groundwater level and precipitation series. In that manuscript handling DD is just one aspect. Given the lack of knowledge about DD in the hydrological literature, we tried first

to include an introduction to DD on top of the other analysis. We realized very quickly that both together is too much and that we have material for two standalone manuscripts. We decided to provide first an introduction to DD for the hydrological community - including all the different aspects that we consider important and that we discussed in the presented manuscript here - before presenting the application to real hydrological data cases.

**Minor comments:**

Combine data set to be one word "dataset".

AR: We will do so.

Avoid using the word "system" which is too broad a term and could mean different things to different people. Be more specific. If you are talking about a catchment, use catchment. If you are talking about a soil column, use soil column.

AR: Here we are using the broad term "system" on purpose because we are presenting the functioning of DD irrespective of the analysed system, be it a catchment or a soil column. We will clarify what we mean by "hydrological system" by adding the following sentence to line 37 in the very beginning of the introduction:

"The approach can be applied to data from very different hydrological systems such as catchments or soil columns."

Abstract: The abstract needs reworking. Currently, the authors spend three quarters of the abstract on describing what DD is and why it's important to consider DD. Only 3-4 sentences are focused on what the paper does. The abstract needs to be re-organized such that the first quarter gives the introduction and background information about DD. The middle two quarters focus on the methodology and results. The last few sentences focus on the implications of the findings.

AR: We will re-organize the abstract trying to follow your suggestions. However, we must consider that the manuscript is not a classic research paper. A central aim of this work is to provide HESS readers an introduction to DD and effectively degenerate multiplets. We will therefore need more space than a quarter of the abstract to introduce these concepts and warn of the resulting pitfalls for hydrological interpretations.

Line 45-50: Could expand the list by adding references of PCA/EOF to hydro-climate research like:

Li et al. (2023): https://link.springer.com/article/10.1007/s00382-021-06017-y

Bieri et al. (2021): https://journals.ametsoc.org/view/journals/hydr/22/3/JHM-D-20-0116.1.xml

AR: Thank you. We will do so.

Line 105: You've defined domain dependence to be DD. Use DD here.

AR: We will do so.

Line 118: "Considering DD is discussed". I don't quite understand. Do the authors mean "in practical, when and how to consider DD is discussed"? Be a bit more specific here.

AR: In this section we want to provide the reader with options how to check for DD and how to deal with it. For clarification, we like to change the quoted sentence in L118 to:

"Finally, different options to consider DD are discussed with respect to detecting DD and diminishing DD."

Move section 3 to data and code availability statement.

AR: We will do so.

Figures 5-6: Show the colorbars for the color shadings.

Figure 7 is just a repeat of the square experiments in Figures 5 and 6. I suggest showing one figure of square experiments, one figure of rectangle experiments, and one figure of triangle experiments. On all the PCs, show the colorbar, the information you showed in the title of Figure 7a.

AR: The overview plots in figures 5, 6, 8, 9 and Figure S5 in the supplement are meant for direct visual comparison (1) with the "classical Buell patterns" shown in Figure 1, and (2) among each other. Therefore, we always provide the same structure, with PC 1–10 as columns and the domain boundaries (a) square, (b) rectangular, (c) triangular in the rows. The focus is here on the spatial patterns only - not their magnitudes. Like in Buell´s original work, we therefore don´t show the scales. This way the overview character of the figure is ensured and the plots can be conveniently compared. In contrast to Buell, we use colour gradients - instead of +/- schemes - to picture the spatial patterns (see caption of Figure 5). We think that this further improves the readability of the figures, especially for the more fine structured patterns of the PCs with small eigenvalues (lower ranked PCs).

The detail plots of Figures 7, 13 and Figure S7 in the supplement are meant as examples to demonstrate what magnitudes of (1) contrasts in the spatial PC patterns and (2) explained variance associated with the PCs can result from DD alone. We think that it is important and informative to show this level of detail once in the presentation of the stochastic DD patterns (Figure 7) and in the discussion of the effective multiplets (Fig 13). In case of the overview plots we think this level of detail would be distracting for the readers. In this logic, Figure S7 in the Supplement is merely the anisotropic counterpart for Figure 7 in the main text, meant to complete the anisotropic set of figures S5–7.

To clarify the different purposes of overview and detail plots, we like to separate their introduction in section 4.1. into separate paragraphs with paragraph breaks in lines 260 and 264.

The section titles can be more informative. Like "4.1 First examples, 4.2 Domain shape, 5 Considering DD"… The authors should use short phrases instead of words for the subheaders. This is a good opportunity to provide more information to summarize the subsections.

AR: Thanks for your suggestion. We will change the section titles of section 4 and 5 to:

4. Exploring the DD effect
4.1. Exploring Buell patterns and their stability

Table 1: When the PC of the subsampled variant does not correlate the best with the all-cell PC of the same rank, i.e., the values with "\", the correlation is significantly lower. For example, 0.52 for PC4 in Square patter, 0.45 for PC5 in Square, 0.52 for PC6 in Rectangle. They are significantly lower than other values in the table. Is there an explanation for that?

AR: Best correlating PCs with different ranks do not always exhibit rather low correlation. What we can see in Table 1 are different levels of variation of the patterns from the homogeneous subsampling variant when compared with the patterns from the all cells variant (the classical Buell patterns). The patterns of the subsampling variant can be:

1)   simply noisy variants of the all cells patterns (e.g. PC 1 and 2 from all domains),
2)   simply noisy variants of the all cells patterns but with different ranking (e.g. PC 3 and 4 from the rectangular domains),
3)   a mix of all cells patterns (e.g. PC 4 and 5 from the square domains[1]), or
4)   very different from the all cells patterns (e.g. PC 10 from all domains[2]).

In this sequence of increasing differences between the patterns from both variants, the last example you were addressing (PC 6 from the rectangular pattern) would be placed somewhere in between 3) and 4). Generally, the differences increase towards the low ranked PCs with the more detailed patterns. But, there are also substantial differences between the patterns from relatively high ranked PCs possible (e.g. PC 4 and 5 from the square domains). Thus, even for rather homogeneous subsampling, the DD patterns are not necessarily simply noisy variants of the classical Buell patterns.

This underlines the main message of section 4.2: " For data sets with identical spatial correlation properties and similar domain size, the DD patterns are original for every domain shape." (L316).
* * *
[1] In the all cells variant, PC 4 exhibits two maxima in the upper left and lower right corner and two minima in the lower left and upper right corner, PC 5 exhibits the maximum in the center and four minima in the four corners. In the subsampling variant, PC 4 exhibits two maxima in the upper left and lower right corner and the minimum in the center, PC 5 exhibits basically the same structure but rotated by 90°.

[2] For PC 10, the patterns of the all cells variant are for all domains already so fine structured that the subsampling results in quite different patterns.

We suggest to integrate the above details, including the footnotes, and extend the paragraph in line 340-347 to:

"For the spatial patterns of the homogeneous subsampling variant and the all cells variant, the correlation analysis confirmed the visual impression of overall similarity (Table 1). But it also showed that there are differences. The patterns of the subsampling variant can be:

1) simply noisy variants of the all cells patterns (e.g. PC 1 and 2 from all domains),
2) simply noisy variants of the all cells patterns but with different ranking (e.g. PC 3 and 4 from the rectangular domains),
3) a mix of all cells patterns (e.g. PC 4 and 5 from the square domains[1]), or
4) very different from the all cells patterns (e.g. PC 10 from all domains[2]).

Transitions between 3) and 4) are possible (e.g. PC 6 and 7 of the rectangular domain). Generally, the differences increase towards the low ranked PCs with the more detailed patterns. But, there are also substantial differences between the patterns from relatively high ranked PCs possible (e.g. PC 4 and 5 from the square domains). Thus, even for rather homogeneous subsampling, the DD patterns are not necessarily simply noisy variants of the classical Buell patterns. The comparison with the heterogeneous variant yielded substantially stronger deviations (Table 2). Thus, generally, visual recognition of Buell like patterns in S-mode PCA results is a concrete indication for DD. However, it is so in particular for the leading PC patterns from domains with rather homogeneous spatial arrangement of locations within boundaries similar to Buell´s archetypes. Even for domains of similar size and identical spatial correlation properties, deviations from strictly regular distribution of locations alone can result in DD patterns substantially deviating from what one might expect with the classical Buell patterns in mind."

It is unclear to me how exactly did you calculate stability. Suggest showing equation when it is first mentioned to illustrate.

AR: The stability $S_j$ of the spatial patterns of PC rank j is calculated as the mean $R^2$ of the pairwise correlations of all spatial patterns with PC rank j from the PCA ensemble:

$$S_j = \frac{1}{N * (N - 1) * 2} \sum_{k<l}^{N} cor(c_{jk}, c_{jl})^2$$

N is the number of PCAs which equals the number of analysed data sets, from the ensemble; k and l are the running indices of the PCAs that are compared.

We will add the above equation and a scheme to the description of step 2 (lines 195–197). A first draft of the scheme is given in Figure 2. The abbreviations from the scheme will be introduced in the main text as well.

---

## Editor Decision (ED1)

[revised manuscript text omitted]

Ignorance about DD can easily lead to wrong interpretations of PCA results. DD patterns are distinct, with strong gradients and contrasts, and therefore highly suggestive to indicate physically meaningful drivers or properties of the analysed system. In the climatological literature DD was intensely discussed (Buell, 1975, 1979; Horel, 1981; Richman, 1986, 1987, 1993; Jolliffe, 1987; Legates, 1991, 1993). Apparently, the topic did not reach the hydrological community, even though the effect of size and shape of the network geometry on the results was observed in early hydrological S-mode PCA applications (Smirnov, 1973; Bartlein, 1982; Lins, 1985b). For that reason, we want to raise attention to the DD effect among PCA users in the hydrological community again to reduce the risk of drawing wrong hydrological conclusions from spatio-temporal PCA. DD is one aspect in the general discussion on the physical interpretation of S-mode PCs. There are strongly diverging opinions, ranging from "never physically interpret any PCs" to "distinct processes can be meaningfully assigned to single PCs". For physical processes or modes of geosystems, the S-mode PC properties orthogonality of spatial patterns, linear uncorrelatedness of temporal patterns and successive maximization of variance are heavy constraints (Buell, 1979; Jolliffe, 2002; von Storch and Zwiers, 2003; Hannachi et al., 2007; Monahan et al., 2009). By extracting maximal variance, different sources of variability can get pulled onto the first eigenvector, thereby mixing the sources (e.g. Figure 14A in Karl and Koscielny, 1982). The successive order of the PCs implies that they should not be interpreted isolated, but only with reference to the preceding PCs. The spatio-temporal patterns of the first PC set the reference for all subsequent PC patterns. Forced by the orthogonality constraint, prominent features of the first spatial PC pattern cascade down to the spatial patterns of the other PCs (Cahalan et al., 1996). The analysis is limited to linear relationships and assumes stationarity of mean and variance of the analysed variable. If single features are assigned to single PCs, this raises the question whether the hydrological features in the analysed system are expected to exhibit orthogonal spatial patterns, to be linearly uncorrelated in time and to successively maximize variance. If not, PCA is simply the wrong model (Jolliffe 1987; 2002).

Rotation of PCs can relax the aforementioned PCA constraints (Richman, 1986; Hannachi et al., 2007; Monahan et al., 2009). It is regularly used in atmospheric mode detection. Several studies found that rotated PCA performed better than unrotated PCA for this purpose, and that their spatial patterns were less prone to DD (Richman, 1986; Compagnucci and Richman, 2006; Huth and Beranova, 2021). Despite these findings, unrotated PCA is still often used (Huth and Beranova, 2021). Regardless of whether rotated or unrotated PCA is used, the physical interpretation depends on the spatial PC patterns and requires that they are not domain dependent. The knowledge which locations carry the most variance can already be helpful to improve the physical understanding of the analysed system (Monahan et al., 2009). In hydrology, unrotated PCA is to our knowledge much more common than rotated PCA. Therefore, we mainly focus on unrotated PCA here.

[revised manuscript text omitted]

**2 Data**

**2.1. Synthetic data**

The synthetic data sets consist of synchronous spatially distributed time series exhibiting spatial but no temporal autocorrelation. Each data set is produced by concatenating realizations of a random field with identical spatial correlation properties (Figure 2). The grid cells (cells) of the random field represent the locations of a data set. The spatial autocorrelation is defined with a spatial covariance model. Each realization of the field represents one instant of time of a data set. Thus, at each location the respective time series consists of a sequence of random numbers. The number of field realizations gives the length of the simulated time series. The random fields were simulated with the "RandomFields" package (Schlather et al. 2015, 2020).

Figure 2 Three realizations of a  $20 \times 20$  random field simulated with an isotropic exponential covariance model and spatial correlation length of 10 cells representing three instants of time of a synthetic data set.

**2.2. Precipitation data**

As an application example based on observed data we use time series of monthly precipitation sums from the years 1991-2020 out of a  $200 \text{ km} \times 200 \text{ km}$  square in northeast Germany (Figure 3). The precipitation series were selected from the  $1 \text{ km} \times 1 \text{ km}$  HYRAS-DE-PR precipitation grid provided by the German Weather Service (Deutscher Wetterdienst, 2025). Amongst others, the HYRAS-DE-PR precipitation product is suggested as input data for hydrological modeling (see the description file at Deutscher Wetterdienst (2025)). The monthly precipitation sums are based on daily measurements of precipitation height at the monitoring stations. The raster layers are interpolated by combining multiple linear regression considering topography with inverse distance weighting. The interpolation method preserves the measured precipitation values at the grid cells of the stations. For details, see Rauthe et al. (2013) and the description file of the data (Deutscher Wetterdienst, 2025). Except from z-scaling, no pre-processing of the precipitation series was applied.

Figure 3 Maps of the precipitation data showing the permanent precipitation stations (crosses) that were used by the German Weather Service to produce raster of monthly precipitation sums in the monitoring period 1991–2020. Left panel: Federal States of Germany (black lines) and the domain selected for PCA (red square). Right panel: Sample raster from the first two months of the selected data set. The maps are in ETRS89 / LAEA Europe projection.

**3 Methods**

**3.1. Principal Component Analysis**

PCA maps an  $m \times n$  data matrix X to n new linearly uncorrelated variables, the Principal Components (PCs), such that the PCs successively maximise represented fractions of the data set's variance (Wilks, 2006). The data set's variance is defined as the sum of variances of the variables x. It equals the sum of the diagonal elements (trace) of its covariance matrix. PCA can be performed as eigenvalue decomposition of the variables' covariance matrix or as singular value decomposition of the variables' matrix with the variables being centred to their mean (Jolliffe, 2002). Unfortunately, the terminology is not used consistently throughout the literature. Here, we follow the terminology used by Jolliffe (2002) and Jolliffe and Cadima (2016) for the eigenvalue approach.

Each PC is associated with an eigenvalue  $\lambda$  and an eigenvector a. The values of a PC are termed scores. The variance of the scores of a PC equals its eigenvalue. The ratio of a PC eigenvalue to the sum of all PC eigenvalues gives the fraction of the data set's variance assigned to that PC. Each PC is calculated as linear combination of all n analysed variables x (non-locality).

$$pc_j = a_j X = \sum_{i=1}^n a_{ij} x_i \tag{1}$$

The coefficients  $a_{ij}$  in this linear combination are termed loadings. The loadings of a PC j are the n elements of the eigenvector  $a_j$  associated with that PC. The eigenvectors of all PCs define the orthogonal basis of the new ordination system into which the analysed data is projected (orthogonality constraint). Subject to the eigenvectors being orthogonal and the PCs being uncorrelated, the linear combinations of the PCs provide the optimal linear functions to successively maximise variance accounted for (variance maximization). The maximum variance that can be described by a linear combination of the analysed variables is assigned to the first PC, the maximum of the remaining variance to the second PC, and so forth. Thus, the leading

PCs provide a compact description of the data set's variance. It is quite common that a few PCs suffice to summarize a major part of a data set's variance.

For the synthetic data, PCA was performed with the function "prcomp" from the default "stats" package (R Core Team, 2019). For the precipitation data, a truncated PCA, calculating the first 20 PCs only, was performed with the "prcomp\_irlba" from the "irlba" package to reduce computation time. The equivalence of the results of both PCA algorithms with respect to the leading PCs was confirmed by comparison of the results from smaller data sets.

**3.1.1. S-mode PCA**

In S-mode PCA, the analysed variables are synchronous time series distributed in space at multiple locations (Figure S1; Richman, 1986). Thus, the PCs are series of the same length as the analysed time series (temporal PC patterns) and the loadings yield values for each location (spatial PC patterns), describing the weighting of the analysed time series to calculate the PC scores. All temporal PC patterns are linearly uncorrelated with each other, each temporal PC pattern is associated with a spatial pattern and all spatial PC patterns are orthogonal to each other. Note that in this study, we perform S-mode PCA only.

**3.1.2. Correlation matrix based PCA, correlation loadings and contrasts of spatial PC patterns**

Normalizing the variables to zero mean and standard deviation one (z-scaling) prior applying PCA ensures equal weighting of the analysed variables. This is important if the range of values between the analysed variables differs substantially. A PCA with z-scaled variables is identical to an eigenvalue decomposition of the correlation matrix of the analysed variables. In hydrology, correlation matrix based PCA is to our knowledge more common than covariance matrix based PCA.

For the eigenvectors, different scaling conventions exist (Wilks, 2006). Here, the eigenvectors that are used to calculate the PCs are of unit length (Equation 1). In correlation matrix based PCA, normalizing the loadings from the unit length eigenvector  $a_j$  of a PC j by multiplying it with the square root of its eigenvalue  $\lambda_j$  is equivalent to the Pearson correlation between the scores of that PC  $pc_j$  and the analysed variables X.

$$c_i = a_i \sqrt{\lambda_i} = cor(pc_i, X) \tag{2}$$

Thus, the loadings are normalized to the commonly well-known Pearson correlation range from -1 to 1 which simplifies reading and interpretation of the PCA results. Here, we use the term "correlation loadings" for these normalized loadings  $c_j$ . We do so to prevent confusion with the coefficients that are used in the linear combination to calculate the PCs, which are not normalized to a common range (Equation 1). The sum of the squared correlation loadings  $c_j$  of a PC j equals its eigenvalue  $\lambda_j$ . Thus, they can be used to calculate the fractions of variance associated with the PCs. In the following, the spatial PC patterns are described with correlation loadings only.

For S-mode PCA, the normalization enables direct comparison of the contrasts of spatial patterns from different PCs or PCAs.

Here, we define the contrast of a spatial PC pattern as the range between the minimum and maximum of the correlation loading values of that PC. Thus, the maximum contrast possible would be 2.

**3.2. DD reference patterns**

DD reference patterns are the DD patterns of a distinct combination of spatial domain and spatial correlation properties. They can be used as null hypothesis in pairwise statistical tests to test whether spatial PC patterns differ significantly from what has to be expected from DD alone.

**3.2.1. Stochastic method**

In the stochastic method, PCA is applied on synthetic data sets (Section 2.1) to derive DD reference patterns. As the data sets consist of spatially correlated white noise time series, their temporal PC patterns are white noise as well. The spatial PC patterns of the data sets are solely determined by the spatial domain and the spatial correlation properties defined in the simulation. The spatial PC patterns of data sets simulated with identically parameterized random fields differ due to the randomness in the simulations. Therefore, a three-step procedure is applied to get stable patterns (Figure 4).

- Step 1: An ensemble of data sets with identical spatial domain and spatial correlation properties is simulated. Each of the data sets is analysed separately with a PCA, resulting in a PCA ensemble.
- 250 Step 2: The stability of the spatial PC patterns is assessed by pairwise correlating the spatial patterns of all possible combinations of PCs with identical ranks from the PCA ensemble. For each PC rank, the mean R2 of the correlations is used to describe the overall similarity of the respective spatial PC patterns.
  - Step 3: For each PC rank (a) the mean spatial patterns from all PCAs of the ensemble and (b) their standard deviation patterns are calculated. They are calculated as the mean and standard deviation of the correlation loadings of PCs with identical rank from the PCA ensemble.
  - The mean spatial PC patterns are the DD reference patterns for data sets with the spatial domain and the spatial correlation properties defined in step 1. The standard deviation patterns serve as their spatially discrete uncertainty estimation. The variance represented with the DD reference patterns ("explained variance") is estimated with the mean and standard deviation of the explained variances of PCs with identical rank from the ensemble.
- PCs with identical rank from different data sets of an ensemble might exhibit basically the same spatial pattern but with opposite signs due to the randomness of the field simulations, i.e. the pattern of one data set might be basically a negative version of another one. For the calculation of mean and standard deviation of the spatial PC patterns of an ensemble (step 3), the spatial patterns of PCs with identical rank are therefore harmonized such that they all are correlating positively. Thus, the correlation loadings of PCs that are correlating negatively with those of identically ranked PCs from the first data set are multiplied by -1 and therefore reversed.
  - Note that the suggested method requires the use of correlation loadings to describe the spatial PC patterns. Thus, it is restricted to correlation matrix-based S-mode PCA, meaning the analysed series have to be z-scaled (Sections 3.1.1 and 3.1.2). Furthermore, the mean spatial PC patterns are derived from a data set ensemble, not from a distinct single data set. Thus, they cannot be used to calculate PC scores.

Figure 4 Stochastic DD reference method. n: number of locations, m: number of time steps, N: number of data sets, respectively PCAs, index j: PC rank, c: correlation loadings, a: loadings,  $\lambda$ : eigenvalue, S: stability, indices k, l: running indices for PCAs from the ensemble,  $\tilde{c}$ : harmonized correlation loadings, V: explained variance.

**3.2.2. Analytic method**

Another possibility to produce DD reference patterns is to perform a PCA with the "analytic", or "exact", covariance matrix (North et al., 1982; Cahalan et al., 1996; Dommenget, 2007) of a spatially homogeneous covariance function (Figure 5). The analytic covariance matrix consists of the covariances between all of the data set's locations calculated directly with their interpoint distances from the function. For consistency with the stochastic method (Section 3.2.1), the eigenvectors (spatial patterns) were scaled to correlation loadings (Section 3.1.2). A brief review of different variants using the analytic covariance matrix to produce PCA reference patterns is given in Appendix A.

For the synthetic examples, the analytic method was performed as eigendecomposition of the analytic covariance matrix with the function "eigen" from the default "base" package (R Core Team, 2019). For the precipitation data, a truncated PCA, calculating the first 20 PCs only, was performed with the function "eigs\_sym" from the "RSpectra" package to reduce computation time. The equivalence of the results of both algorithms with respect to the leading PCs was confirmed by comparing the results from smaller data sets.

Figure 5 Analytic DD reference method.

**3.3. Matching of spatial PC patterns**

The matching of the spatial patterns from different PCAs was quantified with the congruence coefficient (Lorenzo-Seva and ten Berge, 2006) and Pearson correlation. The congruence coefficient  $\varphi$  is defined as the cosine of the angle between two vectors of component or factor loadings  $a_1$  and  $a_2$ , both being based at the origin.

$$\varphi = \frac{\sum a_1 a_2}{\sqrt{\sum a_1^2 \sum a_2^2}} \tag{3}$$

In contrast, Pearson correlation gives the cosine of the angle between two vectors, both being based at the mean loading. Thus, the matching coefficient r in the following equation gives the Pearson correlation when  $\mathbf{b} = \text{mean}(a_1)$ ,  $\mathbf{d} = \text{mean}(a_2)$  and the congruence coefficient when  $\mathbf{b} = \mathbf{d} = 0$  (see the help of R-function "factor.congruence").

$$r = \frac{\sum (a_1 - b)(a_2 - d)}{\sqrt{\sum (a_1 - b)^2 \sum (a_2 - d)^2}} \tag{4}$$

If the compared vectors have zero mean values (mean( $a_1$ ) = mean( $a_2$ ) = 0), both indices are identical. In all other cases, the results differ. The congruence coefficient is sensitive to the addition of constants, because the vector means are not removed (Lorenzo-Seva and ten Berge, 2006). Two eigenvectors with different means can be closely correlated even though their magnitude patterns differ substantially such that some variables load high on the one PC and low on the other (Richman, 1986; Lorenzo-Seva and ten Berge, 2006). In the S-mode PCA case, this means that two spatial PC patterns with different means can be closely correlated even though some locations load high on the one PC and low on the other (thus, the maximum loadings of the two PCs could be in different locations). If that is the case, the congruence coefficient would be low, indicating the difference in magnitude patterns. Thus, in contrast to Pearson correlation it incorporates vector magnitudes in the comparison (Richman, 1986). This is desirable for the comparison of eigenvectors from PCA or factor analysis because the magnitude of the loadings is important for the interpretation of the components (Richman, 1986). Therefore, the congruence coefficient is recommended as matching coefficient over Pearson correlation for the comparison of eigenvectors from PCA or factor analysis (Richman, 1986). However, a major benefit of Pearson correlation is that it is well known and the results in terms of r or R2 can easily be contextualized by the analyst.

Note that the stability analysis of the stochastic approach (step 2, Section 3.2.1) was performed with Pearson correlation only, because all compared PC patterns (i) were of identical rank and (ii) were based on synthetic data sets simulated with identical spatial correlation properties and identical domains. For this setting, we considered the effect of the pattern mean subtraction by Pearson correlation as negligible.

Both indices have a value range from -1 to 1, with 1 indicating a perfect match, 0 no relationship and -1 a perfect inverse match (Richman, 1986). Compared with Pearson correlation, the congruence coefficient is biased towards higher values (Richman, 1986). Several guidelines were suggested that assign specific ranges of absolute congruence coefficients (aCC) to categories of goodness-of-match, or specific thresholds as indication for the identity of components/factors (Richman, 1986; Lorenzo-Seva and ten Berge, 2006). Here, we follow Lorenzo-Seva and ten Berge (2006) who suggested that aCC values between 0.85 and 0.94 indicate fair similarity of the two components, values larger than 0.95 indicate that they can be considered equal and values below 0.85 should not be interpreted as indication for similar components.

The congruence coefficient was calculated with the function "factor.congruence" from the "psych" package. The statistical significance of the correlations was assessed with t-tests and the significance level 0.05 using the function "cor.test" from the default "stats" package (R Core Team, 2019).

**3.4. North's rule of thumb**

Confidence limits to identify clearly separated eigenvalues and eigenvectors can be estimated e.g. with North's rule of thumb (North et al., 1982; Hannachi et al., 2007) based on the data set's effective sample size  $n^*$ , also known as number of independent observations in the sample or the number of degrees of freedom (Hannachi et al., 2007). The 95 % confidence interval of the eigenvalue  $\lambda_k$  is given by  $\delta\lambda_k \sim \lambda_k \sqrt{2/n^*}$ . In our case here,  $n^*$  equals the length of the analysed time series because the series do not exhibit temporal autocorrelation. The confidence interval for the associated eigenvector  $u_k$  can then be estimated with  $\delta u_k \sim (\delta \lambda_k/\Delta \lambda) u_j$  where  $u_j$  is the eigenvector of  $\lambda_j$ , the closest eigenvalue to  $\lambda_k$ , and  $\Delta\lambda$  the spacing  $(\lambda_j - \lambda_k)$  between both eigenvalues.

**3.5. Varimax rotation**

Rotation aims at separating a subset of PCs more clearly such that the association between the eigenvectors and the PCs is more distinct. The goal is to reach a so called "simple structure" with the loadings being either close to zero or close to the maximum possible absolute values as much as the data permit (Wilks, 2006). Thus, the magnitudes of the loadings are changed. The total variance of the rotated subspace is preserved, but the variance among the rotated PCs is redistributed more evenly (Jolliffe, 2002), potentially affecting which PCs are rated dominant. Different rotation methods are available (Richman, 1986). The rotation is performed by multiplication of the selected eigenvectors by a rotation matrix. If the rotation matrix is orthogonal, the rotation is called orthogonal, otherwise oblique (Wilks, 2006). To support the interpretability of the results, the rotation matrix is chosen to optimize a simplicity criterion (Jolliffe and Cadima, 2016). Depending on the selected simplicity criterion, the rotation changes the properties of the eigenvectors and PCs. The results can depend on the number of eigenvectors that are rotated (Jolliffe, 2002; Wilks, 2006). This is different from standard PCA where the patterns and the associated variances from a set of PCs do not depend on the number of considered PCs. For example, in standard PCA the patterns and variance distributions of the first two PCs are identical, regardless of whether only the first two PCs are considered or, say, the first four PCs. Often the results are affected more by the choice of how many eigenvectors are rotated than by the choice of the simplicity criterion (Hannachi et al., 2006; Jolliffe and Cadima, 2016).

Here, we applied varimax rotation with Kaiser normalization (Kaiser, 1958). It is the most popular rotation method (Wilks, 2006). Varimax is an orthogonal rotation that maximizes the sum of the variances of the squared elements from the r selected eigenvectors b by iteratively rotating pairs of eigenvectors (Richman, 1986; Wilks, 2006). With the Kaiser normalization the eigenvectors b are normalized with the communalities b of the b analysed variables (here the time series from the b different locations) prior rotation and renormalized afterwards. The communality b of variable b is the fraction of variance from the variable that is depicted by the b rotated PCs. The normalized varimax criterion b can be calculated as

$$V = \sum_{j=1}^{r} \left\{ \left[ n \sum_{i=1}^{n} (b_{ij}^{2}/h_{i}^{2})^{2} - \left[ \sum_{i=1}^{n} (b_{ij}^{2}/h_{i}^{2}) \right]^{2} \right] / n^{2} \right\}$$
 (5)

Note that the scaling of the eigenvectors that are rotated affects the varimax results (Jolliffe, 1995; Wilks, 2006). Either the orthogonality of the eigenvectors, the uncorrelatedness of the PCs or both get lost. The most popular scaling and the default in many software packages is to use eigenvectors scaled to the square root of their eigenvalues, derived from correlation matrix PCA (what we term correlation loadings here). In that case, the orthogonality of the eigenvectors and the uncorrelatedness of the PCs are lost. Other options are to use unit length eigenvectors which preserves the orthogonality of the eigenvectors, or to divide the unit length eigenvectors by the square root of their eigenvalues which preserves the uncorrelatedness of the PCs. For the introductory purpose we use the most popular variant and rotate correlation loadings only. Varimax rotation was performed with the function "varimax" from the default "stats" package (R Core Team, 2019).

**4 Exploring the DD effect**

**4.1. Exploring Buell patterns and their stability**

[revised manuscript text omitted]

The stochastic reference script enables the production of catalogues of stability plots and DD patterns like in Figure 7 and Figure S2 for data sets with different spatial domains and spatial correlation properties (for sample catalogues see Lehr (2024)). Both plots in combination can be used to explore how the properties of a data set affect the DD patterns. Here, we neglect the effect of measurement errors. However, it can be simulated by adding noise to the realizations of the random field (Figure 2).

Figure 6 (a) Square, (b) rectangular and (c) triangular domain boundaries on the  $20 \times 20$  grid. The grid cells represent locations from a data set.

Figure 7 Stability of the spatial patterns from the leading ten PCs in relation to the time series length of the simulated data within the (a) square, (b) rectangular and (c) triangular domain boundaries of Figure 6. All cells within the boundaries were used. For each domain the results from 12 data set ensembles are shown. Each ensemble consists of 100 data sets simulated with identical time series length, an isotropic exponential covariance model and a spatial correlation length of 10 cells. Each simulated data set was analysed separately with PCA. Symbols depict the mean R2 of the correlation between the spatial patterns of all PCs with identical rank derived from the respective ensemble. The legends in (c) apply also to (a) and (b) of the respective row.

Figure 8 Overview of the leading ten mean spatial PC patterns (DD reference patterns), estimated with the stochastic method from the data set ensembles with time series length 10 000 shown in Figure 7. Instead of the +/- schemes used by Buell (1975) (Figure 1) we use colour gradients to picture the spatial patterns.

Figure 9 As in Figure 8 but for the standard deviation patterns (uncertainty estimation of the stochastic DD reference patterns). From blue to yellow the colour gradients depict increasing uncertainty.

**4.2. Effects of the domain shape**

For data sets with identical spatial correlation properties and similar domain size, the DD patterns are original for every domain shape. This is obvious for domains of such simple and clearly different shape like the three geometric shapes used so far. The sequence of their DD patterns is visually easy to recognize. For more complex shapes, the DD patterns are less predictable, a priori, and visual recognition is more limited.

For demonstration, we compared the DD patterns from data sets with identical spatial correlation properties in which all cells within the three geometric boundaries of Figure 6 were selected (Figure 8) with two variants in which only 40 % of the cells were randomly selected. In the first variant the subsampling was spatially homogeneous (Figure 10), in the second spatially heterogeneous (Figure 11). The domain of the second variant contained a subregion with higher sampling probability than the rest of the domain, i.e. within each domain there is one area in which the locations cluster. Clusters of locations have more weight in the calculation of the PCs analogue to the calculation of a weighted spatial mean (Karl et al., 1982). For the DD pattern of PC 1 the effect is obvious. Its monopole is placed in the centroid of the network. In comparison with the regular variant (Figure 1, Figure 8 and Figure 10) it is therefore shifted according to the density of the locations (Figure 11). The patterns of all other PCs are not predictable without calculating DD reference patterns.

Visually, the domains of the subsampling variants are still clearly of square, rectangular and triangular shape. Their leading DD patterns are recognizable as distinct spatial patterns. Most of those from the homogeneous subsampling variant (Figure 10) appear as noisy counterparts of the all cells patterns (Figure 8). In the heterogeneous case (Figure 11), the patterns of the square domain appear again relatively similar, whereas for the triangular and rectangular domain only a few PCs exhibit visually similar patterns, e.g. PC 2.

The similarity of patterns formed by congruent selections of cells from the different variants is of particular interest. It addresses the question whether the spatial PC patterns calculated from two different domains result in different relations between the values at locations with coincident coordinates. This is visually only poorly assessable. Therefore, we correlated the patterns of the subsampling variants with the patterns formed by the corresponding subsets from their all cells counterpart (that is, the all cells patterns clipped with the coordinates of the subsampling variant). For example, the patterns from the homogeneously subsampled square (Figure 10a) were correlated with the patterns from the all cells square (Figure 8a) clipped with the coordinates of the subsampled square.

For the spatial patterns of the homogeneous subsampling variant and the all cells variant, the correlation analysis confirmed the visual impression of overall similarity (Table 1). But it also showed that there are differences. The patterns of the subsampling variant can be:

1) simply noisy variants of the all cells patterns (e.g. PC 1 and 2 from all domains),

- 2) simply noisy variants of the all cells patterns but with different ranking (e.g. PC 3 and 4 from the rectangular domains),
- a mix of all cells patterns (e.g. PC 4 and 5 from the square domains 1), or
- very different from the all cells patterns (e.g. PC 10 from all domains2).

Transitions between 3) and 4) are possible (e.g. PC 6 and 7 of the rectangular domain). Generally, the differences increase towards the low ranked PCs with the more detailed patterns. But, there are also substantial differences between the patterns from relatively high ranked PCs possible (e.g. PC 4 and 5 from the square domains). Thus, even for rather homogeneous subsampling, the DD patterns are not necessarily simply noisy variants of the classical Buell patterns. The comparison with the heterogeneous variant yielded substantially stronger deviations (Table 2). Thus, generally, visual recognition of Buell like patterns in S-mode PCA results is a concrete indication for DD. However, it is so in particular for the leading PC patterns from domains with rather homogeneous spatial arrangement of locations within boundaries similar to Buell's archetypes. Even for domains of similar size and identical spatial correlation properties, deviations from strictly regular distribution of locations alone can result in DD patterns substantially deviating from what one might expect with the classical Buell patterns in mind. Side note: The spatial PC patterns of the subsampling variants required shorter time series lengths to stabilize (Figure 12 and Figure S3) than the all cells variant (Figure 7). This indicates that the subsampling resulted in a more unbalanced arrangement of locations and therefore a more distinct orientation for the order of the orthogonal spatial PC patterns.

Figure 10 DD reference patterns as in Figure 8 but for a random selection of only 40 % from the cells within the three geometric domain boundaries of Figure 6. The sampling probability was homogeneous across the domain (spatially homogeneous case).

\_

<sup>1 In the all cells variant, PC 4 exhibits two maxima in the upper left and lower right corner and two minima in the lower left and upper right corner, PC 5 exhibits the maximum in the centre and four minima in the four corners. In the subsampling variant, PC 4 exhibits two maxima in the upper left and lower right corner and the minimum in the centre, PC 5 exhibits basically the same structure but rotated by 90°.

<sup>2 For PC 10, the patterns of the all cells variant are for all domains already so fine structured that the subsampling results in quite different patterns.

Figure 11 DD reference patterns as in Figure 8 but for a random selection of only 40 % from the cells within the three geometric domain boundaries of Figure 6. The sampling probability within the small square in the lower left was three times higher than in the rest of the domain (spatially heterogeneous case).

Figure 12 Stability of the spatial PC patterns as in Figure 7 but for the patterns of the homogeneous subsampling variant (Figure 495 10).

|           | Hom PC         | 1    | 2    | 3       | 4       | 5       | 6       | 7    | 8       | 9        | 10      |
|-----------|----------------|------|------|---------|---------|---------|---------|------|---------|----------|---------|
| Square    | aCC            | 1    | 0.99 | 0.96    | 0.73 \5 | 0.64 \4 | 0.88    | 0.85 | 0.74    | 0.77 \10 | 0.62 \9 |
|           | $\mathbb{R}^2$ | 0.94 | 0.97 | 0.93    | 0.52 \5 | 0.45 \4 | 0.79    | 0.73 | 0.55    | 0.59 \10 | 0.38 \9 |
| Rectangle | aCC            | 1    | 0.98 | 0.90 \4 | 0.88 \3 | 0.90    | 0.72 \7 | 0.56 | 0.66 \9 | 0.64 \8  | 0.48    |
|           | $\mathbb{R}^2$ | 0.87 | 0.97 | 0.80 \4 | 0.78 \3 | 0.83    | 0.52 \7 | 0.32 | 0.43 \9 | 0.40 \8  | 0.23    |
| Triangle  | aCC            | 0.99 | 0.96 | 0.96    | 0.93    | 0.96    | 0.89    | 0.91 | 0.92    | 0.65 \10 | 0.53 \9 |
|           | $\mathbb{R}^2$ | 0.89 | 0.95 | 0.92    | 0.86    | 0.93    | 0.80    | 0.84 | 0.84    | 0.42 \10 | 0.28 \9 |

Table 1 Best matches between the DD patterns of the square, rectangular and triangular domains from the homogeneous subsampling variant (Figure 10) and the patterns formed by the corresponding subsets from their all cells counterpart (that is, the all cells patterns (Figure 8 (a) to (c)) clipped with the coordinates of the subsampling variant (Figure 10 (a) to (c))), quantified by the absolute values of the Congruence Coefficient (aCC) and R2. Mostly, the best matches were of identical PC rank. If the best match was with an all cells pattern subset of different rank, that rank is given after the "\". Hom PC: PC ranks from the homogeneous subsampling variant. Bold aCC values indicate fairly similar PC patterns, grey shaded and bold aCC values PC patterns that can be considered equal (Section 3.3).

|           | Het PC         | 1       | 2       | 3       | 4       | 5       | 6       | 7        | 8        | 9        | 10      |
|-----------|----------------|---------|---------|---------|---------|---------|---------|----------|----------|----------|---------|
| Square    | aCC            | 0.93    | 0.82 \3 | 0.97 \2 | 0.67    | 0.77 \6 | 0.55 \5 | 0.65     | 0.66 \9  | 0.50     | 0.54    |
|           | $\mathbb{R}^2$ | 0.56 \3 | 0.85 \3 | 0.97 \2 | 0.43    | 0.61 \6 | 0.32    | 0.44     | 0.43 \9  | 0.25     | 0.30    |
| Rectangle | aCC            | 0.91    | 0.65    | 0.73 \4 | 0.93 \3 | 0.69 \6 | 0.60 \7 | 0.69 \10 | 0.61     | 0.75     | 0.43 \7 |
|           | $\mathbb{R}^2$ | 0.85 \2 | 0.76    | 0.52 \4 | 0.85 \3 | 0.48 \6 | 0.37 \7 | 0.49 \10 | 0.39     | 0.56     | 0.18 \7 |
| Triangle  | aCC            | 0.98    | 0.80    | 0.61    | 0.71    | 0.93 \6 | 0.53 \8 | 0.64     | 0.62 \10 | 0.39 \10 | 0.70 \9 |
|           | $\mathbb{R}^2$ | 0.73    | 0.64    | 0.43    | 0.49    | 0.86 \6 | 0.29 \8 | 0.42     | 0.39 \10 | 0.15 \10 | 0.49 \9 |

Table 2 As in Table 1 but for the heterogeneous subsampling variant (Figure 11).

**4.3. Effects of the domain size and spatial correlation length**

The ratio between domain size and the spatial correlation length affects the fractions of variance allocated to the PCs (Figure 13) as well as the contrasts of the spatial PC patterns (Figure 14). If there is no spatial correlation (spatial "white noise"), the spatial patterns of all PCs are white noise. All PCs represent the same fraction of variance, one divided by the total number of PCs. The magnitudes of the contrasts of their spatial patterns are small and on the same level. For spatial correlation length increasing from zero towards infinity, the data sets' series from all locations get more and more similar, converging towards identity of all series (perfect correlation). If the latter is reached, there is no variance in the data that could be distributed and, consequently, there are no patterns or contrasts in the PC patterns. In between the two extremes, successive allocation of variance to the PCs and spatial PC patterns with distinct contrasts appear.

For the variance allocation, increasing correlation lengths result in increasing accumulation of variance in the leading PCs, converging towards accumulation of the total variance in PC 1.

For the contrasts, it is more complex. The maximum contrasts appear for correlation lengths in the order of magnitude of the domain size. The exact maximum is specific for the different PCs and depends on the particular domain shape. For example, for the triangular domain here (Figure 14c), the contrasts of the PC 1 patterns peak at a correlation length of 13 cells, the ones of the PC 2 patterns at a correlation length of 21 cells (not shown). The increase of the contrasts between zero correlation length and the correlation lengths of the maximum contrasts reflects the increasing fraction of covarying locations that support the poles of the DD patterns. The decrease of the contrasts between the correlation lengths of the maximum contrasts and infinite correlation length reflects the increasing similarity of all locations which leads to smoother spatial PC patterns with contrasts converging towards zero.

Within a DD sequence, the magnitude of the contrasts differs between the PCs. Generally, they peak at PC 2 (Figure 14) and decay with decreasing PC order (Figure S4). In this sequence it is first the coarse structures with stronger contrasts that are described and then the more fined detailed structures which tend to be smoother (Figure 8 and Figure 14). The "spatial average" pattern of the PC 1 monopole generally exhibits contrasts on low to intermediate level compared with the "strongest contrast" pattern of the PC 2 dipole.

Substantial accumulation of variance in the leading PCs is commonly interpreted as indication for dominant processes or modes of the analysed system. In particular the combination with distinct PC patterns exhibiting strong contrasts is highly suggestive. The results demonstrate that both aspects are rather limited indicators and not sufficient for such interpretation.

Quite the contrary, if spatially homogeneous autocorrelation is dominant in the data, both have to be expected.

Note also that the effect of the autocorrelation is spread over all PCs. Thus, for process identification etc., it is the question whether the features of interest cause signatures (spatio-temporal heterogeneities) distinct enough to be salient against the homogeneous background (Cahalan et al., 1996). Next question is whether they get clearly assigned to single PCs or whether they are as well smeared over several, if not all, PCs.

When rotation of PCs (Section 5.2.2) is applied to improve the identifiability of the features of interest it should be considered that the domains size must exceed the correlation length of the respective features. Otherwise, the simplification of the PC patterns by the rotation will not be meaningful for this purpose.

Figure 13 Variance representation of the ten leading PCs modelled with the analytic DD reference method using an isotropic exponential covariance model, nine different spatial correlation lengths and the domain boundaries (a) square, (b) rectangle and (c) triangle from Figure 6. All cells within the boundaries were used. The scale of the Y-axis is square root transformed for better readability.

Figure 14 As in Figure 13 but for the contrasts of the DD patterns.

**4.4. Effectively degenerate multiplets**

Effectively degenerate multiplets are PCs with consecutive ranks, often PC pairs, which are not well separated by the PCA (North et al., 1982). They are indicated by noticeably similar eigenvalues (fractions of explained variance) considering their position in the ranking of the PCs, e.g. PC 2+3 in Figure 13a and PC 3+4 in Figure 13b, both for spatial correlation length of 10 cells. Within the subspace spanned by the multiplets' eigenvectors their rotation is arbitrary. All eigenvectors of the multiplet are needed to adequately describe the multiplets' subspace (North et al., 1982). Consequently, the multiplet should not be split if the subsequent use of the PCs requires an adequate representation of multiplet subspaces, for example in rotation (Jolliffe, 1987, 1989), interpretation of PC patterns or if PCA is used as preprocessing step for other analyses. In particular, special care has to be taken that the truncation point of a PCA does not split a multiplet (North et al., 1982), especially when the amount of variance associated with the excluded PCs is relatively large compared to the amount of variance extracted by those retained. The concept of effectively degenerate multiplets (short: effective multiplets, as in Wilks (2006)) is closely related to degeneracy of eigenvalues. For clarification we provide a brief introduction in Appendix B.

[revised manuscript text omitted]

A real-world data case is shown in Figure 16. Three sets of spatial PC patterns with square, rectangular and triangular domains were derived from raster of monthly precipitation sums from the years 1991 to 2020 in northeast Germany (Section 2.2). The square domain is the 200 km × 200 km square from the 1 km × 1 km precipitation grid in Figure 3. The rectangular and triangular domain were fitted in the square domain, analogue to the proceeding with the synthetic examples (Figure 6). Thus, the data sets consist of time series with 360 months length and 40 000 locations in case of the square domain, and 20 000 locations in case of the rectangular and triangular domains. The DD of the spatial PC patterns is clearly visible (Figure 16). Visually, the spatial PC patterns appear as noisy variants of the already well-known Buell patterns (Figure 1 and Figure 8). The very strong accumulation of variance in the centred monopole pattern of PC 1 (Table 3, Figure 16) is another indication for DD. Thus, in this case the quick check already clarifies the DD of the PCA results.

If the subdomains are of similar size, the focus is primarily on the domain shape aspect. Analogue, the PCA results can be checked for dependency from the selected domain size. However, we assume that commonly an analysis is focused on a specific scale and the domain size as well as the interpretation of results fit to that scale. Thus, usually the dependency from the domain's shape should be more an issue than the dependency from its size.

Figure 16 Overview of the leading ten spatial PC patterns from the PCAs of the precipitation data with the square, rectangular and triangular domain. The location of the square domain is marked in Figure 3. The two other domains are fit in the square domain.

|           | PC 1  | PC 2 | PC 3 | PC 4 | PC 5 | PC 6 | PC 7 | PC 8 | PC 9 | PC 10 |
|-----------|-------|------|------|------|------|------|------|------|------|-------|
| Square    | 79.96 | 4.93 | 3.69 | 1.58 | 1.16 | 0.90 | 0.55 | 0.44 | 0.44 | 0.37  |
| Rectangle | 83.22 | 5.06 | 2.49 | 1.30 | 0.92 | 0.67 | 0.61 | 0.42 | 0.36 | 0.32  |
| Triangle  | 82.79 | 5.04 | 3.01 | 1.35 | 0.78 | 0.61 | 0.56 | 0.47 | 0.39 | 0.31  |

Table 3 Fractions of assigned variances in percent from the PCAs of the precipitation data with the square, rectangular and triangular domain (Figure 16).

**5.1.2. Comparison with DD reference patterns**

DD reference patterns can be tailored for defined spatial domain and spatial correlation properties of a data set. Spatial PC patterns can be visually compared against the reference or checked for significant deviations from the reference with the congruence coefficient or simple correlation analysis (Table 1). With the stability analysis of the stochastic method (Figure 7) or the confidence intervals of the analytic method (Figure S6) it can be identified for each PC rank which time series length is required to reach stable and clearly defined DD patterns. Consequently, PCA results from (observed) data sets with identical spatial domain and spatial correlation properties but shorter time series have to be interpreted with the reservation that the DD might be stronger than the comparison with the reference suggests.

As a real-world data case we look again at the precipitation PCAs (Figure 16). DD reference patterns were fitted for all three domains using an isotropic spherical covariance model and the analytic method (Figure S7). The spatial patterns of the leading precipitation PCs exhibited strong similarity with their DD reference counterparts (Table 4), clearly indicating DD. For the first four PCs, the main difference was the separation of PCs 2+3 from the square domain in the precipitation PCAs (Figure S8) which form a multiplet in the DD reference (Table S1). Meaning, although typical DD patterns occurred, the deviations of the precipitation PCA patterns from the pure theoretical DD case were strong enough to result in a clear ranking. In accordance with the findings of the synthetic experiments (Figure 13), the very large fraction of variance assigned to PC 1 of the precipitation PCAs (Table 3) is reflected in the very large theoretical correlation length of the DD reference (Figure S7), being substantially longer than the domain size.

That the patterns of the low ranked PCs exhibit stronger deviations from the DD reference than those of the leading PCs is no indication against DD. Recall that PCs should not be analysed in isolation, but only in reference to all PCs with preceding ranks. If the leading PCs exhibit DD, DD for the whole sequence of PC patterns can be concluded. It is not necessary to find DD reference patterns that perfectly fit to the patterns of the low ranked PCs for such conclusion. Because of the more finely structured spatial patterns, it can be expected that the patterns of the low ranked PCs from real-world data will deviate stronger from the DD reference than those of the leading PCs. Thus, the comparison with the DD reference confirmed the finding of strong DD from the visual comparison of the patterns from the three domains (Section 5.1.1).

We introduced building DD reference patterns for data sets exhibiting isotropic spatial but no temporal autocorrelation. It enables to test the null hypothesis that the spatial PC patterns from observed data merely result from simple isotropic spatial autocorrelation between random white noise time series. The main feature of the null hypothesis is the ratio of spatial correlation length to the domain size, in particular to the distances between the data set's locations. To our knowledge, such test was suggested first by Cahalan et al. (1996). They fitted models to observed precipitation and temperature data and compared the eigenvalues and spatial PC patterns of observed and modelled data. Significant differences between the two eigenvalue spectra were considered to be "signal" and indicative for spatial anisotropies and inhomogeneities, "inhomogeneous processes", combined space and time correlation, or (secular) trends.

However, DD is not restricted to the isotropic case (see "directional functions" in Buell (1975)). An anisotropic example for our three basic domains is given in the supplements. Compared with the "default" isotropic case, the DD patterns are distorted according to the direction and the ratio between longest and shortest spatial correlation length of the anisotropy (Figure S9 vs Figure 8). The spatial PC patterns tend to stabilize for shorter time series length (Figure S10 vs. Figure 7) and the PCs which form degenerated pairs are better separated (see the bigger differences between the fractions of assigned variance and the smaller magnitudes of the standard deviation patterns in Figure S11 vs. Figure S2). Both aspects reflect that the anisotropy gives a less ambiguous orientation for the DD sequence.

Elaborating on the DD of PC patterns from data sets with homogeneous autocorrelation in space and time is beyond the introductory scope here. However, spatially inhomogeneous temporal trends are indicative for distinct processes, modes or alike. They are likely to spread over more than one PC (Hannachi et al., 2007; Hannachi, 2007) and to affect the variance distribution among the PCs (Vejmelka et al., 2015). Thus, if the goal is not DD assessment but to construct reference patterns for the identification of distinct features, they should be considered.

DD reference patterns are rather well behaved. The main decisions for their construction are the choice between an isotropic or an anisotropic model, and the selection of the correlation length. The first primarily defines the typical patterns that appear (e.g., Figure 8 versus Figure S9), the second the variance distribution (Figure 13, Section 4.3). In comparison, the effects of different spatial covariance model types like exponential, gaussian or spherical are less important. For practical applications, the comparison with the spatial patterns is the main point rather than the exact reproduction of the variance distribution. A perfect fit is not required. The spatial patterns are very similar for a wide range of correlation lengths. This holds in particular for those of the leading PCs which are commonly used in practical applications.

|           | Precip PC      | 1    | 2       | 3       | 4    | 5    | 6       | 7       | 8       | 9        | 10      |
|-----------|----------------|------|---------|---------|------|------|---------|---------|---------|----------|---------|
| Square    | aCC            | 1    | 0.95 \3 | 0.95 \2 | 0.95 | 0.94 | 0.94    | 0.81    | 0.84    | 0.74 \10 | 0.54 \9 |
|           | $\mathbb{R}^2$ | 0.77 | 0.91 \3 | 0.90 \2 | 0.90 | 0.88 | 0.88    | 0.65    | 0.71    | 0.54 \10 | 0.29 \9 |
| Rectangle | DDref PC       | 1    | 0.99    | 0.98    | 0.94 | 0.91 | 0.85    | 0.67    | 0.77    | 0.44 \7  | 0.35 \9 |
|           | $\mathbb{R}^2$ | 0.73 | 0.98    | 0.96    | 0.88 | 0.83 | 0.72    | 0.45    | 0.59    | 0.19 \7  | 0.12 \9 |
| Triangle  | DDref PC       | 1    | 0.95    | 0.93    | 0.96 | 0.91 | 0.86 \7 | 0.89 \6 | 0.69 \9 | 0.77 \8  | 0.69    |
|           | $\mathbb{R}^2$ | 0.76 | 0.90    | 0.86    | 0.93 | 0.82 | 0.74 \7 | 0.80 \6 | 0.47 \9 | 0.59 \8  | 0.48    |

Table 4 As in Table 1 but for the comparison of the spatial PC patterns from the precipitation data (Figure 16) and the corresponding DD reference patterns (Figure S7).

**5.2. Approaches to diminish DD**

**5.2.1. Subsampling of the spatial domain**

Analysing a subsampled data set with enlarged minimal distance between the locations can be used to diminish the DD of the PCA results. Reducing the symmetry of the analysed spatial domain can remove effective multiplets. Both can help to carve out features other than DD. On the other hand, informative local details might be filtered out together with the excluded locations. If there is still DD, the new DD patterns of the subsampled data set might be harder to recognize visually because of the smaller number of locations per area. The selected minimal distance, respectively the selection of locations, is critical for the analysis. Depending on the choice, different features in the results might stick out, get diminished or even disappear. In any case, the spatial resolution of the analysed data set has to be considered in the interpretation of the results. Also, only stable PC patterns should be used to draw conclusions on the analysed system. The stable PC patterns are those which are rather insensitive to the specific selection of analysed locations. They can be identified by comparing the PCA results from different subsamples (Smirnov, 1973; Lins, 1985a; Lehr and Lischeid, 2020).

**5.2.2.** Rotation of PC eigenvectors**

Another option that can diminish DD is to rotate the eigenvectors from the PCs of interest (Richman, 1986; Dommenget, 2007; Compagnucci and Richman, 2008). Often unrotated PCA results exhibit DD patterns, whereas rotated PCA seem to be less affected (Richman, 1986; Huth and Beranova, 2021). This finding is supported by experiments using synthetic data. Compagnucci and Richman (2008) analyzed different synthetic sequences of basic sea level pressure flow patterns ("plasmodes"). The unrotated S-mode patterns were systematically affected by DD. In the rotated variants the DD patterns vanished.

Exemplarily, we varimax rotated the leading spatial PC patterns of the precipitation PCAs (Figure 16) in three variants, using the first two PCs (2rPCs), the first three PCs (3rPCs) and the first four PCs (4rPCs) (Figure 17). No multiplets were split by the rotations (Figure S7) to ensure that the results of the rotation were not affected by multiplet effects (Section 4.4). As expected, the variance distribution among the rotated PCs (Table 5) was much more even compared to the unrotated PCs (Table 3). The newly assigned fractions of variance did not any longer decrease continuously with the PC ranks in all cases.

However, that depends on the software being used. Some packages will sort the rotated PCs by their variance explained. Note that the fractions of variance that are assigned to distinct patterns, for example to the diagonal gradient of the triangular domain, depend on the number of PCs that are rotated (Table 5). The magnitude of the pattern contrasts was more evenly distributed among the rotated PCs (Table S2) than among the unrotated PCs (Table S3). Most of the rotated patterns exhibited only positive or only negative loadings (Table S2), indicating a more "simple structure" (Section 3.5; Richman, 1986) than in the unrotated patterns (Table S3).

In all three varimax rotation variants, the patterns were clearly dependent on the domain geometries (Figure 17). While the dominant PC 1 monopole of the unrotated PCA disappeared, the new dominant patterns are gradients reflecting the domain shape. For example, the patterns of the 2rPCs variant showed gradients from southwest to northeast in the square domain, from west to east in the rectangular domain and from north-west to south-east in the triangular domain. The gradients of the square domain from the 4rPCs variant reflect the rotational symmetry of the square (Figure 17a, right panel). The gradients of the rectangular and triangular domain associated with the major fractions of variance (Table 5) depict in all three rotation variants the longest extent of the domain (Figure 17bc). Thus, here, varimax rotation was not successful in resolving DD.

Note however, that for the introductory scope here, the experiment with the three varimax rotation variants was kept deliberately simple. It is not a full-scale rotation study that would involve finding the best suitable set of rotated PCs for physical interpretation or alike. We did neither investigate which number of rotated PCs resulted in more or less DD, nor did we aim to find an optimum number of rotated PCs with respect to DD. Therefore, the results and their significance are limited. It cannot be ruled out that the DD of the presented results might be an effect of keeping too few PCs (underfactoring). In other words, unrelated signals might be forced on a single PC causing the observed DD. Keeping too many PCs (overfactoring), on the other hand, might split the correlation patterns, respectively the representation of a hydrological feature. However, overfactoring is not an issue here due to the small number of PCs retained.

Despite its limitations the experiment shows that the application of varimax rotation per se – that is without optimizing the number of rotated PCs – is not necessarily sufficient to resolve DD. For practice this implies that, whereas rotated eigenvectors are generally considered to be less prone to DD than unrotated ones (Richman, 1986; Wilks, 2006), it cannot be taken for granted that simply taking the first few PCs of an analysis and varimax rotating them suffices to resolve DD.

Except from being less prone to DD (Richman, 1986; Wilks, 2006), rotated PCA results were found to be robust against sampling errors in case of eigenvalue degeneracy (Richman, 1986), and more robust against spatial (Richman, 1986) and temporal (Cheng et al., 1995) subsampling. Note that spatial instability may be inter-related to DD. In particular, subdomain instability can be a corollary of DD (Richman, 1986). Rotation can support the interpretation of effective multiplets if the resulting PCA patterns are of more simple structure (Jolliffe, 1987; 1989). Rotating only multiplet members limits thereby the changes in the PC properties (Section 3.5) to the multiplet (Jolliffe, 1989; 1995), in particular the dependency of the rotated PC patterns on which and how many PCs are rotated. That rotation results typically in a rather even variance distribution between the PCs is not much of an issue, because the variance in the multiplet is already rather equally spread between the multiplet members before rotation (Jolliffe, 1989). Also, the effects of the scaling of the eigenvectors (meaning the loss of uncorrelatedness and orthogonality in the spatial and/or temporal patterns, see Section 3.5) are diminished, because the eigenvalues of the multiplet are of similar size (Jolliffe, 1989; 1995).

Rotated PCA results were also found to be easier to interpret physically (Richman, 1986). Rotation can be used to systematically relax distinct PCA constraints that hamper physical interpretation (Hannachi et al., 2007; Monahan et al., 2009) by choosing between orthogonal and oblique rotation and selecting a simplicity criterion that suits best to the analysed system. In the aforementioned analysis of synthetic sea level pressure flow patterns, Compagnucci and Richman (2008) found the rotated PC patterns to be superior in depicting the "true" flow patterns. In a study using atmospheric reanalysis data, Huth and Beranova (2021) compared the spatial patterns from four PCA derived modes of climatic variability with autocorrelation maps of the analysed data to identify the true modes of climatic variability. Only the one mode based on rotated PC patterns (North Atlantic Oscillation) corresponded well to underlying autocorrelation patterns, the modes based on unrotated PCA did not. However, these studies indicating that rotated PC patterns are more suitable for physical interpretation focused primarily on atmospheric mode detection.

For future work, we suggest to perform a study similar to Compagnucci and Richman (2006), but with a hydrological focus. Synthetic data from a hydrological simulation model could be analyzed, to test which hydrological features of the model can be uncovered by the patterns of the PCs. The test data could be, for example, spatially distributed groundwater level series simulated with a groundwater model. The experiments could be used to compare the performance in hydrological feature identification of unrotated versus rotated PCA, different orthogonal and oblique rotation methods, but also of S-mode versus T-mode PCA (Richman, 1986; Compagnucci and Richman, 2006; Isaak, et al., 2018) and different scaling of the eigenvectors (Jolliffe, 1995; Wilks, 2006).

Figure 17 Leading varimax rotated spatial PC patterns from the PCAs of the precipitation data with the square, rectangular and triangular domain (Figure 16). The rotation was performed with the first two PCs (2rPCs), the first three PCs (3rPCs) or the first four PCs (4rPCs).

|           | 2rPCs |       | 3rPCs |       |       | 4rPCs |       |       |       |  |
|-----------|-------|-------|-------|-------|-------|-------|-------|-------|-------|--|
|           | rPC 1 | rPC 2 | rPC 1 | rPC 2 | rPC 3 | rPC 1 | rPC 2 | rPC 3 | rPC 4 |  |
| Square    | 44.84 | 40.05 | 32.95 | 30.92 | 24.71 | 27.77 | 24.44 | 20.92 | 17.03 |  |
| Rectangle | 44.00 | 44.28 | 18.36 | 37.43 | 34.97 | 19.28 | 36.85 | 33.16 | 2.78  |  |
| Triangle  | 46.69 | 41.14 | 26.61 | 34.26 | 29.96 | 4.78  | 32.93 | 26.36 | 28.11 |  |

Table 5 Fractions of assigned variances in percent from the varimax rotated spatial PC patterns from the precipitation PCAs (Figure 17).

**6 Conclusion**

Spatial patterns from S-mode PCA are regularly used for hydrological interpretations. In such analysis, homogeneous spatial correlation between the data sets` time series results in spatial PC patterns that are determined by the size and shape of the analysed spatial domain (domain dependence: DD). DD patterns are distinct, with strong gradients and contrasts. We showed that DD can come together with substantial accumulation of explained variance in the leading PCs. Thus, in contrast to what one might expect, neither distinct spatial PC patterns nor large fractions of explained variance in the leading PCs do necessarily indicate dominant hydrological processes or hydrologically meaningful properties. In addition, DD can induce effectively degenerated multiplets (effective multiplets). Without knowledge about DD, the multiplets can be misinterpreted as indication for complex spatio-temporal features. Without knowledge about multiplets, the multiplet members can be mistaken as effects of independent hydrological processes. Without knowledge about the effects of multiplets, DD can be overlooked because the degeneracy can mask the expected DD patterns.

In summary, if DD is predominant, the spatial PC patterns do not reflect the hydrological functioning of the analysed system but rather the functioning of the PCA within the context of the data set's spatial domain. Ignoring DD and effective multiplets easily leads to wrong hydrological interpretations. Consequently, DD should be considered for any application in which the PCs are used to draw conclusions about spatially distinct properties of the analysed system. In other words, it should be checked whether the spatial PC patterns differ significantly from patterns that result from the trivial case of nearby locations being homogeneously more related than those further apart. If PCA is used purely for data reduction, DD is of no interest as the patterns are never examined; they serve only as an efficient set of basis vectors. If, however, the subsequent use of the PCs requires an adequate description of multiplet subspaces, for example if PCA is used as preprocessing step for other analyses, care should be taken that no multiplet is split by the selection of retained PCs. This applies in particular when the amount of variance associated with the excluded PCs is relatively large compared to the amount of variance extracted by those retained. Classical Buell patterns (PC 1: "mean behaviour", PC 2: gradient along the longest extent of the domain, lower ranking PCs: regular multipoles) and leading PCs with remarkably similar eigenvalues (effective multiplets) are an alert for DD. However, deviating patterns or clearly separated PCs are no contra-indication. DD patterns are original for every combination of spatial domain and spatial correlation properties. Thus, visual detection of DD is rather limited. Still, visual comparison of the spatial PC patterns from subdomains with markedly different shapes and/or sizes is practical as quick qualitative check.

To test whether spatial PC patterns differ significantly from DD patterns, reference patterns can be used as null hypothesis in pairwise statistical tests. For most practical applications checking the first few leading PC patterns should be sufficient. If the spatial PC patterns do not differ significantly from DD reference patterns, we recommend to report that and stop any interpretation of individual spatial PC patterns as distinct hydrological features.

We presented two methods to produce DD reference patterns. For the introductory purpose, we focussed on the stochastic method. The comparison of data sets simulated with identical spatial domain and spatial correlation properties showed directly the ambiguity of the PC ranking within DD induced multiplets, including the variations of the predominant patterns.

Furthermore, working with simulated data is less abstract than working with the analytic covariance matrix. For practical applications the analytic method is preferable. Its short computation time is a big advantage, especially when producing DD reference patterns for data sets with many locations.

Passing the check for DD and accounting for effective multiplets in the selection of the PCs are necessary but not sufficient conditions to assure physical meaningfulness. When single PCs, or combinations of PCs, are assigned to distinct hydrological features, it should be carefully considered whether the S-mode PCA constraints (i) successive maximization of variance on the PCs, (ii) orthogonality of spatial PC patterns and (iii) linear uncorrelatedness of temporal PC patterns support such interpretation. The spatio-temporal PC patterns should not only be checked for resemblance with the postulated features, but also the invariance of the spatial and temporal PC patterns against subsampling should be approved. Building on this study, a next research task could be to conduct systematic experiments with synthetic test data derived from hydrological simulation models to evaluate which PCA modes, rotation methods and scaling of the eigenvectors work best for hydrological feature identification.

[revised manuscript text omitted]

A selection of scripts accompanying this technical note is freely available at https://doi.org/10.5281/zenodo.11213430 (Lehr, 2024). It contains: (1) a demo in which the DD of PCs is demonstrated by visual examination of the spatial PC patterns from single simulated data sets, (2) an implementation of the stochastic DD reference method (Section 3.2.1), and (3) an implementation of the analytic method (Section 3.2.2) based on Dommenget (2007) and the associated Matlab scripts. The user can define domains with distinct sizes and shapes, and the spatial correlation properties. The scripts and their documentation can directly be used for educational purposes. We recommend going first step by step through the demo to get into the functioning and logic of the scripts. For the demo and the stochastic reference script, it is best to start with the pdf documentation which includes a formatted version of the script, extra annotations and sample results. All scripts are written in R (R Core Team, 2019).

**930 Data availability**

Data sets containing the spatial domains and spatial correlation properties used in this technical note can be produced with the associated scripts. The grids of the monthly precipitation sums are freely available at the German Weather Service (https://opendata.dwd.de/climate\_environment/CDC/grids\_germany/monthly/hyras\_de/precipitation/). Here, the version HYRAS-DE-PR v6.0 was used.

**935 Acknowledgements**

We thank Philipp Rauneker and Katharina Brüser from the Leibniz Centre for Agricultural Landscape Research (ZALF) and Marcus Fahle from the German Federal Institute for Geosciences and Natural Resources (BGR) for the fruitful discussions. Furthermore, we thank Gunnar Lischeid from ZALF for the in-depth discussions and detailed comments.

**Financial support**

The authors thank the ZALF and the University of Potsdam for the possibility to use the institutional resources.

---

## Author Response (AR2)

**Reply to referee 1 – Report #2**

Comments of referee 1 from the online report mask are in black.

Comments of referee 1 annotated in the pdf of the referee report are in brown.

The comments from the two sources were not in all cases identical. In case

- (1) they were identical, we noted that and omitted the duplicate,
- (2) only one or two sentences, or introductory words were differing, we omitted the duplicate and stated the differences,
- (3) there were more differences than in (2), we listed the complete comments from both sources.

Our explanatory notes for the first two cases are in italic.

Replies of the authors (AR) are in blue.
* * *
**Comments of referee 1**

Review of Technical Note: An illustrative introduction to the domain dependence of spatial Principal Component patterns by Lehr and Hohenbrink.
* * *
The manuscript is much improved. The previous sections (prior to those added in the latest revision) are acceptable. The main revision is now to the part of the paper added in the latest revision, the application of the Varimax rotation to assess DD. Additionally, some of the comments in the new material ought to be addressed in the conclusions.

To begin, the paper is important. I suspect it will be read by those in the hydrologic sciences and beyond. My take on this is as a general issue for EOF/PC analyses is that the analyst can go one of two ways: (1) apply EOF/PCA as a data reduction technique and then stop without any physical interpretation. This seems to be the minority of applications. However, in those cases, DD is of no interest as the patterns are never examined; they serve only as a set of basis vectors. If DD patterns are the most efficient basis set, so be it. (2) However, the majority of analyses set out to interpret individual EOFs, and then DD becomes critical. In those cases, how can the analyst know the EOFs reflect patterns embedded in the covariance matrix rather than DD? You provide a set of patterns and an algorithm in this paper. Are you suggesting that algorithm be applied for everyone using EOFs for interpretation? Given all the results herein, you are in a position to make such a statement or suggest a concrete plan for those using EOF/PCA. I urge you issue a statement. More specifics follow.

AR: Thank you very much for the overall positive assessment and acknowledging the importance of our paper. Yes, our understanding is the same and we generally agree with the two ways you are pointing out. Regarding case (1) we like to add the restriction that if the subsequent use of the PC patterns requires an adequate description of multiplet subspaces, care should be taken that no multiplet is split by the selection of retained PCs (see Section 4.4). We

added corresponding statements in the conclusions at the end of the 2nd paragraph in Lines 806–809:

"If PCA is used purely for data reduction, DD is of no interest as the patterns are never examined; they serve only as an efficient set of basis vectors. If, however, the subsequent use of the PCs requires an adequate description of multiplet subspaces, for example if PCA is used as preprocessing step for other analyses, care should be taken that no multiplet is split by the selection of retained PCs."

**and in Lines 816-818:**

"If the spatial PC patterns do not differ significantly from DD reference patterns, we recommend to report that and stop any interpretation of individual spatial PC patterns as distinct hydrological features."

See also our reply to points 5a and 2 of your comments to the conclusion section.

What we are suggesting is that if spatial PC patterns are used for interpretation, the patterns should be checked for DD before. We state this in Lines 803–806 of the conclusions. We recommend "visual comparison of the spatial PC patterns from subdomains with markedly different shapes and/or sizes … as quick qualitative check" (end of 3rd paragraph of the conclusion). And we recommend DD reference patterns as null hypothesis to "test whether spatial PC patterns differ significantly from DD patterns" (beginning of the 4th paragraph of the conclusion).

On the discussion of rotation and Varimax, there are some important issues that need to be addressed, mostly in Section 5.2.2. Here the level of detail should be commensurate with the rest of the manuscript. Further, the details for Varimax that match the details in the rest of the paper are partly or fully missing (see comments that follow). For example, where is there a Table equivalent to that of Table 4 for Varimax? The r and aCC are needed to evaluate the results.

AR: We highly appreciate the sincereness and precision of your comments and the amount of work you spent. It considerably helped to improve our manuscript, in particular with respect to completeness, precision and explicit statements. However, some of your suggestions point conceptually to a different direction than it is our intention for our study. As we understand it, some of the details you are missing is because of that. To be clear, our goal is not (i) to perform and include a full-scale rotation study here, and we do not want (ii) to go for physical interpretation of individual spatial PC patterns, or to analyse their validity in this regard. In our opinion, your suggestions regarding these two aspects provide material for at least one standalone paper. For us here, it is beyond our scope. And we think it would overload our manuscript. We want to emphasize that we consider these suggestions interesting, substantial and worth implementing in independent studies. We included some of your ideas in our suggestion for future work (Lines 774–780 and at the end of the conclusions).

Here, our goal is to introduce DD and its functioning to the PCA users in the hydrological community. All experiments and presented results focus on that. For this purpose, we decided to use mainly unrotated spatial PC patterns. In our paper, varimax rotation is briefly discussed as one aspect among several in the section 5 on "approaches to consider DD". Therefore, its sub-section should be commensurate to the other sub-sections of section 5. We have deliberately weighted the sub-topics in this way.

Table 4 provides the comparison of the spatial PC patterns from the precipitation data (Figure 16) and the corresponding DD reference patterns (Figure S7). The reasoning is to show how well the patterns of the precipitation data match with those of the DD reference. The DD reference patterns here and throughout the paper are calculated for unrotated correlation matrix based PCs. An equivalent comparison with varimax rotated PCs would require to calculate DD reference patterns for varimax rotated PCs, depending on the k number of PCs selected for rotation. To our knowledge, calculating DD reference patterns for varimax rotated PCs has not been done before. It could be an interesting objective in a future study focussing on rotated PCA (and physical interpretation of PC patterns). However, it is clearly beyond the scope of our study here. We think, it would overload the paper, distract from our intended focus and therefore impair the readability and practical value for PCA users in hydrology.

Here are the specifics (also annotated in the reviewed manuscript)

Line 346: added "as much as the data permit"

AR: We included this specification.

Line 349: So, in this case, the unit length eigenvectors are being rotated, I presume? Please verify/clarify.

AR: Yes, you are right. We stated this explicitly in Line 366 at the end of this section, after introducing varimax and pointing to the importance of the eigenvector scaling for the results in Lines 361–363. See also our reply to your comment to Line 743.

Line 664: A legend is needed to evaluate the magnitudes of the EOFs.

AR: Figures 16 and 17 are overview plots just as Figures 8, 9, 10, 11 and Figures S7 and S9 in the supplement. The focus is on the spatial patterns only - not their magnitudes. We discussed this already in the first review round with referee 2, see our comment there.

The magnitudes of the unrotated and the rotated spatial PC patterns from the precipitation PCAs are provided in form of the contrasts, that is the differences between the minimum and maximum of the scaled eigenvectors (correlation loadings), in Table S3 and S2 (see Lines 742–743). This is in line with the rest of the paper where we showed the contrasts in Figure 14 and Figures S4 and S5 in the supplement.

Line 682: "while" replaced with "although"

AR: We changed that.

Line 738: This new section really needs attention to bring it to the quality level of the rest of the paper. Where is the Table 4 equivalent for the Varimax PCs? The r and aCC is needed to evaluate the results.

AR: Please see our second reply to your introductory comment.

Line 743: "while" replaced with "whereas"

AR: We changed that.

Line 743: Are those patterns unit length eigenvectors or eigenvectors scaled by the square root of the corresponding eigenvalue?

Line 743: "patterns" Are the patterns unit length eigenvectors are scaled eigenvectors? It matters as Varimax was built for the latter. It might work for the former though that has not been tested widely. In either case, if one is to compare the Varimax patterns to the unrotated patterns, the vector scaling (or lack thereof) should be the same. If it is unit length eigenvectors, that might require a sentence stating that the vast majority of applications of Varimax (in all research domains of study) use scaled eigenvectors.

AR: The patterns are eigenvectors scaled by the square root of the corresponding eigenvalue (what we term "correlation loadings" here). We stated that this is the most popular scaling variant for varimax rotation in Lines 361–363. We agree that for comparison of rotated and unrotated patterns, the vector scaling should be the same. Therefore, we use the same scaling (correlation loadings) throughout the manuscript (see Lines 232–233 and 366).

Line 745: Were the unit length eigenvectors or the eigenvectors postmultiplied by the corresponding (square root of the) eigenvalue used here?

If the former (unit length eigenvectors or EOFs), are these comparable to the patterns in the earlier Figures?

If the latter (scaled eigenvectors), they won't be comparable to the EOF patterns but perhaps more amenable to the rotation algorithm by virtue of being scaled.

Some comments on the comparability are in order.

AR: We used the same scaling (eigenvectors scaled by the square root of the corresponding eigenvalue, "correlation loadings) throughout the manuscript to ensure comparability. See also our reply to your comment to Line 743.

Line 746: Was there any testing (e.g., matching the patterns to those of your covariance functions) to optimize the selection of the k=2, 3 or 4 PCs? Selecting 2 - 4 seems arbitrary. Similarly, do we know if perhaps 5, 6, ..., 10 PCs are a better choice? As you state in the paper, the eigenfunctions change as a function of k PCs retained and rotated. Further, Figures 8, 9, 10, 11 and 16 all show up to 10 patterns. Why were 2, 3 and 4 selected here? Doesn't that smaller number bias the results? I suggest rotating sets of 2, 3, ..., 10 PCs and then comparing each to the corresponding covariance matrix vector that is indexed to the largest magnitude PC loading for each PC. Then calculate the congruence coefficient for each vector for each set. By doing so, you can select the one set of k PCs that is best supported by the matrix that is diagonalized.

*The identical comment was given in the online form to* Line 746: "2 rPCs..." *and* Lines 746-747: "The first four precipitation PCs..."

AR: There was no testing to optimize the selection of the k number of PCs for rotation performed. The purpose of the varimax rotation experiment and section in our manuscript is not to identify the one set of k PCs that is best suited for physical interpretation of the rotated precipitation PC patterns or alike. It seems your suggestion is pointing in this direction. We also do not want to perform or include a full-scale rotation study here.

Within the introductory scope of our manuscript, varimax rotation is briefly discussed as one aspect among several in the section 5 on "approaches to consider DD". In this setting the results of three varimax variants are exemplarily shown. To our knowledge, in most hydrological PCA studies only the first few leading PCs are used. Therefore, we selected for our simple experiment the first 2, 3 and 4 PCs for rotation and compared the results. The exemplarily character of our selection is explicitly stated at the beginning of the 2nd paragraph of section 5.2.2 in Line 736.

The spatial patterns of the three varimax rotation variants clearly depend on the shape of the domain. We stated this and did not expand the varimax analysis any further. To clarify that we performed only a simple experiment here, and not an extensive rotation study, we (i) changed the wording in Lines 748–749 to:

"Thus, in our simple experiment here, varimax rotation was not successful in resolving DD."

and (ii) added a few sentences on the limitations of our simple experiment, including some of your ideas, at the end of the 3rd paragraph of section 5.2.2 (Lines 755–759):

"Note however, that for the introductory scope here, the experiment with the three varimax rotation variants was kept deliberately simple. Therefore, the results and their significance are limited. It is not a full-scale rotation study that would involve finding the best suitable set of rotated PCs for physical interpretation or alike. Also, we did not investigate which number of rotated PCs resulted in more or less DD, nor did we aim to find an optimum number of rotated PCs with respect to DD.".

See also our reply to your second comment to Lines 746–747 and our second reply to your introductory comment.

We like to add that we think that the proceeding you are suggesting and the concept behind is rather unknown in hydrology. We assume it is more common in atmospheric sciences for atmospheric mode detection. Thus, for us here, it would require another introduction to a rather unknown concept in our already introduction rich manuscript. It would also add another level of complexity that distracts from the focus and intention of our study. We think that experiments like those we suggest for future work at the end of the conclusions and in the last paragraph of section 5.2.2 (Lines 774–780) with synthetic data from hydrological simulation models "to test which hydrological features of the model can be uncovered by the patterns of the PCs" would be easier accessible for most hydrological readers.

Lines 746-747: This justification is unnecessary and problematic. Once a rotation is applied, it is for physical interpretation of the EOFs in terms of the matrix from which they were derived (e.g., covariance, correlation). This is a **validity issue** whereas eigenvalue degeneracy is a reliability issue. If there is little or no validity to the individual patterns, the reliability makes no difference. The set of EOFs, be they from well-separated eigenvalues, a mixture of well-separated and degenerate eigenvalues or from a full set of degenerate eigenvalues is immaterial because in all those situations, the k EOFs retained is the set that maximizes the total variance retained. It variance maximization is all that is needed from an EOF/PCA, then DD is never an issue. End of story.

However, here, and more generally for those applying EOF/PCA, they are invoking Varimax that is meant to, "*To support the interpretability of the results*" (*line 350 herein*). By invoking interpretability, the concept of what EOF/PCA is used for changes dramatically. No longer is

the data compression the sole goal. Now, the goal is to find physical signals in the individual EOFs. **Finding interpretable signal requires extra work.** The two pieces of that extra work are: (1) finding the best k PCs (something mentioned in this paper) and (2) ensuring that there are valid patterns within those k PCs (something not done yet in this work). Valid patterns arise from what the EOF/PCA does: it diagonalizes a covariance (or correlation) matrix either explicitly or implicitly into a set of k eigenpatterns and each of those k eigenpatterns are examined for physical relevance. Unless those patterns reflect the variation in the matrix that is being diagonalized, they are not valid. What the analysis does is collect a rectangular data sample, relate it via the covariance matrix and then find the most important vectors that summarize that covariance matrix. So step (2) is ensuring each of those patterns relates sufficiently well to the covariance pattern (or correlation pattern) with a congruence exceeding some threshold. In this paper you seem to adopt an aCC magnitude of 0.85 as a minimum for acceptable match. Therefore, you should rotate your various sets of k PCs over a range, such as k=2, 3, ...,10, and calculate the aCC match to the covariance vector indexed to the largest maximum magnitude loading. If there is a set where each EOF matches at that aCC level are larger (I recommend 0.9, but 0,85, 0.9 or 0.95 might be workable), it is an acceptably **valid set.** At that juncture the k PCs in the set can be examined for DD. However, if there are one or more PCs in some sets that fail to meet the aCC match threshold, that whole set is invalid. It is also possible that no set has all EOFs, in all the various k sets tested, exceeding the aCC threshold and then the Varimax solution fails for all sets of k=2, 3, ..., 10 PCs retained. In that case, another rotation might be applied to diagnose if the orthogonality constraint was the problem. If the situation persists, another analysis technique may be needed for physical interpretation. It is also possible there is a large degree of noise in the data, the covariance matrix fails to capture and physical signals that exceed the noise and the EOF/PC approach will not work for such a noisy data set. It is possible no other technique will be able to extract signal from a data set comprised mostly or noise. One need to test (1) and (2) quantitatively to know when to proceed and when to quit.

AR: We agree that, usually, rotation of PCs is applied for physical interpretation of the PCs and their patterns. However, in our study, this is not the case. We did not perform any physical interpretation of the PCA results in the paper and we never meant to. The focus is to introduce DD to the PCA users in the hydrological community. All experiments and presented results focus on that. In this setting, we performed here the comparison of the spatial patterns from unrotated and rotated PCA only, and not their interpretation (see also our replies to your first comment to Lines 746-747 and the second part of your introductory comment). To ensure that the results of the rotation were not affected by multiplet effects, we took care that no multiplets were split by the rotations. To clarify this, we changed the addressed sentence (Lines 737–738 in the revised manuscript) to:

"No multiplets were split by the rotations (Figure S7) to ensure that the results of the rotation were not affected by multiplet effects (Section 4.4)."

The extra work you are suggesting goes clearly beyond the scope of our study. We think, executing the work plan you are suggesting would be sufficient for a standalone paper, focussing on the physical interpretability of (rotated) PCA results. It could be applied for example very well in the performance assessment of our suggestion for future work in Lines 774–780 and at the end of the conclusions.

To clarify that DD is not an issue if the PCA is purely used for data reduction, we added a sentence at the end of the  $2^{nd}$  paragraph of the conclusions. Please see also our replies to the first part of your introductory comment and point 5a of your comments to the conclusion.

Line 753: The sign of the loadings in any loading vector (or EOF) is arbitrary. You can multiply any vector by -1 to make them all (or virtually all) positive with zero loss of meaning or variance explained.

*The identical comment was given in the online form to* Line 753: "only positive or..."

AR: We agree. Our sentence simply states that the loadings of most PCs were of one sign only.

Line 753: added "in"

AR: We added that.

Line 755: See comments below why the analysis of Varimax is problematic as shown. This can be fixed.

AR: We did not aim to identify the best set of k PCs for physical interpretation or alike or to perform a full-scale rotation study. Please see our replies to your two comments to Lines 746–747.

Line 755" "In all three rotation variant...". First, there is no rotation variance, only a change on the number of patterns for a single rotation, Varimax. If you had compared Varimax or several other rotation Variants (e.g., Promax, Oblimin) then the statement would have been correct. Please fix the terminology and the comments left about testing for the optimum k PCs hold.

AR: To be more precise, we changed the phrase into "In all three varimax rotation variants ...".

Lines 757-758: Two comments:

- 1. I don't think you can say this because you have not testing if 2 PCs retained is most appropriate. That testing is essential with rotated PCs.
- 2. Please square your statement with the one highlighted on line 765. Is there more, the same or less DD with rotation in your experiment?

*The identical comment was given in the online form to* Lines 757-758: "Thus, in our case here, varimax rotation was not..."

AR: 1. We changed the wording to

"Thus, in our simple experiment here, varimax rotation was not successful in resolving DD."

2. We added a few sentences at the end of the section to clarify the limitations of our study.

Please see also our reply to your first comment to Lines 746–747.

Line 758: This is vague, at best. What does this mean? Do the Varimax rotated PC loadings resemble the correlations more of less than the unrotated PC loadings? If not, state so. If so, then this suggests it is the underlying correlation patterns that are affected by the DD. If that is the case, the whole idea of explicitly or implicitly using correlations (or covariances) and a matrix to relate the data comes into question. Further, because PCA diagonalizes the correlation (or covariance) matrix, what advice can you offer the readers.

The identical comment was given in the online form to Line 758: "Instead, the patterns..."

AR: We did not investigate the underlying correlation patterns of the precipitation data in the way you are suggesting in your first comment to Lines 746–747, nor did we aim to identify the set of PCs (rotated or unrotated) that resemble best these patterns. Please see our replies to your two comments to Lines 746–747 for our reasoning, and the text we added in the revised manuscript to clarify the scope and limitations of our study (Lines 755–759).

Lines 759-760: See previous comment. The potential problem is not clear. Either this gradient reflects the correlation patterns or it does not. If it is the former, then the problem is not with Varimax but how the domain shape captures the underlying correlation pattern. If it is the latter, then it is a failure of the Varimax rotation. This needs to be addressed.

Lines 759-760: "...the new dominant patterns are..." See previous comment. The potential problem is not clear. Either this gradient reflects the correlation patterns or it does not. If it is the former, then the problem is not with Varimax but how the domain shape captures the underlying correlation pattern. If it is the latter, then it is a failure of the Varimax rotation. The relationship between the covariance/correlation patterns and selecting a domain shape should be addressed.

AR: See previous reply. We did not perform any identification of underlying correlation patterns here. Please see also our replies to your two comments to Lines 746–747 for our reasoning, and the text we added in the revised manuscript to clarify the scope and limitations of our study (Lines 755–759).

Line 760: Again, the analyst is not going to select multiple sets of Varimax PCs to move forward. They should keep a range of solutions and test those to find the optimum set for the single best set of k PCs that captures the patterns embedded in the covariance matrix. Here you show k=2, 3 and 4 with no such tests and for all we know the optimum number for k should be some other number (e.g., 5, 6, ..., 10). If that is the case, perhaps what you term DD is the distortion caused by under-retention (or over-retention) of PCs.

*The identical comment was given in the online form to* Line 760: "...4rPC variant..."

AR: We simply stated that in our simple experiment the spatial patterns of the varimax rotated PCs were clearly dependent on the domain geometries. We did not test this for other numbers k or speculate whether this would be different for other numbers k. We included a statement on the limitations of our simple varimax experiment at the end of the 3rd paragraph of section 5.2.2. Please see also our reply to your first comment to Lines 746–747.

Line 763: "while" replaced with "whereas"

AR: We changed that.

Line 764: Two comments:

1. This general web page is not a peer-reviewed publication. It is more of an opinion piece because, other than a single citation on sampling errors, there are no citations to any of the statements.

2. The statement about the possible shortcoming of Varimax EOFs from that web page is: "Still, REOF methods have issues. The patterns are may still be domain dependent and the initial number of EOFs retained is arbitrary"

Note there are no citations in this web page that support this quoted claim.

The identical comment was given in the online form to Line 764: "NCAR 2013". In bullet point one of the comment, there was in addition the following sentence at the end: ... Such flimsy non-peer reviewed web pages are best omitted.

AR: Thanks for pointing out this weakness. We removed the citation of the web page.

Line 765: Is this the finding in this paper or from the two studies cited, or both?

Line 765: "Except for being less prone to DD,..." Is this the finding in this paper or from the two studies cited, or both? Please clarify.

AR: The statement refers to the last sentence of the previous paragraph and the citation there. In the revised manuscript, we included the citation here again to clarify.

"Except from being less prone to DD (Richman, 1986; Wilks, 2006), rotated PCA results ..."

Lines 788-790: A legend is needed to evaluate the magnitudes of the EOFs.

Lines 788-790: Figure 17. A legend is needed to evaluate the magnitudes of the EOFs. Further, see other comments on adding and testing for validity 5rPCs, ..., 10rPCs.

AR: Regarding the magnitudes, see our reply to your comment to Line 664. Regarding your comments on adding and testing for validity of further PCs please see our reply to your second comment to Lines 746–747.

Line 794: Although this Table shows the changes in the variance of the Varimax PCs, we have no idea if 2, 3 or 4 is the optimal numbers. See above comments for the suggested remedy.

*The identical comment was given in the online form to* Line 794: Table 5.

AR: Please see our reply to your first comment to Lines 746–747.

**Line 795 onward:**

The identical comment was given in the online form to Line 795 onward: Conclusions. ... For better readability we replied directly after each bullet point.

Somewhere in the conclusions there should be advice given to those who apply EOF/PCA for hydrological applications. Specifically:

1. How to select a domain shape? Some of the domains used herein, such as the triangle are less likely to be used compared to a rectangle, for example. However, the bigger take away is that examining the covariance functions for their shapes and then selecting a domain shape is important and a logical extension of your work. Foe pattern that is zonally oriented might be best captured by either a hemispheric domain or a large rectangular patch.

- AR: 1. We assume that in practice, the analysed domain will be mainly determined by the research task. Our take away for the practice is that for a given domain, we recommend to perform a visual comparison of the spatial PC patterns from subdomains with clearly different shapes as quick qualitative check for DD (see the end of the 3rd paragraph of the conclusions).
- 2. If their EOFs look essentially the same as your DD patterns, what do you recommend. Do they forge ahead with physical interpretation despite the similarity or terminate the analysis?
- AR: 2. We added the following statement to the conclusions in Lines 816–818: "If the spatial PC patterns do not differ significantly from DD reference patterns, we recommend to report that and stop any interpretation of individual spatial PC patterns as distinct hydrological features." Please see also our reply to the first part of your introductory comment.

However, we like to point out, that it is not sufficient to visually compare the spatial PC patterns with the patterns shown in our paper here. DD patterns are original for every combination of domain and spatial correlation properties. We stated this explicitly in the introduction Lines 112–113 and the conclusions Lines 812–813 of the manuscript.

- 3. If their EOFs look somewhat like the same as your DD patterns, but also have substantial deviations, what do you recommend. Do they forge ahead with physical interpretation despite the similarity or terminate the analysis? How can they separate the covariance part of the pattern from the DD part?
- AR: 3. We think this depends on the purpose of the analysis. For example, Dommenget (2007) "suggested using the spatial PC patterns from an analytic covariance matrix as null hypothesis to find spatial PC patterns "that are most distinguished from those of the null hypothesis". These so called Distinct Empirical Orthogonal Functions (DEOFs) are derived by rotating the eigenvectors of the observed data to maximum difference in explained variance between the EOFs of observed data and those of the analytic covariance matrix." (see the 3rd paragraph in Appendix A, Lines 849–852). However, we don't want to get more specific or explicit in our conclusions regarding this aspect, because we did not show this in our manuscript. It would require a thorough analysis focussing on signal identification and interpretation of the results, which is not our focus here (see our second reply to your introductory comment and our replies to your comments to Lines 746–747).
- 4. If their EOFs look different from your DD patterns, is it fair to say there is no DD effect?
- AR: 4. No. DD reference patterns are original for every combination of spatial domain and spatial correlation properties (see Lines 112–113 in the introduction and Lines 812–813 in the conclusions). Visual similarity or dissimilarity with patterns from different domains is not sufficient (see the 3rd paragraph of the conclusions.)
- 5. Despite the DD patterns you show throughout the paper:
- a. If the EOFs are used purely for data reduction, with no interpretation of the individual patterns, such as a preprocessing step for other analyses to lower dimensionality (e.g., to make the computing more efficient), should they care about DD at all? My thought is, no, DD makes zero difference it the EOF is used solely for data reduction with no physical interpretation of the individual patterns. However, that is not the way the majority of EOF/PC studies are analyzed.

AR: 5a. We agree, with the restriction that if the subsequent use of the PC patterns requires an adequate description of multiplet subspaces, care should be taken that no multiplet is split by the selection of retained PCs (see Section 4.4). To clarify this, we added a statement at the end of the 2nd paragraph of the conclusions in Lines 806–809:

"If PCA is used purely for data reduction, DD is of no interest as the patterns are never examined; they serve only as an efficient set of basis vectors. If, however, the subsequent use of the PCs requires an adequate description of multiplet subspaces, for example if PCA is used as preprocessing step for other analyses, care should be taken that no multiplet is split by the selection of retained PCs."

Please see also our reply to the first part of your introductory comment.

b. If the individual EOFs are to be used for physical interpretation, only then does DD become critical. In those cases, **how** can the analyst know the EOFs reflect patterns embedded in the covariance matrix rather than DD? You provide a set of patterns and an algorithm in this paper. Are you suggesting that algorithm be applied for everyone using EOFs for interpretation? Given all the results herein, you are in a position to make such a statement. I urge you to do so.

AR: 5b. Prior physical interpretation, we suggest (i) visual comparison of the spatial PC patterns from domains with clearly different shapes as quick qualitative check (see the end of the 3rd paragraph of the conclusions) and (ii) the comparison with DD reference patterns to check for significant differences (see the beginning of the 4th paragraph of the conclusions).

Line 882: "while" replaced with "yet"

AR: We changed that.

A couple of additional issues:

1. Please add color legends to Figures, where possible.

AR: For the overview figures we deliberately did not include legends. See also our reply to your comment to Line 664.

2. A few minor wording issues (e.g., use of "while" that should be reserved for time comparisons). I attempted to locate and replace those.

AR: Thank you. We changed that.

---

## Author Response (AR3)

**Reply to referee 1 – Report #1 from 2025-09-15**

Comments of referee 1 are in black.

The comments from the online report mask and those annotated in the pdf of the referee report were identical. The bold highlighting in some comments were taken from the pdf referee report.

The referee structured the comments in three sections. To simplify cross-referencing, the authors added the section titles: A - Introductory comments, B - Comments on the last set of author replies, and C - Comments on the last version of the manuscript; and added the respective capital letter to the enumeration of the comments. Thus, for example comment B-3 refers to the third comment in section B.

Replies of the authors (AR) are in blue.
* * *
**Comments of referee 1**

Review of latest revision of Technical Note: An illustrative introduction to the domain dependence of spatial Principal Component patterns by Lehr and Hohenbrink.

**A - Introductory comments**

I appreciate all of the work put into this manuscript. Given the latest responses to the last revision, I believe that the present manuscript is about as complete as I can hope for, with the exception of cleaning up and further explaining a few issues, mostly in the section on rotation. Given the hesitation to investigate fully the domain effects on rotated solutions, I believe there are three good options. 1. Remove the incomplete rotation section. 2. Leave the very limited experiments using the single algorithm (Varimax) over a limited number of PCs AND issue appropriate caveats, rather than the vague sentences in this last version (see my comments in the annotated manuscript). 3. Complete a comprehensive section of rotation using several algorithms and testing for the optimal number of PCs to retain and test those for DD and for validity. I believe that options 1 and 2 are more viable given the comments in the last response about the manuscript size and your intended scope. That is fine as there are other paths to finalize this manuscript.

AR: Thank you very much for your positive evaluation and appreciation. We likewise appreciate all the work you spend for reviewing and improving our manuscript. We also see mainly the three options you are pointing out and agree that options 1 and 2 are more suitable for our scope. We decided to go for option 2. See also our further replies.

The present version could stand as an important contribution to the literature, with appropriate caveats issued. If you wish to proceed to a larger manuscript, it is important to be fully prepared to investigate all the possible permutations arising from each step when rotated solutions are investigated. Those permutations would involve running a much more detailed set of experiments for the number of EOFs/PCs to retain, as that number now effects the pattern morphology (unlike the unrotated solutions where increasing the number retained adds a new

pattern but does not change the previous ones), assessing the degree of simple structure in their data sets and then selecting the optimal rotational algorithm from a sizable number of algorithms (again, this step requires validating many sets of patterns to the largest correspondence to the similarity matrix from which those patterns were derived). Given all the domain shapes, such a set of experiments is cumbersome so, unless you are really motivated to go that route, I don't feel comfortable prescribing that as the required bar and deem the present manuscript is acceptable after the issuance of some caveats. Specifically, issue caveats that the analyses presented for Varimax have no way to deconvolute the effects of keeping too few PCs or too many PCs from DD and temper comments such as "the Varimax way of displaying DD" (lines 751-752). For all we know, it has nothing to do with Varimax but is a function of keeping too few PCs and forcing multiple correlation modes onto too few PCs. Acknowledge the possibility. Not investigating it is okay but, then, either don't mention it or list it as one of a number of possibilities.

AR: Thank you for the positive assessment of our work and providing context for the caveats you suggest. The outline of the experimental study you are suggesting is what we meant by "full-scale rotation study" in our last replies (see also Lines 758–759). We agree that such work would be cumbersome, but valuable addition. Given the extent of the work we find it more suitable as standalone future study. Correspondingly, and in line with your suggested option 2 above, we issued several caveats in the revised manuscript (see our replies to your further comments).

In particular, we included (i) a caveat that the presented analyses did not cover the effects of keeping too few or too many PCs (Lines 761–766), (ii) acknowledged the possibility that multiple correlation modes were forced onto too few PCs (Lines 761–763) and (iii) deleted the phrasing "Varimax way of displaying DD". For the details regarding these points, please see our replies to your comments C-8 to C-12.

**B** - Comments on the last set of author replies**

Here are my comments on the last set of replies:

B-1. "If PCA is used purely for data reduction, DD is of no interest as the patterns are never examined; they serve only as an efficient set of basis vectors. If, however, the subsequent use of the PCs requires an adequate description of multiplet subspaces, for example if PCA is used as preprocessing step for other analyses, care should be taken that no multiplet is split by the selection of retained PCs."

Response: I think as a general statement that is solid advice. However, there are at least two other considerations:

(a) Small magnitude eigenvalues are thought to be associated with noise. Rarely, if ever, if that assumption tested. Small scale signals that extract little variance would be indistinguishable from noise through investigating eigenvalue magnitude, particularly without resampling (there is no resampling in the North et al. test).

(b) Ignoring point (a), the effects of the infusion of noise into the signal+noise PC patterns may or may not be an issue depending upon the amount of variance extracted by those retained. For example, if the retained PCs account for 95% of the total variance, the remaining 5% would be expected to have a minimal effect on the patterns retained, particularly if rotated. In contrast, if that percent variance were 60%, there exists a larger risk of rotation being effected by the noise introduced.

That said, the conservative advice may be safe as it should prevent problems regardless of the total amount of residual noise or if some signal is being discarded.

AR: We addressed your point (b) by extending the sentence in Lines 561–563 to

"In particular, special care has to be taken that the truncation point of a PCA does not split a multiplet (North et al., 1982), especially when the amount of variance associated with the excluded PCs is relatively large compared to the amount of variance extracted by those retained."

and added a sentence at the end of the second paragraph in the conclusion (Lines 826–827):

"This applies in particular when the amount of variance associated with the excluded PCs is relatively large compared to the amount of variance extracted by those retained."

B-2. "If the spatial PC patterns do not differ significantly from DD reference patterns, we recommend to report that and stop any interpretation of individual spatial PC patterns as distinct hydrological features."

Comment: This is good advice, though providing some thoughts on "significantly" or how to assess that would be helpful.

AR: We added a specification in the first sentence of the paragraph from the addressed sentence (Lines 833–834) to clarify that we mean significant differences in the sense of statistical testing here:

"To test whether spatial PC patterns differ significantly from DD patterns, reference patterns can be used as null hypothesis in pairwise statistical tests."

A similar specification was added in the Methods Section 3.2, please see the revised manuscript in Line 240. In our study, we used simple t-tests (see the last sentence of Section 3.3).

B-3. "What we are suggesting is that if spatial PC patterns are used for interpretation, the patterns should be checked for DD before. We state this in Lines 803–806 of the conclusions. We recommend "visual comparison of the spatial PC patterns from subdomains with markedly different shapes and/or sizes ... as quick qualitative check" (end of 3rd paragraph of the conclusion). And we recommend DD reference patterns as null hypothesis to "test whether spatial PC patterns differ significantly from DD patterns" (beginning of the 4th paragraph of the conclusion)."

Comment: Yes, this is good advice as the first step. In practice that null hypothesis would have an alternative that the patterns are different (the alternative hypothesis), a statistical test is than brought to bear on the sample pattern and the probability of a Type I error arises is produced (perhaps with a yes/no decision). One possibility (or many) to avoid a purely subjective determination would be to relate each EOF/PC pattern to the patterns derived in your

manuscript for the best matching domain shape. One could apply the aCC statistic and (because it arises from an unknown distribution) apply a permutation test to determine the p-value.

AR: In principle yes, but there is one important difference. What we suggest here is not to compare the PC patterns from other analysis with the sample patterns derived in our manuscript for the best matching domain shape. Instead, we suggest to fit DD reference patterns to the specific spatial domain that is analysed because "DD patterns are original for every combination of spatial domain and spatial correlation properties" (Lines 112–113, 830–831). We present two methods to do so (fifth paragraph of the conclusion). Then a statistical test can be used to assess the PC patterns for significant differences with the specifically fitted DD reference patterns. We agree that applying a permutation test to determine the p-value of the significance test would be one of the possibilities. Here, we used simple t-tests.

B-4. "AR: We highly appreciate the sincereness and precision of your comments and the amount of work you spent. It considerably helped to improve our manuscript, in particular with respect to completeness, precision and explicit statements. However, some of your suggestions point conceptually to a different direction than it is our intention for our study."

Comment: That is a fair statement. To run a proper set of DD tests on rotated solutions would require being fully prepared to investigate all the possible permutations arising from each step when rotated solutions are investigated. Those permutations would involve running a much more detailed set of experiments for the number of EOFs/PCs to retain (k), as that number now effects the pattern morphology (unlike the unrotated solutions where increasing the number retained adds a new pattern but does not change the previous ones), assessing the degree of simple structure in their data sets and then selecting the optimal rotational algorithm from a sizable number of algorithms (again, this step requires validating many sets of patterns to the largest correspondence to the similarity matrix from which those patterns were derived). Given all their domain shapes, such a set of experiments is cumbersome. Perhaps list this as an area of investigation for others or for a future paper?

AR: We agree. We provided a suggestion for future research in the last paragraph of Section 5.2.2. Compared to the previous version, we slightly extended the wording at the end of the paragraph to:

"... The experiments could be used to compare the performance in hydrological feature identification of unrotated versus rotated PCA, different orthogonal and oblique rotation methods, but also of S-mode versus T-mode PCA (Richman, 1986; Compagnucci and Richman, 2006; Isaak, et al., 2018) and different scaling of the eigenvectors (Jolliffe, 1995; Wilks, 2006)."

Our suggestion is revisited in prominent position as final sentence of the conclusion:

"Building on this study, a next research task could be to conduct systematic experiments with synthetic test data derived from hydrological simulation models to evaluate which PCA modes, rotation methods and scaling of the eigenvectors work best for hydrological feature identification."

Given the above statements it seems there are three possibilities to satisfy Occam's razor in two broad categories:

1. Pure data reduction with no physical interpretation: keep the process simplest and don't bother with DD investigations or with post processing with rotation. That is now discussed in the manuscript.

- 2. Data reduction with physical interpretation:
- a. No longer can the simplest process be assumed to work without further investigation, so test for DD. If DD is present in a "significant" amount, either stop, try step 2b or try a methodology other than EOF/PCA.
- b. Investigate if the patterns reflect the patterns of variation in the similarity matrix to ensure physically valid patterns.
- (i) If there is not a significant amount of DD present (that is why we are at step 2b) but there is a "significant" correspondence for each unrotated pattern retained to those in the similarity matrix, then interpret each pattern in the set of k patterns or try 2biii to determine if the matches become more significantly matched to the similarity matrix.
- (ii) If there is not a significant amount of DD present (again that is why we are at step 2b), but there is not a "significant" correspondence for every unrotated pattern retained to those in the similarity matrix, then (a) reduce the number of EOF/PCs retained and retest each in the set of patterns for correspondence to the similarity matrix or (b) try 2biii to determine if the matches become more significantly matched to the similarity matrix patterns.
- (iii) If there is not a "significant" correspondence for every unrotated pattern retained for any set of k EOFs/PCs retained, or one is interested in determining if the matches are improved by post-processing the unrotated EOF/PC patterns, then try rotating the patterns and assessing those rotated patterns for a "significant" correspondence for each pattern retained to those in the similarity matrix, then interpret each of the rotated patterns.
- (iv) If either step 2bii or step 2biii fails to find a set of k PCs where each shows insignificant amount of DD and each is significantly related to the patterns on the similarity matrix, then try a different non-EOF/PC approach or stop.

I believe some comments on this could be added to Lines 774–780 and at the end of the conclusions.

AR: Thank you for summarizing the possibilities you see for including the aspect of DD in a PCA in a decision tree. We agree, that this often helps to clearly see the available possibilities. However, in the decision tree you are suggesting the assessment of the physical validity of the spatial PC patterns (2b) is an integral part. We did not cover this here. Given your earlier comments (second part of the introductory comment A, first part of this comment here), we think that you agree that including such work is material for a standalone manuscript and not required here. Therefore, we think that the remainder of the content you are summarizing is already explicit and clear enough in our manuscript. Thus, instead of repeating that content and adding another paragraph in the conclusion, we rather like to keep it as it is.

We think your point 1 is clearly covered in the conclusion in Lines 823–824:

"If PCA is used purely for data reduction, DD is of no interest as the patterns are never examined; they serve only as an efficient set of basis vectors."

Focusing on what we presented in our manuscript here, that is, leaving aside the test for physically valid patterns (your matches with the similarity matrix), the aspects of your point 2 are covered in Lines 820–823:

"Consequently, DD should be considered for any application in which the PCs are used to draw conclusions about spatially distinct properties of the analysed system. In other words, it should be checked whether the spatial PC patterns differ significantly from patterns that result from the trivial case of nearby locations being homogeneously more related than those further apart."

and in Lines 834-836:

"If the spatial PC patterns do not differ significantly from DD reference patterns, we recommend to report that and stop any interpretation of individual spatial PC patterns as distinct hydrological features."

The possibility to apply another method than PCA, we did not explicitly state. We assume that this is obvious to the reader.

B-5. "Table 4 provides the comparison of the spatial PC patterns from the precipitation data (Figure 16) and the corresponding DD reference patterns (Figure S7). The reasoning is to show how well the patterns of the precipitation data match with those of the DD reference. The DD reference patterns here and throughout the paper are calculated for unrotated correlation matrix based PCs. An equivalent comparison with varimax rotated PCs would require to calculate DD reference patterns for varimax rotated PCs, depending on the k number of PCs selected for rotation."

Comment: Because the analysis of precipitation or any other field cannot be assumed to be well-rendered by Varimax, such an experiment, if it were made, should examine a range of rotations. Investigation of various Python and R libraries (e.g., the R package GPArotation) suggest dozens of possibilities (other than Varimax).

AR: Thank you for this addition. We understand it as a valuable hint in the context of future studies.

B-6. "To our knowledge, calculating DD reference patterns for varimax rotated PCs has not been done before. It could be an interesting objective in a future study focussing on rotated PCA (and physical interpretation of PC patterns)."

Comment: See the comment above where eliminating DD or reducing DD to an "insignificant" amount is a necessary but not sufficient step for physical interpretation. To meet the more stringent level of sufficient for physical interpretation, each of the retained EOF/PC patterns must reflect the correlation (or covariance) patterns well. However, because that sufficiency applies to both unrotated and rotated EOFs/PCs, some mention of that could be made because it is possible a set of patterns may not show DD but may not reflect well the patterns of data similarity (e.g., recall the heavy lift issues of maximum variance and orthogonality that are perhaps related to but not the same as DD).

AR: We agree. We have statements on this in the last paragraph of the conclusion (Lines 843–848):

"Passing the check for DD and accounting for effective multiplets in the selection of the PCs are necessary but not sufficient conditions to assure physical meaningfulness. When single PCs, or combinations of PCs, are assigned to distinct hydrological features, it should be carefully considered whether the S-mode PCA constraints (i) successive maximization of variance on the PCs, (ii) orthogonality of spatial PC patterns and (iii) linear uncorrelatedness of temporal PC patterns support such interpretation. The spatio-temporal PC patterns should not only be

checked for resemblance with the postulated features, but also the invariance of the spatial and temporal PC patterns against subsampling should be approved."

It resumes the discussion about physical interpretation of spatial PC patterns in paragraphs four to six of the introduction (Lines 68–104). See also our reply to your comment B-4 above.

Regarding the suitability of spatial PC patterns for hydrological feature identification (what you call here sufficiency for physical interpretation) we included a suggestion for future studies performing "systematic experiments with synthetic test data derived from hydrological simulation models" in the last paragraph of Section 5.2.2 and the last sentence of the conclusion.

B-7. "See the comment above where eliminating DD or reducing DD to an "insignificant" amount is a necessary but not sufficient step for physical interpretation. To meet the more stringent level of sufficient for physical interpretation, each of the retained EOF/PC patterns must reflect the correlation (or covariance) patterns well. However, because that sufficiency applies to both unrotated and rotated EOFs/PCs, some mention of that could be made because it is possible a set of patterns may not show DD but may not reflect well the patterns of data similarity (e.g., recall the heavy lift issues of maximum variance and orthogonality that are perhaps related to but not the same as DD)."

Comment" Thanks for the clarification and note that in some fields unit length eigenvectors (e.g., EOFs) are more often used than the scaled version. This has led to a morass in terminology.

AR: We assume your comment refers to our reply to the "scaling of the eigenvector" topic (It seems there was a copy-paste mistake regarding the reference of your comment. The text in quotation marks here is your previous comment). However, thank you for your acknowledgment and hint regarding the different standards in different fields. Regarding the terminology: Yes, we agree. A morass, indeed.

B-8. "AR: There was no testing to optimize the selection of the k number of PCs for rotation performed. The purpose of the varimax rotation experiment and section in our manuscript is not to identify the one set of k PCs that is best suited for physical interpretation of the rotated precipitation PC patterns or alike. It seems your suggestion is pointing in this direction. We also do not want to perform or include a full-scale rotation study here."

Comment: Understood. I left some comments earlier in case you decide to try such an experiment. It would involve more than testing Varimax over a broader range of PCs retained.

AR: Thank you for your comment. We feel appreciated and understood. We share the same understanding that such study would require a systematic and much more detailed analysis than what we did here and what we wanted to present. The extent of such a study gets clear from the second part of your introductory comment A and the first part of your comment B-4 ("... investigate all the possible permutations arising from each step"). Nevertheless, we agree that it would be an interesting task to perform such study in the future and appreciate your comments for this purpose.

B-9. "Thus, in our simple experiment here, varimax rotation was not successful in resolving DD"

Comment: Yes, that is true for the limited scope of your design. I would suggest adding the caveat that your design was not comprehensive, so you can't say if Varimax could reduce DD to an insignificant level.

AR: Thank you. We are aware of the limitation of our design that we have chosen for our scope here and agree that it is good to clearly point this out. We stated this explicitly in Section 5.2.2, Lines 758–761.

"Note however, that for the introductory scope here, the experiment with the three varimax rotation variants was kept deliberately simple. It is not a full-scale rotation study that would involve finding the best suitable set of rotated PCs for physical interpretation or alike. We did neither investigate which number of rotated PCs resulted in more or less DD, nor did we aim to find an optimum number of rotated PCs with respect to DD. Therefore, the results and their significance are limited. ..."

See also our replies to your comments C-8 to C-12.

B-10. "Note however, that for the introductory scope here, the experiment with the three varimax rotation variants was kept deliberately simple."

Comment: Read my comments in the revised manuscript why the deliberately simple experiment cannot deconvolute between the way varimax creates patterns or the way keeping a specific subset of k PCs creates patterns.

AR: For our replies to your comments in the manuscript, please see section C below.

B-11. "Also, we did not investigate which number of rotated PCs resulted in more or less DD, nor did we aim to find an optimum number of rotated PCs with respect to DD.".

Comment: I provided comments in the manuscript about this too. Not having DD patterns (unless that DD pattern happens to be the correlation pattern) is necessary for physical interpretation. However, the sufficiency comes from relating the EOFs/PCs to the correlation matrix. If the EOF/PC patterns are not supported by the correlations, then some other factor(s) are creating them. Most likely these are mis-specifying the domain size (or shape) that fails to capture the data variability scale, maximum variance of the PCs, orthogonality of the PCs or combinations thereof.

AR: For our replies to your comments in the manuscript, please see section C below. That not having DD patterns is necessary but not sufficient condition for physical interpretation is stated explicitly in the last paragraph of the conclusion.

The effects of mis-specifying the domain size with respect to the data variability scale are described in the first paragraph of Section 4.3 and some newly added lines at the end of Section 4.3 in the revised manuscript (see our reply to your comment C-1).

That the PCA features maximization of variance and orthogonality of the PCs constrain physical interpretation of PCA results (beyond DD) is stated in the introduction and the last paragraph of the conclusion. Assessing the suitability of spatial PC patterns for hydrological feature identification (what you address here as sufficiency for physical interpretation) is addressed in our suggestion for future studies in the last paragraph of Section 5.2.2 and the last sentence of the conclusion. See also our replies to your comments B-4 and B-6.

B-12. "AR: We agree that, usually, rotation of PCs is applied for physical interpretation of the PCs and their patterns. However, in our study, this is not the case. We did not perform any physical interpretation of the PCA results in the paper and we never meant to. The focus is to introduce DD to the PCA users in the hydrological community."

Comment: Yes, but my understanding of the conclusions in the manuscript is that if the EOFs/PCs contain significant amounts of DD, they should not be interpreted. I offer up an additional thought on this. If there is not significant DD, there may or may not be validity for other reasons.

AR: We agree. We included many of your thoughts and mentioned further requirements for (the validity of) physical interpretation in the manuscript, for example in the last paragraph of the conclusion. In addition, we included a suggestion for future studies focusing on the identifiability of hydrological signals from PCA results in the last paragraph of Section 5.2.2. In addition, we addressed the aspect of underfactoring/overfactoring in the revised manuscript (see our reply to your comment C-8).

**C - Comments on the last version of the manuscript**

Comments in the revised manuscript:

C-1. Line 508: "4.3. Effects of the domain size and spatial correlation length"

Comment: Yes, and consider the following. For domain size less than or equal to the correlation length, one cannot apply rotation as the goal of rotation is to identify patterns that are subsets of the domain. If the correlations span the domain, there can be no meaningful simplification. Application of rotation in such cases will attempt to simplify patterns that should not be simplified. This is why examining the PC loadings and comparing them to the correlation patterns is so important. Only in cases where the domain size exceeds the correlation length can rotation be examined in a meaningful way for improving the resolving of correlation modes, as finding spatially simplified configurations of the data is supported by the data. [As an aside, small domains, relative to the spatial correlation length, is rarely an issue in weather and climate studies but may be an issue for other fields of study]

AR: We agree. Thank you for this precision. We added a statement on this in Lines 540–542 at the end of Section 4.3.

"When rotation of PCs (Section 5.2.2) is applied to improve the identifiability of the features of interest it should be considered that the domains size must exceed the correlation length of the respective features. Otherwise, the simplification of the PC patterns by the rotation will not be meaningful for this purpose."

C-2. Lines 558-559: "In particular, special care has to be taken that the truncation point of a PCA does not split a multiplet (North et al., 1982)."

Comment: This is reasonable general advice but the likelihood of problems arising is a function of the amount of variance explained by those PCs retained. If there is a substantial percentage of variance beyond the truncation point, there is a probability of more noise contaminating those eigenvectors retained. In such cases, the North et al. test is more critical. If the retained

eigenvectors explain a large majority of the total variance, the small amount of residual variance (thought to represent noise) is much less and the application of the test becomes less important. Further, because rotation is immune to degenerate multiplet distortion for closely spaced eigenvalues, the problem is lessened in that situation.

AR: Thank you for this precision. We added a statement on this in Lines 561–563 and at the end of the second paragraph of the conclusion. See also our reply to your comment B-1.

C-3. Line 719: "Analysing a subsampled data set..."

Comment: Assuming this refers to spatial subsampling rather than time subsampling. Please clarify.

AR: Yes, you are right. For clarification, we changed the title of the subsection in "Subsampling of the spatial domain" and included the specification "analysed spatial" in the second sentence (Line 723):

"Reducing the symmetry of the analysed spatial domain can remove effective multiplets."

C-4. Line 730: "5.2.2. Rotation of PC eigenvectors"

Comment: At this juncture, given the comments in the previous response about a full rotation analysis being beyond the intended scope, it is easier to clean up the details as listed below rather than embark on a full rotation analysis for this particular manuscript. However, should you decide to go there now or later in a separate manuscript, it is important to be fully prepared to investigate all the possible permutations arising from each step when rotated solutions are investigated. Those permutations would involve running a much more detailed set of experiments for the number of EOFs/PCs to retain, as that number now effects the pattern morphology (unlike the unrotated solutions where increasing the number retained adds a new pattern but does not change the previous ones), assessing the degree of simple structure in their data sets and then selecting the optimal rotational algorithm from a sizable number of algorithms (again, this step requires validating many sets of patterns to the largest correspondence to the similarity matrix from which those patterns were derived). Given all their domain shapes, such a set of experiments is cumbersome so, unless the authors are really motivated to go that route, I don't feel comfortable prescribing that as the acceptable bar and deem the present manuscript can be made acceptable. What I do request is that you issue caveats that you have no way to deconvolute the effects of keeping too few PCs or too many PCs from DD and they need to temper comments such as "the Varimax way of displaying DD" (lines 751-752). For all we know, it has nothing to do with Varimax but is a function of keeping too few PCs and forcing multiple correlation modes onto too few PCs. Please acknowledge the possibility. Not investigating it is okay but, then, either don't mention it or list it as one of a number of possibilities.

AR: Ok. Please see our replies to the second part of your introductory comment A and your specific comments to Section 5.2.2 below.

C-5. Lines 739-740: "No multiplets were split by the rotations (Figure S7) to ensure that the results of the rotation were not affected by multiplet effects (Section 4.4)."

Comment: Although it is fine to say this, keep two things in mind:

1. Eigenvalues are a property of unrotated EOFs/PCs. The property is destroyed by rotating. After rotation, the variance on individual PC loading vectors can be tallied by summing the

squared PC loadings for that vector. The total variance for the k rotated PCs will be identical to the k unrotated total variance, but the variance of individual rotated PCs will differ from the variance defined by the eigenvalues on individual unrotated PCs.

2. Rotated PCs are immune from the effects of closely spaced eigenvalues. It is a good practice not to select the truncation point, k, in the middle of a degenerate multiplet if the total variance explained by those k PCs is not large. I have commented on this in other parts of the paper.

AR: Thank you for advice. Regarding the effect of the ratio between the amount of variance associated with the retained PCs versus those excluded (signal to noise ratio) we added statements in Lines 561–563 and at the end of the second paragraph of the conclusion. See also our replies to your comment B-1 and your comment C-2.

C-6. Lines 741-742: "Note, that the newly assigned fractions of variance do not any longer decrease continuously with the PC ranks in all cases."

Comment: That depends on the software being used. Some packages will sort the rotated PCs by their variance explained.

AR: Thanks for the hint. We were not aware of that. We added this information here after a slightly rephrased version of the addressed sentence (Lines 744–746):

"The newly assigned fractions of variance did not any longer decrease continuously with the PC ranks in all cases. However, that depends on the software being used. Some packages will sort the rotated PCs by their variance explained. Note that the fractions of variance ..."

C-7. Line 746: "simple structure"

Comment: Yes. In fact, this fits into the comments made earlier about domain size and rotation. If the domain is too small to expect near-zero values on some subset of locations, rotation should not be applied.

AR: Yes. In our example here, we took care that the correlation length is smaller than the domain size. Regarding your hint that the ratio of spatial correlation length and domain size of the data should be considered when applying rotation, we included a statement at the end of Section 4.3. See also our reply to your comment C-1.

C-8. Line 749: "2rPCs variant"

**Comment:**

I think there are two issues here, in the abstract. Only one might apply to your study, but both should be mentioned because they involve domain shape.

- 1. A. If the data correlation scale is larger than the spatial domain selected, the shape of the domain will affect the EOF/PC patterns because on has sampled a subset of the correlation pattern (think of it as, for example, having a large circular correlation pattern and then applying a triangular cookie cutter to that pattern, distorting the original shape). One cannot determine DD as envisioned in this paper in such a situation. Obviously, the correlation patterns should be examined prior to deciding on a domain shape.
- 1. B. If the data correlation scale is approximately the same spatial scale the domain selected, the shape of the domain may affect the EOF/PC patterns because on has sampled a subset of the correlation pattern (think of it as, for example, having a circular correlation pattern and then

applying a triangular cookie cutter, of approximately the same size, to that pattern), distorting the original shape. One can test for DD in such cases but there could be a competing effect of truncating the physical correlation patterns. Obviously, the correlation patterns should be examined prior to deciding on a domain shape.

In cases 1.A. and 1.B., rotation cannot work as it requires correlation scales to be smaller than the domain size.

- 1. C. If the data correlation scale is smaller than the domain selected, DD can be tested as suggested in the manuscript. Obviously, the correlation patterns should be examined prior to deciding on a domain shape.
- 2. For your experiment with varimax, it is possible 2 rPCs was insufficient and you are forcing more than 2 unique signal patterns onto 2 PCs. We know that keeping too few PCs (known as "underfactoring") forces unrelated signals on a single PC, which could be mistaken for DD. Similarly, keeping too many PCs (known as "overfactoring") splits the correlation patterns (e.g., waves with positive and negative loadings, into two separate PCs, each with one piece of the pattern). Although the overfactored rotated PCs may not show DD, then may be non-physical just the same. **That is why optimizing the k in rotated PCs is so important.** You need to add a caveat that your experiment did not involve this optimization step, so claiming that DD is present is not testable in your framework and is why I object to you claiming (for now, until a full rotated test is performed) that it is a "varimax way of displaying DD". **For all you know, it is a 2rPC way of portraying DD, where there is an unfortunate choice of k, and not the rotation method. The same might hold for 3rPC, ...**

AR: Based on your enumeration, we assume the two issues you ask us to address are:

- (1) the effect of domain size versus correlation length (1.A–1.C) and
- (2) that the missing optimization step for varimax can cause underfactoring or overfactoring that cannot be distinguished from DD with our simple experiment.

In addition, you emphasize once again the importance of checking the PCA patterns for physical validity ("Obviously, the correlation patterns should be examined prior to deciding on a domain shape."). We clarified already in the previous rounds of the review process that we did not include this in our work here. Please see our replies there and to your comments B-4 and B-6.

Leaving the aspect of testing for physical validity aside, we addressed issue (1) in its own Section 4.3. The aspects you are adding to issue (1) in the subitems 1.A–1.C are very specific. Adding them as claims to the manuscript would require (a) further experiments in our manuscript to demonstrate the claimed aspects first or (b) references from the literature. We are not aware of any literature in this regard and we do not want to add any further experiments to our study. Therefore, we stay with issue (1) on the level that is covered in Section 4.3.

Regarding issue (2) we declared that we did not perform a full-scale rotation study that would involve the optimization step you are mentioning (Lines 759–761):

"It is not a full-scale rotation study that would involve finding the best suitable set of rotated PCs for physical interpretation or alike. We did neither investigate which number of rotated PCs resulted in more or less DD, nor did we aim to find an optimum number of rotated PCs with respect to DD. Therefore, the results and their significance are limited."

However, the possibility of underfactoring/overfactoring and the possible interference with DD, were not mentioned so far. To point out the importance of optimizing the number of k rotated PCs in (varimax) rotation also in this regard, we added in the revised manuscript in Lines 762–765 the following lines immediately after the aforementioned lines on the missing optimization step.

"It cannot be ruled out that the DD of the presented results might be an effect of keeping too few PCs (underfactoring). In other words, unrelated signals might be forced on a single PC causing the observed DD. Keeping too many PCs (overfactoring), on the other hand, might split the correlation patterns, respectively the representation of a hydrological feature. However, overfactoring is not an issue here due to the small number of PCs retained."

However, acknowledging the limitations of our experiments and study, we think that we still can say something based on our experiment. Namely, that simply taking the first few PCs and performing varimax rotation - that is without any optimization of the number of rotated PCs - is not per se sufficient to resolve DD. We think this finding is of practical value because we assume that (varimax) rotation is regularly applied without the extensive optimization set up you were outlining.

Therefore, we added the following sentence after the issued caveats in Lines 766–769:

"Despite its limitations the experiment shows that the application of varimax rotation per se – that is without optimizing the number of rotated PCs – is not necessarily sufficient to resolve DD. For practice this implies that, whereas rotated eigenvectors are generally considered to be less prone to DD than unrotated ones (Richman, 1986; Wilks, 2006), it cannot be taken for granted that simply taking the first few PCs of an analysis and varimax rotating them suffices to resolve DD."

All together the former third paragraph of Section 5.2.2. was revised substantially. It now reads (Lines 751–769):

"In all three varimax rotation variants, the patterns were clearly dependent on the domain geometries (Figure 17). While the dominant PC 1 monopole of the unrotated PCA disappeared, the new dominant patterns are gradients reflecting the domain shape. For example, the patterns of the 2rPCs variant showed gradients from southwest to northeast in the square domain, from west to east in the rectangular domain and from north-west to south-east in the triangular domain. The gradients of the square domain from the 4rPCs variant reflect the rotational symmetry of the square (Figure 17a, right panel). The gradients of the rectangular and triangular domain associated with the major fractions of variance (Table 5) depict in all three rotation variants the longest extent of the domain (Figure 17bc). Thus, here, varimax rotation was not successful in resolving DD.

Note however, that for the introductory scope here, the experiment with the three varimax rotation variants was kept deliberately simple. It is not a full-scale rotation study that would involve finding the best suitable set of rotated PCs for physical interpretation or alike. We did neither investigate which number of rotated PCs resulted in more or less DD, nor did we aim to find an optimum number of rotated PCs with respect to DD. Therefore, the results and their significance are limited. It cannot be ruled out that the DD of the presented results might be an effect of keeping too few PCs (underfactoring). In other words, unrelated signals might be forced on a single PC causing the observed DD. Keeping too many PCs (overfactoring), on the

other hand, might split the correlation patterns, respectively the representation of a hydrological feature. However, overfactoring is not an issue here due to the small number of PCs retained.

Despite its limitations the experiment shows that the application of varimax rotation per se – that is without optimizing the number of rotated PCs – is not necessarily sufficient to resolve DD. For practice this implies that, whereas rotated eigenvectors are generally considered to be less prone to DD than unrotated ones (Richman, 1986; Wilks, 2006), it cannot be taken for granted that simply taking the first few PCs of an analysis and varimax rotating them suffices to resolve DD."

As an aside, PCA is regularly used to identify dominant processes or alike in hydrometric monitoring data. That is, non-synthetic data that does not stem from controlled experimental set ups. In this application, the number of dominant processes is usually not known beforehand. Thus, analogue to the selection of the truncation point in unrotated PCA, the question arises how to identify the optimum number of retained PCs there? We think one possibility might be to start with k=2 and stepwise increase k until the patterns of the retained PCs remain rather stable. However, this is one of the aspects that can be analyzed in future studies.

C-9. Lines 751-752: "seemed to be the varimax way of displaying DD."

This is both vague. and not tested (see earlier comment). What is "the varimax way of displaying DD"? Such a statement would suggest that no matter what the underlying correlations, varimax PCs would give the same set of patterns. Is that the case? From what I can see, it might as well be "the k Varimax way" (whatever that is) for a single example and not generalizable to other correlation functions as it is for unrotated EOFs/PCs? See earlier comments on the distortions known to occur when underfactoring/overfactoring. Of course, the added step (not examined in this manuscript) of relating the PC patterns to the correlation patterns will instantly confirm if DD is a potential issue for any analysis (unrotated, rotated). If that comparison has a poor match between the PCs and the correlations, then some other factors (perhaps including DD) might play a role. We just can't tell from this experiment.

AR: Thank you for pointing this out. We agree that it is not possible to generalize from these few examples to a general "varimax way of displaying DD". Our intention was not to make such general claim here. It was meant as a descriptive phrase to summarize that — for the examples we showed — the shape of the domain was clearly reflected in the patterns of the varimax variants (2rPC, 3rPC, 4rPC). So, it was meant as description of our results, not as a generalization. Now that we read the paragraph again, we realized that we stated this already in the first sentence of the third paragraph of Section 5.2.2.:

"In all three varimax rotation variants, the patterns were clearly dependent on the domain geometries (Figure 17)."

Thus, to prevent giving the misleading impression that it would be meant as generalization, we deleted the sentence you were addressing in the revised manuscript. Furthermore, we rearranged some sentences in the paragraph to adjust for the missing sentence. Please see our reply to your precious comment C-8 for how the paragraph now reads.

Regarding the caveats of our simple experiment, including the possible effects of underfactoring/overfactoring, please see our reply to your previous comment C-8.

The issue of the missing check for physical validity ("Of course, the added step (not examined in this manuscript) of relating the PC patterns to the correlation patterns will instantly confirm

if DD is a potential issue for any analysis (unrotated, rotated).") was clarified already in the previous rounds of the review process. Please see our replies there and in this document here (i) the beginning of our reply to your comment C-8 and (ii) our reply to the first part of your introductory comment A.

C-10. Line 758: "deliberately simple."

Comment: You need to tell the reader what "deliberately simple" means. Hopefully, I have left sufficient comments about the critical need to test each k PCs when rotated to determine if any of those sets gives a valid result.

AR: We explicitly listed the limitations of our study in Lines 758–765. To be even more clear, we restructured the paragraph such that the limitations follow immediately after the sentence which states that our experiment is simple. Furthermore, we added in the revised manuscript another caveat addressing the aspect of underfactoring/overfactoring. Please see also our reply to your comment C-8.

The issue of the missing check for physical validity ("Hopefully, I have left sufficient comments about the critical need to test each k PCs when rotated to determine if any of those sets gives a valid result.") was clarified already in the previous rounds of the review process. Please see our replies there and in this document here (i) the beginning of our reply to your comment C-8 and (ii) our reply to the first part of your introductory comment A.

C-11. Line 759: "are limited."

Comment: Suggested addition to this sentence: "or misattributed to the rotation method rather than underfactoring/overfactoring."

AR: In the revised manuscript, we moved the addressed sentence and added a caveat regarding the aspect of underfactoring/overfactoring in the directly following sentences. Please see our reply to your comment C-8 for how the paragraph now reads.

C-12. Lines 760-761: "Also, we did not investigate which number of rotated PCs resulted in more or less DD, nor did we aim to find an optimum number of rotated PCs with respect to DD."

Comment: Lines 760-761: What you have not deconvoluted is the DD effect of selecting a different number of PCs to retain from the rotation method applied. Equally important, because the PCs are not compared to the correlation patterns, the physical validity cannot be established. [I realize you don't want to go there, though if the paper is concluding that EOF/PC patterns with DD should not be interpreted; therefore, failing to tell the reader when the patterns should be interpreted (and the method to support the interpretation) is less than satisfying.] Once again, the conclusion in this paragraph could be because the data are not well represented by the k PCs retained, or the data are not well represented by a varimax rotation, or by a combination of both.

AR: In the setting of the option 2 you were suggesting in your first introductory comment A, we issued further caveats in the manuscript and restructured the paragraph in which the addressed sentence appears. In particular, we issued caveats regarding the aspects of (i) underfactoring/overfactoring ("What you have not deconvoluted is the DD effect of selecting a different number of PCs to retain from the rotation method applied") and (ii) physical validity ("Equally important, because the PCs are not compared to the correlation patterns, the physical validity cannot be established. ...").

For (i) and how the paragraph of the addressed sentence reads now, please see our replies to your comment C-8. For (ii), please see our replies of the previous rounds and to your previous comments in this document here.

C-13. Lines 762-763: "be more robust against spatial"

Comment: It may be the spatial instability is inter-related to DD. Can you comment on that from these experiments?

AR: Yes, it may be. We assume you are referring to Richman (1986)'s statement that "subdomain instability .... is a corollary of domain shape dependence". We are not sure which experiments you mean. We assume it is the one from Section "Subdomain stability" in Richman (1986). However, we included the aforementioned aspect as a general statement (Lines 770–773):

"Except from being less prone to DD (Richman, 1986; Wilks, 2006), rotated PCA results were found to be ... more robust against spatial (Richman, 1986) and temporal (Cheng et al., 1995) subsampling. Note that spatial instability may be inter-related to DD. In particular, subdomain instability can be a corollary of DD (Richman, 1986)."

C-14: Line 763: "and less sensitive to degeneracy (Richman, 1986)."

Actually that study showed essentially no sensitivity to degeneracy for rotated PCs, as the eigenvalues were the same to many decimal places. It did show some sampling variability at very small sample sizes.

AR: Thank you for the precision. We assume you mean that the eigenvectors ("loadings" in Richman (1986)) were the same to many decimal places and that you are referring to the experiment described in the sub-section "Sampling errors" with the results being summarized in Table II (Richman, 1986). There, unrotated PCA and an oblique rotation method (Direct Artificial Personal Probability Factor Rotation (DAPPFR criterion)) were compared in their performance regarding the recovery of population eigenvectors with nearly equal eigenvalues (degeneracy) when using different sample sizes. The DAPFR method was found to be robust against sampling errors even for very small sample sizes. We changed the wording in Lines 770–772 to:

"Except from being less prone to DD (Richman, 1986; Wilks, 2006), rotated PCA results were found to be robust against sampling errors in case of eigenvalue degeneracy (Richman, 1986), and more robust against spatial (Richman, 1986) and temporal (Cheng et al., 1995) subsampling."

C-15. Line 765: "drawbacks of rotation"

These are undefined in this manuscript. Jolliffe's comments applied to the loss of uncorrelatedness and orthogonality in the spatial and/or temporal patterns and the extra work involved in running a rotated analysis. You have previously criticized orthogonality in this paper as one factor hindering interpretation, so it's not clear what is meant here. I do mention the situation when the domain size is smaller than or equivalent in size to the correlation scale as factors against rotation. However, in such cases of small domains (relative to the correlation scale), Jolliffe's suggestion of rotating select PCs will not help physical interpretation, regardless of the eigenvalue spacing.

AR: We wanted to point out that when rotating only multiplet members, it is also only the multiplet members that are affected by the changed PC properties, compared to standard PCA. This means the loss of uncorrelatedness and orthogonality in the spatial and/or temporal patterns (depending on the applied scaling of the eigenvectors), and in particular that the PC patterns depend on which and how many PCs are rotated (Section 3.5). In the context of Jolliffe (1989; 1995) these changes were addressed as "drawbacks". We agree that in our context it makes more sense to use the neutral "changed PC properties". However, Jolliffe (1989; 1995) was also pointing out that if the rotation is restricted to a multiplet the effect of

- (i) rather even redistribution of variance among the rotated PCs is not much of an issue, as the variance was already rather evenly distributed in the multiplet before rotation, and
- (ii) the scaling of the eigenvectors on the results is diminished when the rotation is restricted to PCs that exhibit similar eigenvalue sizes (which is the case for multiplets).

Therefore, we rephrased the sentences in Lines 774–780 to:

"Rotating only multiplet members limits thereby the changes in the PC properties (Section 3.5) to the multiplet (Jolliffe, 1989; 1995), in particular the dependency of the rotated PC patterns on which and how many PCs are rotated. That rotation results typically in a rather even variance distribution between the PCs is not much of an issue, because the variance in the multiplet is already rather equally spread between the multiplet members before rotation (Jolliffe, 1989). Also, the effects of the scaling of the eigenvectors (meaning the loss of uncorrelatedness and orthogonality in the spatial and/or temporal patterns, see Section 3.5) are diminished, because the eigenvalues of the multiplet are of similar size (Jolliffe, 1989; 1995)."

Jolliffe, I. T.: Rotation of Ill-Defined Principal Components, Applied Statistics, 38, 139–147, https://doi.org/10.2307/2347688, 1989.

Jolliffe, I. T.: Rotation of principal components: choice of normalization constraints, Journal of Applied Statistics, 22, 29–35, https://doi.org/10.1080/757584395, 1995.

C-16. Lines 818-820: "If the spatial PC patterns do not differ significantly from DD reference patterns, we recommend to report that and stop any interpretation of individual spatial PC patterns as distinct hydrological features."

Comment: Yes, though I believe you can say more about this. See my previous comments.

AR: We assume you are addressing the issue of "significance" regarding the differences from DD reference patterns. In the revised manuscript, we clarified that we mean "significant differences" in the sense of statistical testing. Please see our replies to your comments B-2 and B-3.
* * *
I hope my comments are useful in finalizing the manuscript.

AR: Yes, they were. Thank you for your work and spending so much of your time!